# Organ aging signatures in the plasma proteome track health and disease

Hamilton Se-Hwee Oh[1,2,3,22], Jarod Rutledge[2,3,4,22], Daniel Nachun[5], Róbert Pálovics[2,3,6], Olamide Abiose[3,6], Patricia Moran-Losada[2,3,6], Divya Channappa[2,3,6], Deniz Yagmur Urey[2,7], Kate Kim[2,3,6], Yun Ju Sung[8,9], Lihua Wang[8,9], Jigyasha Timsina[8,9], Dan Western[8,9,10], Menghan Liu[8,9], Pat Kohlfeld[8,9], John Budde[8,9], Edward N. Wilson[3,6], Yann Guen[6,11], Taylor M. Maurer[5], Michael Haney[2,3,6], Andrew C. Yang[12,13,14], Zihuai He[6], Michael D. Greicius[6], Katrin I. Andreasson[3,6,15], Sanish Sathyan[16], Erica F. Weiss[17], Sofiya Milman[16], Nir Barzilai[16], Carlos Cruchaga[8,9], Anthony D. Wagner[3,18], Elizabeth Mormino[6], Benoit Lehallier[6], Victor W. Henderson[3,6,19], Frank M. Longo[3,6], Stephen B. Montgomery[5,20,21] & Tony Wyss-Coray[2,3,6✉]

Animal studies show aging varies between individuals as well as between organs within an individual[1–4], but whether this is true in humans and its effect on age-related diseases is unknown. We utilized levels of human blood plasma proteins originating from specific organs to measure organ-specific aging differences in living individuals. Using machine learning models, we analysed aging in 11 major organs and estimated organ age reproducibly in five independent cohorts encompassing 5,676 adults across the human lifespan. We discovered nearly 20% of the population show strongly accelerated age in one organ and 1.7% are multi-organ agers. Accelerated organ aging confers 20–50% higher mortality risk, and organ-specific diseases relate to faster aging of those organs. We find individuals with accelerated heart aging have a 250% increased heart failure risk and accelerated brain and vascular aging predict Alzheimer's disease (AD) progression independently from and as strongly as plasma pTau-181 (ref. 5), the current best blood-based biomarker for AD. Our models link vascular calcification, extracellular matrix alterations and synaptic protein shedding to early cognitive decline. We introduce a simple and interpretable method to study organ aging using plasma proteomics data, predicting diseases and aging effects.

Aging results in organism-wide deterioration of tissue structure and function that drastically increases the risk of most chronic diseases. Comprehensive studies of the molecular changes that occur with aging across multiple organs in mice have identified unique molecular aging trajectories and timings[1–4], and susceptibility and resilience to diseases of aging in specific organs such as the brain, heart and kidney varies substantially across the population[6]. However, little is known about how human organs change molecularly with age. A molecular understanding of human organ aging is of critical importance to address the massive global disease burden of aging and could revolutionize patient care, preventative medicine and drug development[7]. In particular, preclinical studies have demonstrated that rejuvenating interventions affect organs differently[3,8]. To translate these studies into transformative medicines, we must be able to accurately measure aging across the body and understand the diversity of human aging not only across but also within individuals.

While many methods to measure molecular aging in humans have been developed[9–11], most of them provide just a single measure of aging for the whole body. This is difficult to interpret given the complexity of human aging trajectories. Some recent methods have used clinical chemistry markers which include some markers of organ function[12–15]. However, many of these markers have low organ specificity, making them difficult to interpret for organ-specific aging. Methods to measure brain aging have used MRI-based brain volume and functional connectivity measurements, which are costly and do not provide molecular insights[16], or have required tissue samples, which prevents their

[1]Graduate Program in Stem Cell and Regenerative Medicine, Stanford University, Stanford, CA, USA. [2]The Phil and Penny Knight Initiative for Brain Resilience, Stanford University, Stanford, CA, USA. [3]Wu Tsai Neurosciences Institute, Stanford University, Stanford, CA, USA. [4]Graduate Program in Genetics, Stanford University, Stanford, CA, USA. [5]Department of Pathology, Stanford University School of Medicine, Stanford, CA, USA. [6]Department of Neurology and Neurological Sciences, Stanford University School of Medicine, Stanford, CA, USA. [7]Department of Bioengineering, Stanford University School of Engineering, Stanford, CA, USA. [8]Department of Psychiatry, Washington University in St Louis, St Louis, MO, USA. [9]NeuroGenomics and Informatics Center, Washington University School of Medicine, St. Louis, MO, USA. [10]Division of Biology and Biomedical Sciences, Washington University School of Medicine, St. Louis, MO, USA. [11]Quantitative Sciences Unit, Department of Medicine, Stanford University School of Medicine, Stanford, CA, USA. [12]Departments of Neurology and Anatomy, University of California San Francisco, San Francisco, CA, USA. [13]Gladstone Institute of Neurological Disease, Gladstone Institutes, San Francisco, CA, USA. [14]Bakar Aging Research Institute, University of California San Francisco, San Francisco, CA, USA. [15]Chan Zuckerberg Biohub, San Francisco, CA, USA. [16]Departments of Medicine and Genetics, Institute for Aging Research, Albert Einstein College of Medicine, New York, NY, USA. [17]Department of Neurology, Montefiore Medical Center, New York, NY, USA. [18]Department of Psychology, Stanford University, Stanford, CA, USA. [19]Department of Epidemiology and Population Health, Stanford University, Stanford, CA, USA. [20]Department of Genetics, Stanford University School of Medicine, Stanford, CA, USA. [21]Department of Biomedical Data Science, Stanford University School of Medicine, Stanford, CA, USA. [22]These authors contributed equally: Hamilton Se-Hwee Oh, Jarod Rutledge. ✉e-mail: twc@stanford.edu

application in living persons[17]. Building off the wealth of literature and clinical practice that uses certain organ-specific plasma proteins to non-invasively assess aspects of organ health, such as alanine transaminase for liver damage, we hypothesized that comprehensive quantification of organ-specific proteins in plasma could enable minimally invasive assessment and tracking of human aging for any organ.

## Plasma proteins can model organ aging

To test this, we measured 4,979 proteins in a total of 5,676 subjects across five independent cohorts (Supplementary Table 1) and mapped the putative organ-specific plasma proteome, which we used to train models of organ aging (Fig. 1a). We mapped the organ-specific plasma proteome using human organ bulk RNA sequencing (RNA-seq) data from the Genotype-Tissue Expression (GTEx) project[18]. We classified genes as 'organ enriched' if they were expressed at least four times higher in one organ compared to any other organ, according to the definition proposed in the Human Protein Atlas[19] (Extended Data Fig. 1, Supplementary Tables 2 and 3, and Methods). We annotated the 4,979 human proteins measured by the SomaScan assay with this information and found 893 (18%) proteins met this definition, with the highest number from the brain. We performed additional quality control to remove proteins with a high coefficient of variation or a low correlation between the two different versions of the SomaScan assay present across our cohorts, leaving us with 4,778 proteins (856 organ enriched, 17.9%) which were used for downstream analysis (Supplementary Fig. 1 and Supplementary Tables 4 and 5).

We and others have previously shown that plasma proteins can be used to train machine learning models to estimate chronological age in independent cohorts[20,21]. For each individual, an aging model produces an 'age gap', a measure of that individual's biological age relative to other same-aged peers based on their molecular profile[9] (Fig. 1a). Several studies have shown associations between age gaps and mortality risk or other age-related phenotypes[9], supporting the hypothesis that the age gap contains information about relative biological aging.

Based on this concept, we trained a bagged ensemble of least absolute shrinkage and selection operator (LASSO) aging models for 11 major organs using the mutually exclusive organ-enriched proteins we identified as inputs (Fig. 1a, Extended Data Fig. 2a,b, Supplementary Fig. 3 and Supplementary Tables 6–8). We chose to restrict our analyses to adipose tissue, artery, brain, heart, immune tissue, intestine, kidney, liver, lung, muscle and pancreas because of their relatively well-understood contributions to diseases of aging and the availability of relevant age-related phenotype data in the tested cohorts. We also trained an 'organismal' aging model using the 3,907 organ-nonspecific plasma proteins as inputs to compare the contribution of specific organs to an organ-shared aging signature, and a 'conventional' proteomic aging model using all 4,778 proteins to compare the organ aging models to a global plasma proteomic aging signature as previously reported[20,21]. We trained our models in 1,398 healthy participants from the Knight Alzheimer's Disease Research Center (Knight-ADRC) cohort (mean age = 75, age range = 27–104) and then tested these models in four fully independent cohorts and in held-out test participants with dementia in the Knight-ADRC. (Fig. 1a, Extended Data Figs. 2 and 3, and Supplementary Fig. 2). All 11 organ aging models and the organismal model significantly estimated age in all five cohorts after multiple test correction (Supplementary Fig. 3b). Organ-specific proteins selected by our approach were highly enriched for organ-specific functions (Supplementary Information).

We observed across all cohorts that individuals with the same conventional age gap had diverse organ aging profiles (Fig. 1b). At the population level, this resulted in a low-to-moderate correlation between the age gaps of different organs (mean pairwise Pearson $r$ = 0.29, Fig. 1c). While organ aging is correlated, the majority of variance in one organ age gap is not explained by others, with the exception of the organismal

and conventional age gaps which were highly correlated. Further, we observed that some individuals had extreme aging in one or more organs relative to the general population (Fig. 1d). We scored individuals across all cohorts as outliers for a given organ age gap using a two standard deviation cutoff and clustered individuals into extreme aging types (e-ageotypes) (Fig. 1e and Extended Data Fig. 4a–c). Although it might be expected that extreme aging in one organ would co-occur with extreme aging in other organs, we instead observed segregation into distinct organ e-ageotypes. We found that approximately 18.4% of individuals had a highly organ-specific e-ageotype that was dominated by the aging of only one organ. Only approximately 1.7% of individuals showed extreme aging in multiple organs; the only multi-organ e-ageotype discovered through unbiased clustering was defined by extreme adipose, brain, conventional, heart, immune, liver and organismal age gaps. These observations suggest that organ age gaps may capture unique aging information, which may have implications for organ-specific biological aging and diseases of aging.

## Organ age predicts health and disease

To assess the relationship between organ age and biological aging, we tested whether organ e-ageotypes were associated with nine age-related disease states for which we had sufficient data in at least two independent cohorts; AD, atrial fibrillation, cerebrovascular disease, diabetes, heart attack, hypercholesterolaemia, hypertension, obesity and gait impairment. Organ e-ageotypes were associated with specific disease states with known high impact on their respective organs (23 of 117, 20%, associations significant in a meta-analysis after multiple testing correction, Extended Data Fig. 4d and Supplementary Table 9). The kidney ageotype was the most significantly associated with metabolic diseases (diabetes, obesity, hypercholesterolaemia and hypertension), the heart ageotype was the most significantly associated with heart diseases (atrial fibrillation and heart attack), the muscle ageotype was the most significantly associated with gait impairment, the brain ageotype was the most significantly associated with cerebrovascular disease and the organismal ageotype was the most significantly associated with AD. At the whole population level, the relationships between organ age gaps and disease showed the same trends as ageotypes, but more diseases were significantly associated with age gaps due to higher statistical power (65 of 117, 56%, statistically significant after multiple test correction, Extended Data Fig. 4e and Supplementary Table 10).

At the population level, the two most significant associations between disease and age gap were between the kidney age gap and metabolic disease traits. Individuals with hypertension had kidneys that were approximately one year older than their same-aged peers, while individuals with diabetes had kidneys approximately 1.3 years older (Fig. 2a,b and Supplementary Tables 8 and 10). The third and fourth top associations were between the heart age gap and the heart aging traits atrial fibrillation (2.8 years older) and heart attack (2.6 years older) (Fig. 2c,d). Overall, we found that certain diseases, such as heart attack and AD, were associated with accelerated aging in virtually all organs, while others had impacts on a particular organ or subset of organs (Extended Data Fig. 4e and Supplementary Table 10).

Kidney aging proteins were highly expressed by kidney cell types (Fig. 2e,f) and had known roles in kidney biology and disease. Using feature importance plots, the model identified renin (REN), a kidney enzyme known to regulate blood pressure via the renin-angiotensin pathway[22], as an important protein in kidney aging. It also identified the putative longevity factor klotho (KL)[23], as well as multiple proteins with unknown functions including uromodulin (UMOD) and kidney associated antigen 1 (KAAG1), as important kidney aging proteins. UMOD has been genetically linked to chronic kidney disease, where it is observed to have age-dependent effects[24], and rare mutations are the major cause of autosomal dominant tubulointerstitial kidney disease[25].

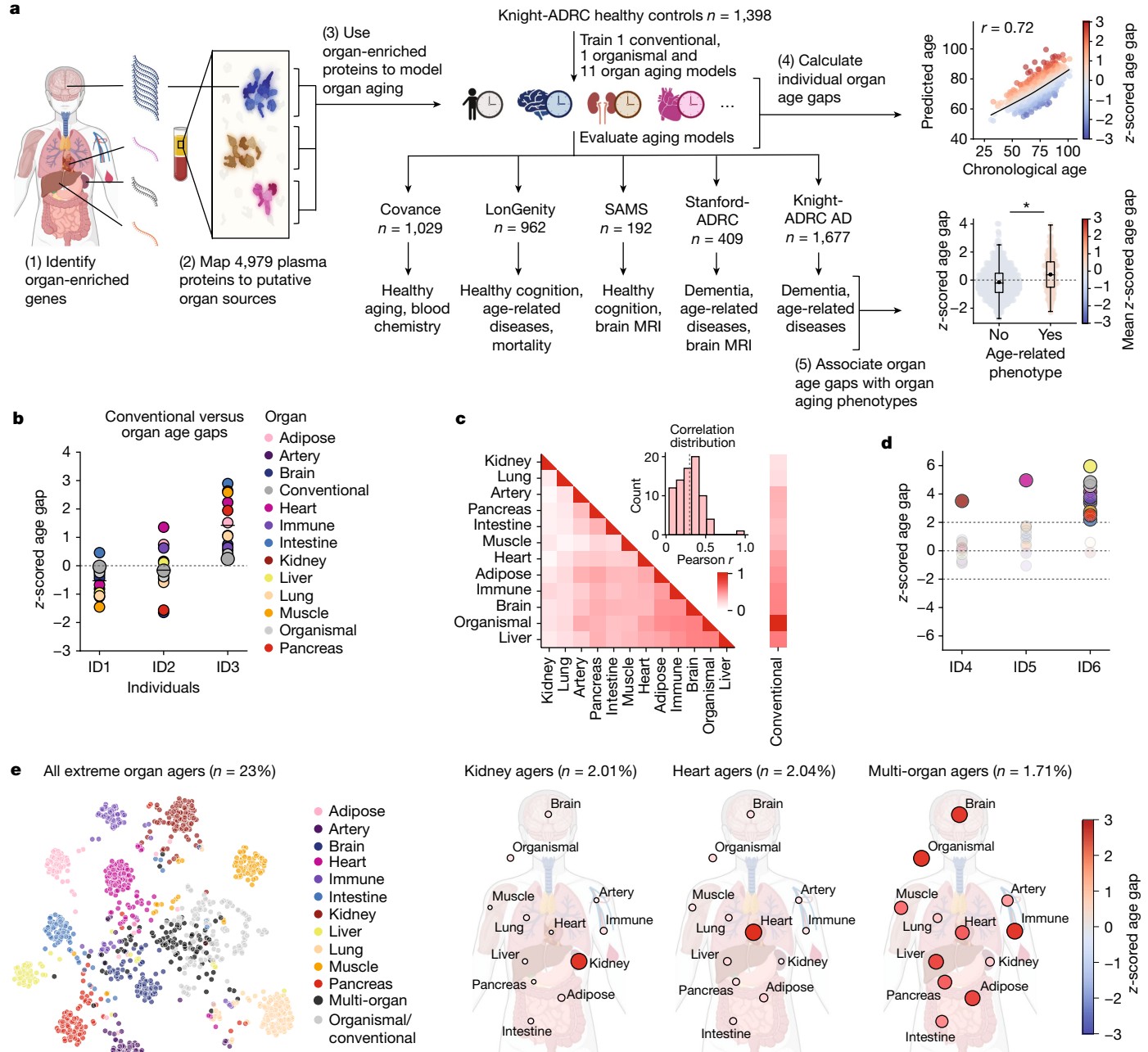

**Fig. 1 | Plasma proteins can model organ aging. a**, Study design to estimate organ-specific biological age. A gene was called organ-specific if its expression was four-fold higher in one organ compared to any other organ in GTEX bulk organ RNA-seq. This annotation was then mapped to the plasma proteome. Mutually exclusive organ-specific protein sets were used to train bagged LASSO chronological age predictors with data from 1,398 healthy individuals in the Knight-ADRC cohort. An 'organismal' model, which used the nonorgan-specific (organ shared) proteins, and a 'conventional' model, which used all proteins regardless of specificity, were also trained. Models were tested in four independent cohorts: Covance (*n* = 1,029), LonGenity (*n* = 962), SAMS (*n* = 192) and Stanford-ADRC (*n* = 420); models were also tested in the AD patients in the Knight-ADRC cohort (*n* = 1,677). To test the validity of organ aging models, the age gap was associated with multiple measures of health and disease. An example age prediction (predicted versus chronological age) and an example

age gap versus phenotype association (age gap versus phenotype, standard boxplot) are shown. **b**, Individuals (ID) with the same conventional age gap can have different organ age gap profiles. Three example participants are shown. Bar represents mean age gap across *n* = 13 age gaps. **c**, Pairwise correlation of organ age gaps from *n* = 3,774 healthy participants across all cohorts. Distribution of all pairwise correlations is shown in inset histogram, with dotted line median correlation. The control age gap was highly correlated with the organismal age gap (*r* = 0.98), the sole outlier in the inset distribution plot. **d**, Identification of extreme agers, defined by a two standard deviation increase or decrease in at least one age gap. A representative kidney ager, heart ager and multi-organ ager are shown. **e**, All extreme agers were identified (23% of all *n* = 5,676 individuals) and clustered after setting age gaps below an absolute *z*-score of 2 to 0. The mean age gaps for all organs in the kidney agers, heart agers and multi-organ agers clusters are shown.

Heart aging proteins were expressed primarily by cardiomyocytes (Fig. 2g,h) and had known roles in heart biology and disease. Pro-brain natriuretic peptide (NPPB), a negative regulator of blood pressure

that increases in response to heart damage, and troponin T (TNNT2), a heart muscle protein involved in contraction, had the strongest weights in the heart aging model (Fig. 2g). They are both established

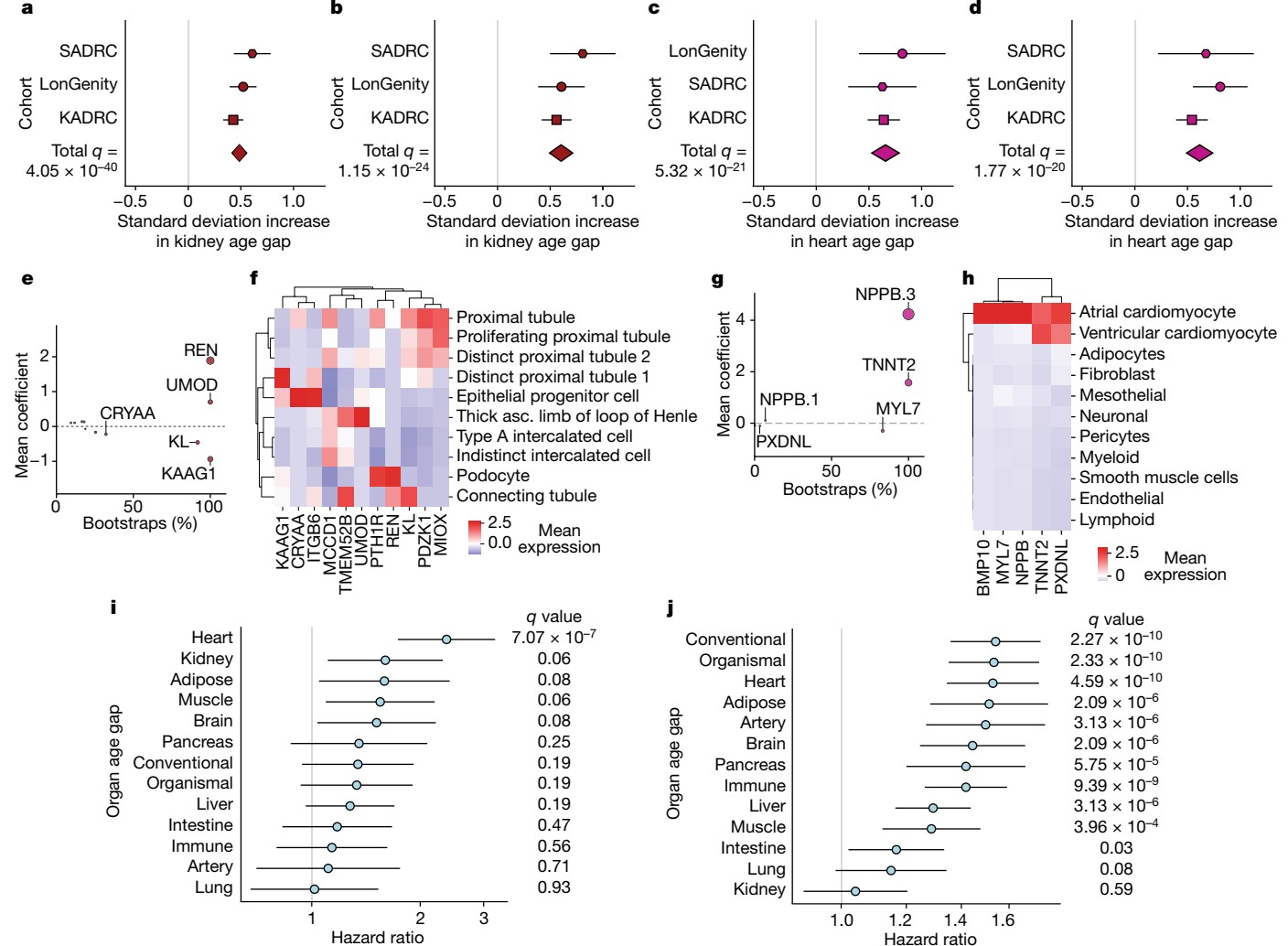

**Fig. 2 | Organ age predicts health and disease. a**, A cross-cohort meta-analysis of the association (linear regression) between the kidney age gap and hypertension (with hypertension $n = 1,566$, without $n = 1,561$). False discovery rate (FDR) $P$ value$_{meta} = 4.05 \times 10^{-40}$, effect size$_{meta} = 0.486$. (Supplementary Table 10). **b**, As in **a**, kidney age gap versus diabetes (with diabetes $n = 335$, without $n = 2,839$). FDR $P$ value$_{meta} = 1.15 \times 10^{-24}$, effect size$_{meta} = 0.604$. **c**, As in **a**, heart age gap versus atrial fibrillation or pacemaker (with atrial fibrillation $n = 239$, without $n = 2,936$). FDR $P$ value$_{meta} = 5.32 \times 10^{-21}$, effect size$_{meta} = 0.657$. **d**, As in **a**, but for heart age gap versus heart attack (with heart attack history $n = 280$, without $n = 2,904$). FDR $P$ value$_{meta} = 1.77 \times 10^{-20}$, effect size$_{meta} = 0.615$. **e**, All kidney aging model coefficients. $x$ axis shows % of model instances in the bagged ensemble that include the protein. Size of bubbles is scaled by the absolute value of the mean model weight across model instances (absolute

value of $y$ axis) (Supplementary Table 7). **f**, Single-cell RNA expression of kidney[51] aging model proteins. Mean normalized expression values shown. **g**, As in **e**, but for the heart aging model. **h**, Human heart single-cell RNA expression of heart[52]. Mean normalized expression values shown. **i**, Cox proportional hazard regression analysis of the relationship between organ age gap and future congestive heart failure risk over 15 years of follow-up in the LonGenity cohort for those without heart failure history at baseline ($n = 26$ events in 812 individuals). FDR $P$ value$_{Heart} = 7.07 \times 10^{-7}$, hazard ratio$_{Heart} = 2.37$. (Supplementary Table 11). **j**, Cox proportional hazard regression analysis of the relationship between organ age gap and future mortality risk, over 15 years of follow-up in the LonGenity cohort ($n = 173$ events in 864 individuals). FDR $P$ value$_{Conventional} = 2.27 \times 10^{-10}$, hazard ratio$_{Conventional} = 1.54$. (Supplementary Table 12). All error bars represent 95% confidence intervals.

clinical markers of acute heart failure[26], and NPPB has been previously associated with heart attack risk[27]. This suggests the possibility of a link between subclinical heart disease and the 'normal' heart aging process, which should be investigated further with more detailed heart imaging and electrophysiology. Less well-characterized heart proteins include cardiac myosin light chain (MYL7), peroxidasin like (PXDNL) and bone morphogenetic protein 10 (BMP10). MYL7 is expressed by atrial cardiomyocytes and has recently become a promising target for hypertrophic cardiomyopathy[28], suggesting that this could be a repurposing target for heart aging more generally.

Given the strong associations between heart aging traits and the heart age gap, we used longitudinal follow-up among healthy participants in the LonGenity cohort to test if organ age was significantly associated with future heart failure risk (Fig. 2i and Supplementary Table 11).

We found that among people with no active disease or clinically abnormal biomarkers at baseline, every 4.1 years of additional heart age (one standard deviation) conferred an almost 2.5-fold increased risk of heart failure over a 15-year follow-up (23% increased risk per year of heart aging, Fig. 2i). Age gaps from multiple other tissues, but not the conventional aging model, also trended towards significance.

We next tested the associations between organ age gaps and all-cause mortality. We found that the age gaps from 10 out of 11 organs, the organismal model and the conventional model were significantly associated with future risk of all-cause mortality after multiple test correction in the LonGenity cohort over 15 years of follow-up (Fig. 2j and Supplementary Table 12). A standard deviation increase (approximately four years of extra organ aging, Supplementary Table 8) in heart, adipose, liver, pancreas, brain, lung, immune or muscle age gap each

conferred between 15–50% increased all-cause mortality risk. These hazard ratios are a similar size to methylation-based mortality predictors in independent aging cohorts over similar follow-up times, despite the fact that organ aging models are trained to predict chronological age instead of mortality directly (DNAm GrimAge hazard ratio = 1.3, 14 year mortality follow-up[29]). Further, we found that for some organs, there was a nonlinear relationship between the age gap and mortality risk (Supplementary Information, Supplementary Fig. 4 and Supplementary Table 13).

Finally, to better understand the relationship between organ age and additional markers of health and disease, we tested the associations between organ age gaps and 43 clinical biochemistry and cell count markers in the test cohort Covance (Extended Data Fig. 5 and Supplementary Fig. 5, see Supplement Information for additional discussion). We also used these markers to calculate Phenotypic age[14] (PhenoAge), a clinical biochemistry-based aging clock which predicts mortality and morbidity risk, for all participants in Covance (Extended Data Fig. 5a). We found that the PhenoAge age gap was significantly correlated with multiple organ age gaps, but only a small portion of the variance in any model was explained by another (Extended Data Fig. 5b).

We found 226 out of 559 (40%) associations between organ age gaps and clinical biochemistry markers were significant after multiple testing correction (Extended Data Fig. 5c and Supplementary Table 14). The strongest associations included associations between liver age gap and blood AST:ALT ratio, a clinical marker of liver health and function that is known to change with age (adjusted Pearson $r = 0.25$, $q = 6.13 \times 10^{-17}$), and between kidney age gap and serum creatinine, the standard clinical marker of kidney function (adjusted Pearson $r = 0.23$, $q = 1.65 \times 10^{-16}$). While these results are highly significant, they only partially explain the relationship between organ age gaps and disease phenotypes. Even after correcting for estimated glomerular filtration rate (eGFR), the kidney age gap is still significantly associated with hypertension and diabetes (Supplementary Fig. 6).

Collectively, organ age gap associations with disease and blood biochemistry demonstrate that aging models derived from organ-specific plasma proteins capture disease-relevant heterogeneity of aging within and across individuals, which is not captured by other aging clocks or clinical markers.

## Brain aging in cognitive decline and AD

Although the largest risk factor for neurodegenerative diseases is age, little is known about the contribution of molecular brain aging to disease. The brain age gap correlated significantly with AD in held-out participants in the Knight-ADRC, but did not replicate in the Stanford Alzheimer's Disease Research Center (Stanford-ADRC) (Supplementary Table 10). Therefore, to better understand how underlying proteins contributed to the brain aging model's predictive abilities for brain aging phenotypes, we developed the feature importance for biological aging (FIBA) algorithm, which uses feature permutation to generate a per-protein importance score for both chronological and biological age, as defined by a particular age-related trait (Extended Data Fig. 6a and Methods). We applied FIBA to the brain age model using the trait global clinical dementia rating (CDRGLOB) in the Knight-ADRC cohort to understand how brain proteins contributed to the association between the age gap and cognitive decline. We observed that some proteins, such as complexins, increased both the model age prediction accuracy and the age gap association with dementia severity (FIBA+), while others decreased the age gap association with dementia severity (FIBA−) (Fig. 3a and Supplementary Table 15).

We used this information to train a second-generation brain aging model, which we term the CognitionBrain aging model, by only using CDRGLOB FIBA+ brain-specific proteins (Fig. 3b and Supplementary

Tables 16–19). This method is similar to second-generation methylation aging clocks which are trained jointly on chronological age and aging phenotypes[14]. We found that the CognitionBrain age gap had a stronger association with AD than the first-generation brain age gap and the conventional age gap in the Knight-ADRC cohort (Extended Data Fig. 6b). This result replicated in the independent test cohort Stanford-ADRC. In a meta-analysis, individuals with AD had approximately two years of additional CognitionBrain aging ($P$ value$_{meta}$ = $9.23 \times 10^{-36}$) compared to individuals without AD (Fig. 3c and Supplementary Table 20). The CognitionBrain age gap was also significantly associated with risk of future dementia progression in both ADRC cohorts. A standard deviation increase in the CognitionBrain age gap conferred a 34% increased risk ($P$ value$_{meta}$ = $1.03 \times 10^{-15}$) of a clinically relevant two-point increase in the Clinical Dementia Rating Sum-of-Boxes score (CDR-SB) within five years (Supplementary Table 21). We also tested associations between CognitionBrain age gap and changes in brain volume using matched volumetric MRI in the Stanford-ADRC and Stanford Aging and Memory Study (SAMS) cohorts (Extended Data Fig. 6c, Supplementary Table 22, Supplementary Fig. 7 and Supplementary Information), and found CognitionBrain age gap significantly predicted brain volume in multiple AD-sensitive regions.

Given its associations with AD status, cognitive decline risk and brain volume, we asked whether the CognitionBrain aging model could be used in combination with other biomarkers of AD and predictors of cognitive decline, including plasma pTau-181 (ref. 5) and an AD polygenic risk score[30], to better stratify AD patients for future clinical outcomes. We tested a multivariate dementia progression cox proportional hazard model with baseline CDRGLOB, age, CognitionBrain age gap, plasma pTau-181 and an AD polygenic risk score (Fig. 3d) in the Stanford-ADRC. We found that the CognitionBrain age gap had the highest adjusted hazard ratio (hazard ratio = 1.57; $P = 8.95 \times 10^{-3}$) of the AD biomarkers, and that both plasma pTau-181 and CognitionBrain age gap were additive for risk prediction (estimated combined hazard ratio = 2.08, Fig. 3e). Individuals with fluid biomarker levels two standard deviations above average had a 75% probability of dementia progression, while individuals with levels two standard deviations below average had under a 10% probability of dementia progression within five years. Pairwise correlation between all biomarkers also showed that the CognitionBrain age gap was largely independent from other biomarkers (Extended Data Fig. 6d). Taken together, these data suggest CognitionBrain age gap provides molecular information about brain aging not captured by other approaches.

Given the significant associations between the CognitionBrain age model and several brain aging metrics, we sought to uncover new insights into brain aging mechanisms by examining the proteins that make up the model. A total of 47 of the 49 model proteins were detectable in human brain single-cell RNA sequencing (scRNA-seq) data and most could be mapped to neurons and glia with high specificity (Fig. 3f). Proteins with the largest positive weights in the model (Fig. 3c) included the synaptic proteins complexin 1 (CPLX1), complexin 2 (CPLX2) and neurexin 3 (NRXN3)—which all have genetic links to cognition and AD[31–33]—and stathmin 2 (STMN2) and olfactomedin 1 (OLFM1)—which are involved in neurite outgrowth and axon growth cone collapse[34,35]. Proteins with large negative weights in the model such as Aldolase Fructose-Bisphosphate C (ALDOC), neuronal pentraxin receptor (NPTXR), carnosine dipeptidase 1 (CNDP1) and Lanc Like Glutathione S-Transferase 1 (LANCL1). ALDOC, NPTXR and CNDP1 are expressed in astrocytes, neurons and oligodendrocytes, respectively (Fig. 3f) and have been proposed as CSF biomarkers for AD[36,37]. LANCL1, which is primarily expressed in oligodendrocytes (Fig. 3f), has been shown to be crucial for neuronal health in mouse models[38]. The model also implicated alterations in the glycosylated extracellular matrix through the proteins tenascin R (TNR), neurocan (NCAN) and heparan sulfate-glucosamine 3-sulfotransferase 4 (HS3ST4), underlining the role of the extracellular matrix in brain aging.

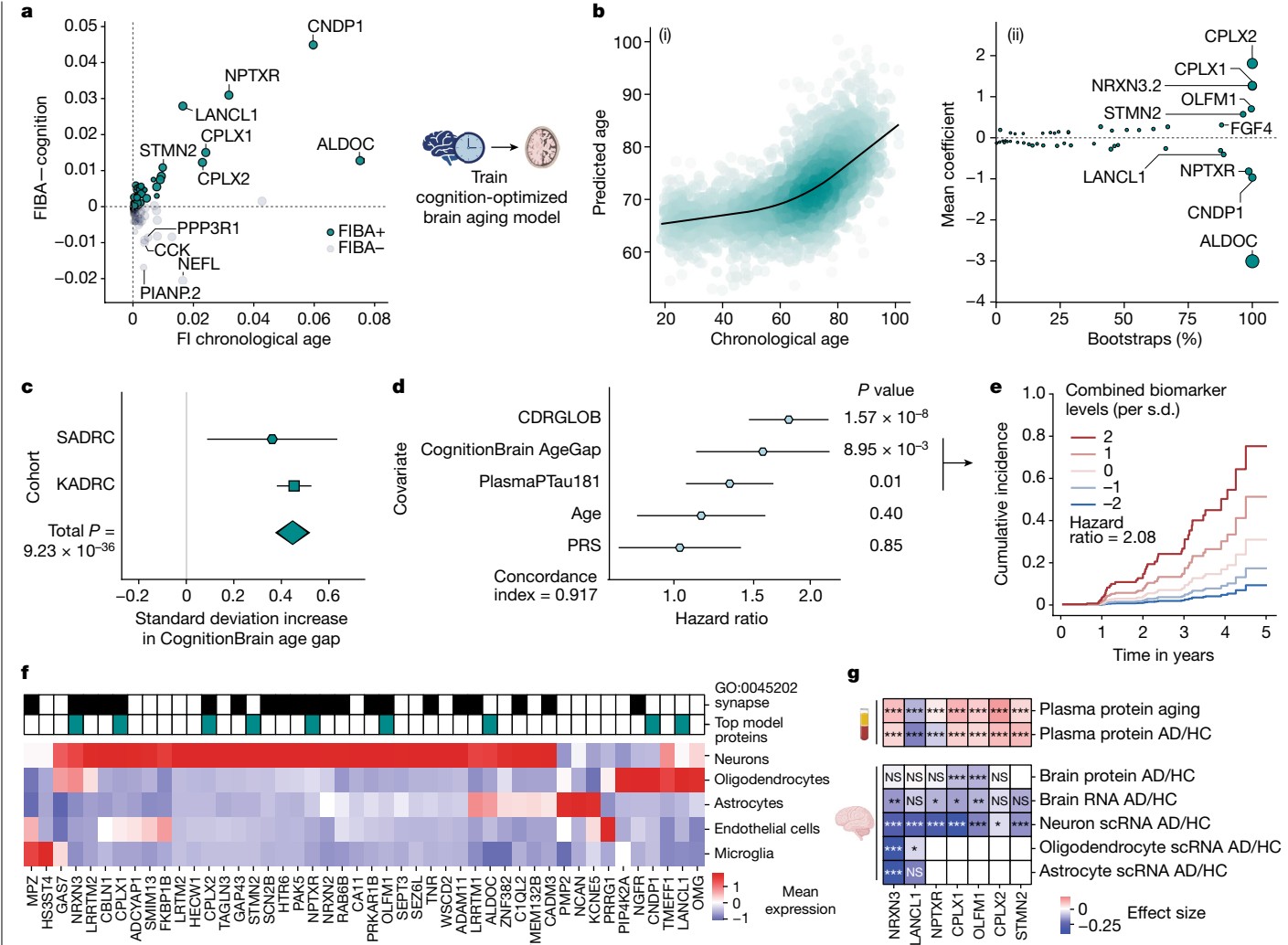

**Fig. 3 | Brain aging in cognitive decline and AD. a**, FIBA was used to test the contributions of brain aging proteins to associations between brain age gap and global clinical dementia rating (CDRGLOB) (*y* axis) or chronological age prediction accuracy (*x* axis). Permutation of some proteins reduced the brain age gap association with CDRGLOB (FIBA+), while permutation of others strengthened it (FIBA−). FIBA+ brain aging proteins were used to train a cognition-optimized brain aging model (CognitionBrain) from cognitively unimpaired individuals in Knight-ADRC. (Supplementary Table 15). FI, feature importance. **b**, CognitionBrain aging model. Age estimation in all cohorts (ii) and bootstrap aging model coefficients (ii). Size of bubbles is scaled by the absolute value of the mean model weight. (Supplementary Table 15). **c**, A cross-cohort meta-analysis of the association (linear regression) between the CognitionBrain age gap and AD diagnosis (with AD *n* = 1,441, without *n* = 2,052). *P* value$_{meta}$ = 9.23 × 10$^{-36}$, effect size$_{meta}$ = 0.448. (Supplementary Table 15). **d**, A multivariate cox proportional hazard model of future dementia

progression risk over five years in Stanford-ADRC (*n* = 48 events in 325 individuals). *P* value$_{CognitionBrain}$ = 8.95 × 10$^{-3}$, hazard ratio$_{CognitionBrain}$ = 1.57. **e**, Kaplan–Meier curve for the CPH model in **f**. Risk of dementia progression for different levels of CognitionBrain AgeGap and PlasmaPTau181 while all other covariates are held constant. Displayed hazard ratio is a first-order estimate of the combined hazard ratio. **f**, Human brain single-cell RNA expression[53] of CognitionBrain aging proteins. Mean normalized expression values shown. Top model proteins and proteins in the GO:CC synapse pathway are highlighted. **g**, Changes with age and AD of top CognitionBrain proteins across tissues (plasma and brain) and molecular layers (protein, bulk RNA and single-cell RNA). Changes in plasma were assessed using linear models from the Stanford- and Knight- ADRC cohorts (*n* = 3,226 individuals). Statistics for brain tissue were pulled from refs. 39,53. Proteins with significant changes across tissues shown. Asterisks represent FDR-adjusted *P* value thresholds: **q* < 0.05; ***q* < 0.01; ****q* < 0.001. All error bars represent 95% confidence intervals. NS, not significant.

We assessed the highest weighted CognitionBrain proteins for their changes with age and AD in the Knight-ADRC and Stanford-ADRC cohorts, as well as their changes with AD in brain tissue at the protein[39], bulk RNA[39] and single-cell RNA levels from publicly available datasets (Fig. 3g). We observed a consistent pattern of decreases in AD brain tissue and increases in the blood with age and AD. This suggests that the increase of synapse and neurite growth related protein levels in the blood could reflect a loss or alteration in protein processing and subsequent shedding of these crucial factors in the brain. A similar inverse relationship between fluid and brain protein levels is seen with amyloid beta, whereby lower CSF AB42 is correlated with increased AB plaques in the brain[40].

## Organ aging in cognitive decline and AD

We next sought to apply the FIBA optimization framework to other organ aging models to understand how the aging of other organs contributes to brain aging phenotypes (Fig. 4a). As with the brain aging model, we applied CDRGLOB FIBA to all aging models using the Knight-ADRC (Extended Data Figs. 7 and 8). The CognitionArtery, CognitionBrain, CognitionOrganismal and CognitionPancreas age gap associations with AD replicated in both ADRCs (Fig. 4b and Extended Data Fig. 8c,d), so we focused on these four aging models to understand peripheral versus central contributions to cognitive decline.

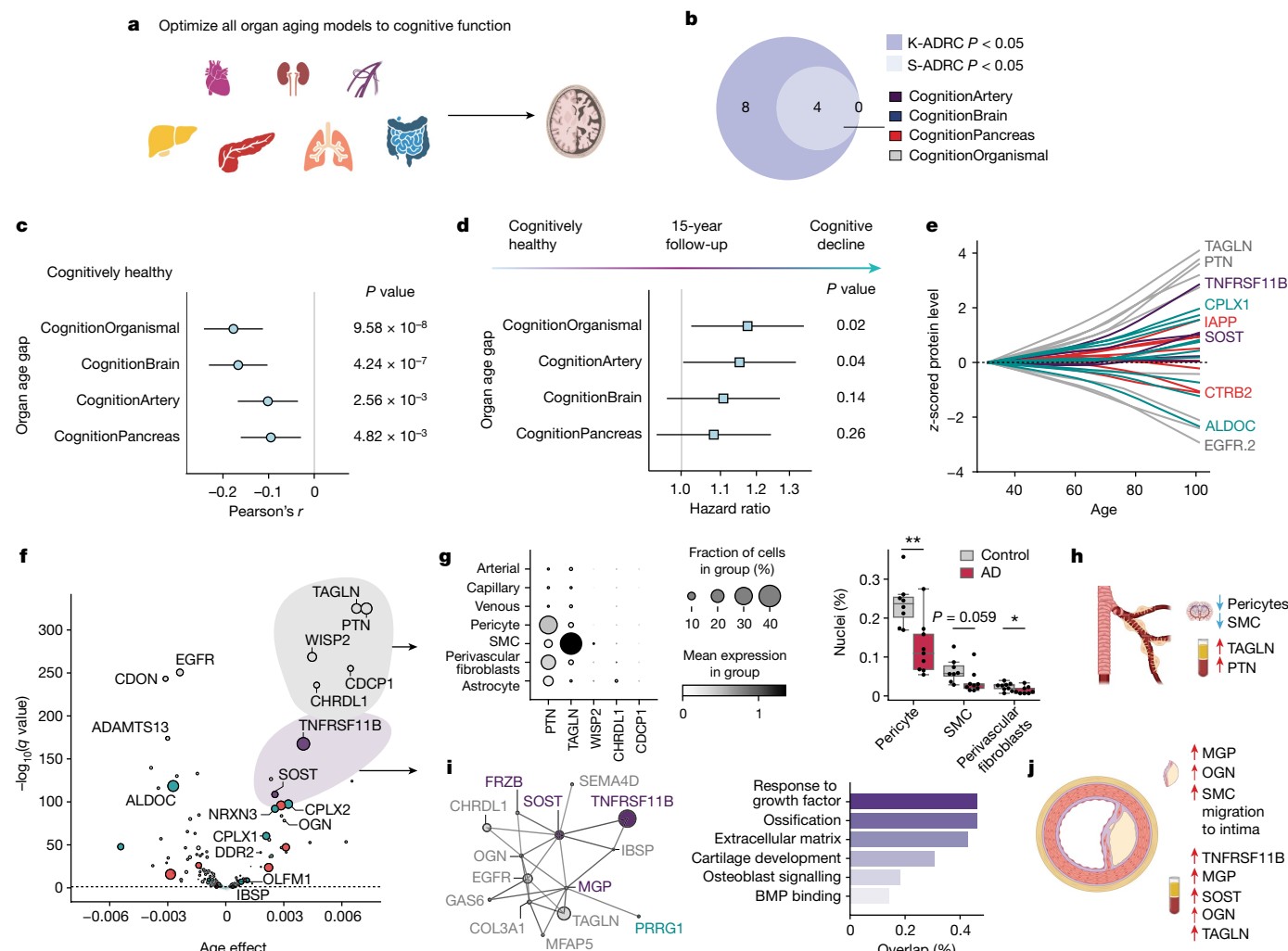

**Fig. 4 | Organ aging in cognitive decline and AD. a**, CDRGLOB FIBA was applied to all organ aging models using the Knight-ADRC (K-ADRC) to understand body-wide contributions to brain aging phenotypes (Supplementary Table 15). **b**, Associations (linear regression) between AD and the CognitionArtery ($P\,value_{meta} = 6.02 \times 10^{-16}$), CognitionBrain ($P\,value_{meta} = 9.23 \times 10^{-36}$), CognitionOrganismal ($P\,value_{meta} = 2.03 \times 10^{-28}$) and CognitionPancreas ($P\,value_{meta} = 1.11 \times 10^{-21}$), age gaps replicated in the Stanford-ADRC (S-ADRC) (Supplementary Table 20). **c**, Associations (linear regression) between organ age gaps and a composite score of overall cognition in the LonGenity cohort ($n = 888$). $P\,value_{CognitionOrganismal} = 9.58 \times 10^{-8}$, $P\,value_{CognitionBrain} = 4.24 \times 10^{-7}$, $P\,value_{CognitionArtery} = 2.46 \times 10^{-3}$ and $P\,value_{CognitionPancreas} = 4.8 \times 10^{-3}$ (Supplementary Table 23). **d**, Cox proportional hazard regression analysis, organ age gap and risk of conversion from cognitively normal to cognitive impairment (CDR-Global $0 \rightarrow \geq 0.5$) over 15 years follow-up in the Knight-ADRC ($n = 226$ events in 940 individuals). $P\,value_{CognitionOrganismal} = 0.02$, $P\,value_{CognitionArtery} = 0.04$, $P\,value_{CognitionBrain} = 0.14$ and $P\,value_{CognitionPancreas} = 0.26$ (Supplementary Table 24). **e**, Aging trajectories of top ten weighted model proteins in healthy individuals

($n = 3,774$) across the four study cohorts. Top CognitionOrganismal proteins change with age earliest and at the highest rate. **f**, Changes with age of top cognition-optimized aging model proteins in healthy individuals ($n = 3,774$) across the four study cohorts. Age effect and negative $\log_{10}$ FDR-corrected $P$ values from a linear model are shown. Size of bubbles is scaled by the absolute value of the average model weight (Supplementary Table 25). **g**, Left, human brain vasculature single-cell RNA expression[42] of top five CognitionOrganismal aging proteins. Mean normalized expression values and fraction of cells expressing the genes are shown. Right, pericytes, smooth muscle cells (SMC) and fibroblasts are lost in AD. Asterisks represent $P$ value thresholds from a two-tailed t-test: *$P < 0.05$; **$P < 0.01$. **h**, Model of age-related cellular degradation of the human brain vasculature reflected in the plasma proteome. **i**, StringDB protein–protein interaction network of CognitionArtery and interacting proteins (score $\geq 0.4$), and related pathway enrichments (percent overlap between proteins and pathway gene sets). **j**, Model of age-related vascular calcification and extracellular matrix alterations reflected in the plasma proteome. All error bars represent 95% confidence intervals.

To understand the full temporal sequence of cognitive decline, we tested if age gaps were associated with cognition in cognitively normal individuals using a composite score of overall cognition in the LonGenity cohort. The decreased cognitive function was significantly associated with all four age gaps (Fig. 4c, Extended Data Fig. 9a and Supplementary Table 23). We replicated these associations in the healthy SAMS cohort, where we observed that individuals with worse memory recall had higher CognitionOrganismal and CognitionBrain age gaps (Extended Data Fig. 9b and Supplementary Table 23).

We next tested associations between age gaps and risk of transition from cognitively normal to mild cognitive impairment (MCI)

(CDR-Global Score 0 to greater than or equal to 0.5) using 15 years of clinical cognitive assessment in the Knight-ADRC (Fig. 4d and Supplementary Table 24). We found that the CognitionOrganismal (hazard ratio = 1.17, $P = 0.02$) and CognitionArtery (hazard ratio = 1.15, $P = 0.04$) age gaps significantly predicted conversion to MCI, while the CognitionBrain (hazard ratio = 1.11, $P = 0.14$) trended towards significance (Fig. 4d). The prediction of future conversion to MCI over 15 years is unlikely to be explained by undiagnosed cognitive impairment, placing changes detected by these aging models early in the causal chain of cognitive decline and neurodegenerative disease.

To understand the biological processes and proteins involved in early cognitive decline, we plotted the aging trajectory of all model proteins and found that highly weighted CognitionOrganismal and CognitionArtery proteins changed with age earlier and at a faster rate than CognitionBrain and CognitionPancreas proteins (Fig. 4e). The earliest changes occurred in a highly correlated cluster of Cognition-Organismal proteins: pleiotrophin (PTN), transgelin (TAGLN), WNT1 Inducible Signalling Pathway Protein 2 (WISP2), CUB Domain Containing Protein 1 (CDCP1) and chordin like 1 (CHRDL1; Fig. 4f). Though not organ-specific, these genes were all highly expressed in the arteries and brain (Extended Data Fig. 10a). Single-cell expression of these genes in human vasculature[41,42], indicated these genes are expressed primarily by smooth muscle cells, pericytes and fibroblasts (Fig. 4g and Extended Data Fig. 10b). Loss of brain pericytes, smooth muscle cells and perivascular fibroblasts is associated with age and AD[42,43] (Fig. 4g), and pericyte-specific deletion of PTN renders neurons prone to ischaemic and excitotoxic injury[44]. This early changing signature in the CognitionOrganismal model may thus represent degenerative changes to the cellular integrity of the brain vasculature and the loss of its neuroprotective functions with aging (Fig. 4h).

The five proteins composing the CognitionArtery model, TNF receptor superfamily member 11b (TNFRSF11B), sclerostin (SOST), melanocortin 2 receptor accessory protein (MRAP2), frizzled related protein (FRZB) and matrix gla protein (MGP) were also primarily expressed in vascular smooth muscle cells, pericytes and fibroblasts[41] (Extended Data Fig. 10c) and are all strongly implicated in vascular calcification. TNFRSF11B/APOE double knockout mice show increased calcium deposition by vascular smooth muscle cells[45], MGP deficiency-causing mutations in humans leads to Keutel syndrome, a disease characterized by soft tissue calcification[46], and SOST and FRZB are negative regulators of WNT signalling that drive calcification and are increased in the plasma of people with vascular calcification[47,48]. We found that CognitionArtery proteins and the vascular signature in the CognitionOrganismal proteins form an interaction network using StringDB (Fig. 4i). Additional model proteins in this interaction network included integrin binding sialoprotein (IBSP), osteoglycin (OGN), collagen type III alpha 1 chain (COL3A1), proline rich and gla domain 1 (PRRG1) and growth arrest specific 6 (GAS6). In total, this protein network is involved in extracellular matrix, cartilage development and osteoblast signalling pathways, and implicates vascular calcification and extracellular matrix alterations as a major component of aging that underlies the early phases of cognitive decline and neurodegenerative disease (Fig. 4i,j).

## Discussion

Our study introduces a framework for modelling organ health and biological aging using plasma proteomics. The resulting organ aging models can predict mortality, organ-specific functional decline, disease risk and progression and aging heterogeneity between tissues. This approach is minimally invasive, requiring only a small blood sample, and could be easily applied to understand the effects of health interventions, such as lifestyle modifications and drug therapies, at the organ level. We provide a large and comprehensive resource of organ aging information in nearly 6,000 individuals spanning the adult lifespan and multiple age-related disease states, and we have developed an easy-to-use python package called organage to calculate the organ ages of any plasma proteomics sample from the SomaScan assay.

There are many future directions for this work. While we have shown that plasma proteomic organ aging models are distinct from previous proteomics models, clinical chemistry-based models and imaging-based models, future studies should assess how proteomic organ aging relates to other molecular measures of aging and disease such as methylation aging clocks and disease-specific prediction models. Although we were unable to perform direct comparisons, our models predict mortality with comparable effect sizes to models trained specifically to predict mortality and heart disease in independent cohorts[49,50]. We demonstrated that our approach added increased value to established biomarkers of AD, and we expect that multimodal aging and disease prediction models may have similar impacts in other diseases.

We present one of the largest studies of plasma proteome aging to date, but as larger plasma proteomics resources emerge, the power of this approach will further increase. Our current models rely on approximately 5,000 proteins measured with the SomaScan assay, but the approach is platform agnostic, and we expect that even more biological information could be gained with additional proteomic coverage, including cell and organ-specific splice isoforms and post-translational modifications. The rapidly growing number of human gene expression maps at single-cell resolution[41] will help further refine organ and cell-type specific aging models and allow for a comprehensive understanding of organismal physiology based on the plasma proteome.

Another question for future studies is which organ-specific aging proteins are causal drivers of aging, given that multiple plasma proteins have been shown to directly modulate aging phenotypes[8]. Of note, many of the proteins with large weights in the models, such as KLOTHO, UMOD, MYL7, CPLX1, CPLX2 and NRXN3, have genetic associations with diseases of their respective organs or are validated therapeutic targets, suggesting a potential causal role of these proteins in organ aging. Future genomic studies should further investigate the genetic architecture of organ aging clocks and their relationships to disease using GWAS and post-GWAS methods such as colocalization and Mendelian randomization.

This study has multiple limitations. First, we have limited the study to a subset of organs to avoid over-interpretation of models for which we lacked convincing organ-relevant aging phenotypes. It remains unclear if this approach will generalize to all organs in the body, such as reproductive organs, and future studies should address this question. Second, we observe many instances of nonlinear dynamics in the plasma proteome and in aging phenotypes. While our current models serve as a proof of principle for this approach, since they are trained and evaluated largely on older adults, caution should be used when applying them to young people. More sophisticated nonlinear machine learning methods such as neural networks or random forests may further improve the accuracy and generalizability of this approach in the future. Lastly, the models were trained and tested on American and Caucasian-skewed cohorts, and future studies should assess the generalizability of the findings in more ethnically and geographically diverse populations.

Altogether, we show that large-scale plasma proteomics and machine learning can be leveraged to noninvasively measure organ health and aging in living people. We show that biologically motivated modelling, in which we use sets of organ-specific proteins and the FIBA algorithm to further subset to physiological age-related proteins, enables deconvolution of the different rates of aging within an individual and measurement of aging at organ-level resolution.

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

# Methods

## Human cohorts

**Covance.** Details of the Covance study have been previously published[54]. Briefly, Covance is a multi-site cross-sectional study of health across the lifespan collected at five hospital sites in the United States in 2008. A total of 1,028 subjects were included in analyses for this study. Cohort demographic characteristics are summarized in Supplementary Table 1. Exclusion criteria for the study included uncontrolled hypertension, self-reported treatment for a malignancy other than squamous cell or basal cell carcinoma of the skin in the last two years, self-reported pregnancy, self-reported chronic infection, autoimmune condition or other inflammatory condition, self-reported chronic kidney or liver disease, chronic heart failure or diagnosed with myocardial infarction in the last three months, self-reported diabetes (HbA1c > 8% if known), self-reported acute bacterial or viral infection in the past 24 h or a temperature greater than 38 °C within 24 h of enrolment, self-reported participation in any therapeutic study within 14 days before blood sampling and taking more than 20 mg of prednisone or related drugs.

Clinical blood chemistry was performed on the same samples, including a complete blood count and comprehensive metabolic panel, lipid panel and liver function tests. Basic physical workup (blood pressure, pulse and respirations) was also collected. Lifestyle information was also collected from all participants using a survey which asked about smoking, alcohol, exercise, habits and frequency of consumption of different meats and vegetables.

**LonGenity.** Details of the LonGenity cohort have been previously published[55,56]. Briefly, LonGenity is an ongoing longitudinal study initiated in 2008 and designed to identify biological factors that contribute to healthy aging. The LonGenity study enrols older adults of Ashkenazi Jewish descent with an age range of 65–94 years at a baseline. Approximately half of the cohort consists of offspring of parents with exceptional longevity, defined as having at least one parent who survived to 95 years of age. The other half of the cohort includes offspring of parents with usual survival, defined as not having a parental history of exceptional longevity. A total of 962 subjects were included in analyses for this study. The cohort characteristics are summarized in Supplementary Table 1. LonGenity participants are thoroughly characterized demographically and phenotypically at annual visits that include collection of medical history and physical and detailed neurocognitive assessments (described in detail below). The LonGenity study was approved by the institutional review board (IRB) at the Albert Einstein College of Medicine.

Subjects in the LonGenity cohort underwent extensive cognitive examination. The Overall Cognition Composite score was determined by the relative performance of the subject in the Free and Cued Selective Reminding Test, WMS-R Logical Memory I, RBANS Figure Copy, RBANS Figure Recall, WAIS-III Digit Span, WAIS-III Digit Symbol Coding, Phonemic Fluency (FAS), Categorical Fluency, Trail Making Test A and Trail Making Test B. For each task a standardized score ($z$) was calculated based on the population. The $z$-score for each task was then combined to create the overall cognition composite.

**Stanford Alzheimer's Disease Research Center.** Samples were acquired through the National Institute on Aging (NIA)-funded Stanford Alzheimer's Disease Research Center (Stanford-ADRC). The Stanford-ADRC cohort is a longitudinal observational study of clinical dementia subjects and age-sex-matched nondemented subjects. The collection of plasma was approved by the Institutional Review Board of Stanford University and written consent was obtained from all subjects. Blood collection and processing were done according to a rigorous standardized protocol to minimize variation associated with blood draw and blood processing. Briefly, about 10 cc of whole blood was collected in a vacutainer ethylenediaminetetraacetic acid (EDTA) tube (Becton Dickinson vacutainer EDTA tube) and spun at 3,000 RPM for 10 mins to separate out plasma, leaving 1 cm of plasma above the buffy coat and taking care not to disturb the buffy coat to circumvent cell contamination. Plasma processing times averaged approximately one hour from the time of the blood draw to the time of freezing and storage. All blood draws were done in the morning to minimize the impact of circadian rhythm on protein concentrations. Plasma pTau-181 levels were measured using the fully automated Lumipulse G 1200 platform (Fujirebio US, Inc, Malvern, PA) by experimenters blind to diagnostic information, as previously described[57].

All healthy control participants were deemed cognitively unimpaired during a clinical consensus conference that included board-certified neurologists and neuropsychologists. Cognitively impaired subjects underwent Clinical Dementia Rating and standardized neurological and neuropsychological assessments to determine cognitive and diagnostic status, including procedures of the National Alzheimer's Coordinating Center (https://naccdata.org/). Cognitive status was determined in a clinical consensus conference that included neurologists and neuropsychologists. All participants were free from acute infectious diseases and in good physical condition. A total of 409 subjects were included in analyses for this study. Cohort demographics and clinical diagnostic categories are summarized in Supplementary Table 1.

**Stanford Aging Memory Study.** SAMS is an ongoing longitudinal study of healthy aging. Blood collection and processing were done by the same team and using the same protocol as in Stanford-ADRC. Neurological and neuropsychological assessments were performed by the same team and using the same protocol as in Stanford-ADRC. All SAMS participants had CDR = 0 and a neuropsychological test score within the normal range; all SAMS participants were deemed cognitively unimpaired during a clinical consensus conference that included neurologists and neuropsychologists. A total of 192 cognitively SAMS participants were included in the present study. The collection of plasma was approved by the Institutional Review Board of Stanford University and written consent was obtained from all subjects. Cohort demographics and clinical diagnostic categories are summarized in Supplementary Table 1.

**Knight Alzheimer's Disease Research Center.** The Knight-ADRC cohort is an NIA-funded longitudinal observational study of clinical dementia subjects and age-matched controls. Research participants at the Knight-ADRC undergo longitudinal cognitive, neuropsychologic, imaging and biomarker assessments including Clinical Dementia Rating (CDR). Among individuals with CSF and plasma data, AD cases corresponded to those with a diagnosis of dementia of the Alzheimer's type (DAT) using criteria equivalent to the National Institute of Neurological and Communication Disorders and Stroke-Alzheimer's Disease and Related Disorders Association for probable AD[58], and AD severity was determined using the Clinical Dementia Rating (CDR)[59] at the time of lumbar puncture (for CSF samples) or blood draw (for plasma samples). Controls received the same assessment as the cases but were nondemented (CDR = 0). Blood samples were collected in EDTA tubes (Becton Dickinson vacutainer purple top) at the visit time, immediately centrifuged at 1,500$g$ for 10 min, aliquoted on two-dimensional barcoded Micronic tubes (200 ul per aliquot) and stored at −80 °C. The plasma was stored in monitored −80 °C freezer until it was pulled and sent to Somalogic for data generation. The Institutional Review Board of Washington University School of Medicine in St. Louis approved the study and research was performed in accordance with the approved protocols. A total of 3,075 participants were included in the present study. Cohort demographics and clinical diagnostic categories are summarized in Supplementary Table 1.

## Proteomics data acquisition and quality control

**SomaScan assay.** We used the SomaLogic SomaScan assay, which uses slow off-rate modified DNA aptamers (SOMAmers) to bind target

proteins with high specificity, to quantify the relative concentration of thousands of human proteins in plasma. The assay has been used in hundreds of studies and described in detail previously[54,60]. Two versions of the SomaScan assay were used in this study. The v.4 assay (4,979 protein targets) was applied to the Covance and LonGenity cohorts, and the v.4.1 assay (7,288 protein targets) was applied to the SAMS, Stanford-ADRC and Knight-ADRC cohorts. All v.4 targets are included in the v.4.1 assay based on SeqId, and only the v.4 targets were analysed for this study.

**Somalogic normalization and quality control.** Standard Somalogic normalization, calibration and quality control were performed on all samples[54,61–63]. Briefly, pooled reference standards and buffer standards are included on each plate to control for batch effects during assay quantification. Samples are normalized within and across plates using median signal intensities in reference standards to control for both within-plate and across-plate technical variation. Samples are further normalized to a pooled reference using an adaptive maximum likelihood procedure. Samples are additionally flagged by SomaLogic if signal intensities deviated significantly from the expected range and these samples were excluded from analysis. The resulting expression values are the provided data from Somalogic and are considered 'raw' data.

The v.4 → v.4.1 multiplication scaling factors provided by Somalogic were applied to the raw v.4 assay expression values to allow for direct comparisons across two v.4 and three v.4.1 cohorts. We discarded proteins for which the correlation was low between assay versions v.4 and v.4.1 and low estimated replicate coefficient of variation[64] (Supplementary Fig. 1). This resulted in 4,778 proteins for downstream analysis. The raw data were $\log_{10}$ transformed before analysis, as the assay has an expected log-normal distribution.

**Somalogic probe validation.** Somalogic has analysed close to 1 million samples with their technology at the time of this publication, resulting in some 700 publications (https://somalogic.com/publications/). There is minimal replicate sample variability[64,65] (coefficient of variation, CV). The majority of SomaScan protein measurements are stable and a subset of proteins have been validated as laboratory-developed tests (LDTs), and have been delivered out of Somalogic's CLIA-certified laboratory to physicians and patients in the context of medical management[66].

1. All 7,524 probes on the assay undergo rigorous primary validation of binding and sensitivity to the target protein.
a. Determination of equilibrium binding affinity dissociation constant ($K_D$).
b. Pull down assay of cognate protein from buffer.
c. Demonstration of dose-responsive in the SomaScan Assay.
d. Estimation of endogenous cognate protein signals in human plasma above limit of detection.
2. A total of 70% of their probes have at least one orthogonal source of validation (Supplementary Fig. 1b) from:
a. Mass spectrometry: approximately 900 probes which measure mostly high and mid abundance proteins (due to sensitivity limitations of mass spectrometry), have been confirmed with either data dependent acquisition (DDA) or multiple reaction monitoring (MRM) mass spectrometry.
b. Antibody: approximately 390 probe measurements correlate with antibody based measurements.
c. Cis-protein quantitative trait loci (pQTL): approximately 2,860 probe measurements are associated with genetic variation in the cognate protein-encoding gene.
d. Absence of binding with nearest neighbour: approximately 1,150 probes do not detect signal from the protein that is most closely related in sequence to the cognate protein.
e. Correlation with RNA: approximately 1,460 probe measurements correlate with mRNA levels in cell lines.

## Identification of organ-enriched plasma proteins

We used the Gene Tissue Expression Atlas (GTEx) human tissue bulk RNA-seq database[18] to identify organ-enriched genes and plasma proteins (Extended Data Fig. 1). Tissue gene expression data were normalized using the DESeq2 (ref. 67) R package. We define organ-enriched genes in accordance with the definition proposed by the Human Protein Atlas[19]: a gene is enriched if it is expressed at least four times higher in a single organ compared to any other organ. Within GTEx, we grouped tissues of the same organ together, such that a gene's expression level for a given organ was the maximum gene expression value among its subtissues. For example, all GTEx brain regions were considered subtissues of the brain organ. We define the immune organ, which is not a GTEx tissue, as expression in the blood and the spleen tissues. Organ-enriched genes were mapped to the 4,979 plasma proteins quantified in the v.4 SomaScan assay.

## Bootstrap aggregated LASSO aging models

To estimate biological age using the plasma proteome, we built LASSO regression-based chronological age predictors (Extended Data Figs. 2–3 and Supplementary Fig. 3) using the scikit-learn[68] python package. We employed bootstrap aggregation for model training. Briefly, we resampled with replacement to generate 500 bootstrap samples of our training data (Knight-ADRC: 1,398 healthy individuals). Each bootstrap sample was the same size as the training data, 1,398. For each bootstrap sample, we trained a model on $z$-scored $\log_{10}$ normalized protein expression values with sex ($F = 1$, $M = 0$) as a covariate to predict chronological age. For model training, we performed hyperparameter tuning of the L1 regularization parameter, $\lambda$, with five-fold cross validation using the GridSearchCV function from scikit-learn. To reduce model complexity and avoid overfitting, we selected the highest $\lambda$ value that retained 95% performance relative to the best model. The mean predicted age from all 500 bootstrap models was used.

We trained our models in 1,398 cognitively unimpaired participants from the Knight-ADRC cohort. We evaluated their performance in the Covance ($n = 1,029$), LonGenity ($n = 962$), SAMS ($n = 192$), Stanford-ADRC ($n = 409$) cohorts and Knight-ADRC cognitively impaired subjects ($n = 1,677$). Models that included sex as a covariate and models trained separately on males and females showed similar age prediction performance on both sexes, so we controlled for sex to extend the generality of the findings and reduce analytic complexity (Supplementary Fig. 3a–c). There was a correlation between age estimation accuracy and the number of proteins used as input to each model (Supplementary Fig. 3c,d). However, several models with few protein inputs, such as the adipose (five proteins) and heart models (ten proteins), predicted chronological age better than models with more protein inputs (Extended Data Fig. 3).

## Age gap calculation and independent validation

To calculate each individual sample age gap for each aging model, we performed the following steps for each aging model. We fit a local regression between predicted and chronological age using the lowess function from the statsmodels[69] python package with fraction parameter set to 2/3 to estimate the true population mean (Supplementary Fig. 3e). A local regression is used in place of a simple linear regression because of extensive evidence that the plasma proteome changes nonlinearly with age[1], which we see replicated in all five cohorts (Supplementary Fig. 8). Individual sample age gaps were then calculated as the difference between predicted age and the lowess regression estimate of the population mean. Age gaps were calculated separately per cohort to account for cohort differences (Supplementary Fig. 3e). Age gaps were $z$-scored per aging model to account for the differences in model variability (Supplementary Fig. 3f). This allowed for direct comparison between organ age gaps in downstream analyses.

## Phenotypic age calculation

We used the published coefficients[14] to calculate the phenotypic age of participants in the Covance cohort using albumin, creatinine, glucose, c-reactive protein, % lymphocyte, mean cell volume, red cell distribution width, alkaline phosphatase, white blood cell count and age.

## Statistical methods to associate organ age gaps with age-related phenotypes

**Study design.** A flowchart of the study design is provided in Supplementary Fig. 2. Each box in the flowchart was treated as a separate analysis for the purpose of multiple testing correction. Multiple testing correction was done using the Benjamani–Hochberg method and the significance threshold was a 5% false discovery rate. To summarize the flowchart, the age gaps from all 11 organ aging models, the organismal model and the conventional model were used in the following analyses: prediction of future mortality in the LonGenity cohort with a cox proportional hazards model (CPH) (12 of 13 tests significant after FDR), prediction of future heart disease in the LonGenity cohort with a CPH (12 of 13 tests significant after FDR), association with nine diseases of aging in a cross-cohort meta-analysis (66 of 17 tests significant after FDR) and association with 42 clinical biochemistry markers in the Covance cohort (237 of 588 tests significant after FDR, PhenoAge gap also tested for 14 × 42 tests).

The 12 cognition-optimized models (11 organs + organismal model) were tested on additional brain aging phenotypes. The CognitionBrain age gap only was tested for association with 65 MRI brain volumes and an MRI-based brain age gap (40 of 66 tests significant after FDR). The CognitionBrain age gap only was included in a multivariate CPH model of dementia progression in AD (1 of 1 tests significant, no FDR). The 12 cognition-optimized model age gaps were tested for association with AD status in the Knight-ADRC (12 of 12 tests significant after FDR), then a replication analysis was performed in Stanford-ADRC (4 of 12 tests significant at $P < 0.05$, no FDR). The four models which replicated CognitionBrain, CognitionOrganismal, CognitionArtery and CognitionPancreas were then tested for associations with overall cognition in healthy elderly people (LonGenity, 4 of 4 tests significant and no FDR), memory function in the Stanford-ADRC (2 of 4 tests significant, no FDR) and 15-year prediction of conversion from normal cognition to mild cognitive impairment in the Knight-ADRC with a CPH model (2 of 4 tests significant, no FDR).

**Linear modelling.** Estimation of chronological age is not sufficient in determining whether an organ aging model measures the age-related physiological dysfunction of an organ. To determine whether estimated organ age contains physiologically relevant information, we associated organ age gaps with various age-related phenotypes across Covance, LonGenity, SAMS, Stanford-ADRC and Knight-ADRC cohorts. Most organ age gap versus trait associations in this study (Figs. 2a–d and 3c and Extended Data Figs. 4d,e, 5c, 6b,c, 7, 8c,d and 9) were assessed using linear models controlled for age and sex as follows: age gap ≈ trait + age + sex and adjusted for multiple testing burden using the Benjamini–Hochberg method when appropriate. To describe disease associations in relation to years of additional aging in the main text, we took the coefficient for the trait variable—which provides an estimate of the mean difference in $z$-scored age gaps between disease and control—and converted that to an estimate of mean difference in raw age gaps, using the standard deviation of raw age gaps provided in Supplementary Table 8.

**Meta-analyses.** Meta-analyses to compare and aggregate effect sizes and confidence intervals from multiple cohorts were performed in R using the metafor[70] package with an inverse variance weighted fixed effects model.

**Cox proportional hazard modelling.** Cox proportional hazards models were used to assess the association between organ age gaps and future risk of mortality, congestive heart failure and increase in clinical dementia rating using the following model: event risk ≈ organ age gap + age + sex. Models were tested using the lifelines[71] python package. Kaplan Meyer curves were generated at population-average covariate values in the relevant subject populations.

## Extreme agers

Extreme agers were defined as individuals who had an age gap value two standard deviations above or below the mean ($z$-scored age gap greater than 2 or $z$-scored age gap less than −2) for at least one aging model. A total of 23% of the population across all cohorts were extreme agers. All extreme agers showed accelerated aging; no individuals displayed extreme youth signatures without extreme aging signature in a different organ (Extended Data Fig. 4a). To identify different groups of extreme agers with similar aging profiles, we performed $k$-means clustering ($n = 13$) of the extreme agers. $Z$-scored age gap values above 2 or below −2 were set to zero before clustering. The clusters showed distinct organ agers (Fig. 1e and Extended Data Fig. 4b). A multi-organ ager cluster was also identified. Individuals who were extreme agers in at least five different organs were manually set to multi-organ agers. Extreme ageotypes (clusters) were associated with major age-related diseases using logistic regression (trait ≈ e-ageotype) in a cross-cohort meta-analysis (Extended Data Fig. 4d and Supplementary Table 9)

## Feature importance for biological aging

FIBA is an adaptation of permutation feature importance (PFI)[72] (Extended Data Fig. 6a). PFI is traditionally used in machine learning to assess how much a model depends on a given feature for prediction accuracy of the target variable. The PFI score is defined as the decrease in a model's performance when values from a single feature are randomized. In our case, for chronological age predictors, the PFI score would be calculated as the difference between the model's original prediction accuracy (Pearson correlation between predicted and chronological age) and the model's prediction accuracy after randomization of a single feature. The final PFI score is the mean PFI score from five randomizations.

FIBA builds on the concept of PFI and applies it to the field of aging to assess the importance of a feature in measuring biological age, instead of the target variable, chronological age. We assume that information about biological age lies in the model age gap and its association with an age-related trait. Thus, randomization of an important feature would reduce the association between the model age gap and the trait (in the expected direction). The FIBA score for a protein is calculated based on this logic and is defined as the difference between the model age gap's original association with a trait and the association with that trait after randomization of a single feature.

We applied FIBA to understand aging model protein contributions to associations with cognition using the CDR-Global score. The mean FIBA score after five permutations was calculated for all 500 bootstraps for all organ aging models (Supplementary Table 15). A protein was defined as significant (FIBA+) if less than 5% (empirical single-tailed $P < 0.05$) of its FIBA scores across bootstraps was negative. Only proteins with nonzero coefficients in at least 100/500 bootstraps were considered. FIBA+ organ-specific proteins were used to train new cognition-optimized aging models from cognitively unimpaired individuals in the Knight-ADRC cohort.

## Biological pathway enrichment and protein–protein interaction analysis

Biological pathway enrichment analyses were performed using g:Profiler[73] with the all human genes set as the background distribution. Protein–protein interaction networks were generated using the STRING database[74].

### Single-cell RNA sequencing analysis

Preprocessed human heart[52] and kidney[51] scRNA-seq data were accessed from studies in the Human Cell Atlas. Preprocessed brain scRNA-seq data were accessed from ref. 53. Preprocessed human brain vasculature scRNA-seq data were accessed from ref. 42. Preprocessed human vasculature scRNA-seq data were accessed from Tabula Sapiens[41]. Gene expression counts data were log(CPM + 1) transformed and z-scored for visualization.

### Brain tissue bulk proteomics and RNA sequencing

Differential expression statistics of proteins and RNA from AD versus control brains were accessed from ref. 39.

### Brain MRI data from Stanford-ADRC and SAMS cohorts

**MRI acquisition.** Whole-brain MRI scans were collected from all subjects in the Stanford-ADRC and SAMS cohorts. All MRI data was collected at the Stanford Richard M. Lucas Center for Imaging. A total of 271 subjects underwent MRI scanning on a 3 T MRI scanner (GE Discovery MR750). T1-weighted SPGR scans were collected (TR/TE/TI = 8.2/3.2/900 ms, flip angle = 9, 1 × 1 × 1 mm) and used to define grey matter volumes. A total of 134 subjects underwent MRI scanning on a hybrid PET/MRI scanner (Signa 3 tesla, GE Healthcare). T1-weighted SPGR scan were collected (TR/TE/TI = 7.7/3.1/400 ms, flip angle = 11, 1.2 × 1.1 × 1.1 mm) and used to define grey matter volumes.

**Structural MRI processing.** Region of interest (ROI) labelling was implemented using the FreeSurfer[75] software package v.7 (http://surfer.nmr.mgh.harvard.edu). In brief, structural images were bias field corrected, intensity normalized and skull stripped using a watershed algorithm. These images underwent a white matter-based segmentation, grey/white matter and pial surfaces were defined, and topology correction was applied to these reconstructed surfaces. Subcortical and cortical ROIs spanning the entire brain were defined in each subject's native space, using the aparc+aseg atlas in FreeSurfer.

**MRI brainageR algorithm.** Using matched brain MRI and plasma proteomic data from n = 541 samples in SAMS and Stanford-ADRC, we compared our plasma proteomic organ clocks with established brain MRI-based clocks, brainageR[16] and BARACUS Brain-Age[76].

We used a pretrained machine learning algorithm (https://github.com/james-cole/brainageR) and raw T1-weighted MRI scans to estimate brain age. This software uses SPM12 (https://www.fil.ion.ucl.ac.uk/spm/software/spm12/) to perform tissue segmentation and normalization of individual scans to Montreal Neurological Institute (MNI) template space. The software relies on a model that used Gaussian process regression to predict brain age on 3,777 participants from seven publicly available datasets (mean age = 40.1, range = 18–90 years). It applies the results of this training to predict brain age in any new T1-w data, utilizing the RNifti (v.1.4.5) and kernlab (v.0.9-32) packages within R v.4.2.

We also used another pretrained algorithm, BARACUS (https://github.com/bids-apps/baracus, ref. 76) to estimate brain age from FreeSurfer v.5.3 processed T1-w scans. The vertex-wise cortical thickness and surface area values (transformed from subject space to fsaverage4 standard space), along with the subcortical volumetric statistics, were used as input to BARACUS's linear support vector machine model. This model was trained on 1,166 participants with no objective cognitive impairment (566 female, mean age = 59.1, range = 20–80 years). It returns a 'stacked-anatomy' prediction among its results, which we used as the estimate of brain age for this method.

**MRI regions of interest analysis.** The volume of the AD signature region was calculated as the sum of the volumes of the parahippocampal gyrus, entorhinal cortex, inferior parietal lobules, hippocampus and precuneus. Following best practice, ROIs were linearly adjusted for estimated total intracranial volume to account for the differences in human size that is unrelated to cognitive function and neurodegeneration. Associations between organ age gaps and adjusted brain ROIs were tested using a linear model controlled for age and sex. Associations were performed for all ROIs in the aparc+aseg atlas.

### Alzheimer's disease polygenic risk score in the Stanford-ADRC cohort

AD polygenic risk scores (PRS) were calculated in the Stanford-ADRC cohort to compare to the CognitionBrain age gap. PRSs were determined from whole-genome sequencing. The Genome Analysis Toolkit workflow Germline short variant discovery was used to map genome sequencing data to the reference genome (GRCh38) and to produce high-confidence variant calls using joint-calling[77]. Six individuals were excluded from further whole-genome sequencing analysis due to discordance between their reported sex and genetic sex. APOE genotype (ε2/ε3/ε4) was determined using allelic combinations of single nucleotide variants rs7412 and rs429358. The independent loci identified in the largest AD GWAS to date were used to compute AD PRS. Namely, the 84 variants and their effect size available from Tables 1 and 2 in ref. 30 were used, in addition to rs7412 (odds ratio = 0.6) and rs429358 (odds ratio = 3.7). Plink1.9 (ref. 78) with the '−score' flag was used to formally compute the PRS, while providing the individual genotypes and the list of variants with their effect size as input. Three individuals with pathogenic mutations PSEN1 or GBA were removed from this analysis.

### Reporting summary

Further information on research design is available in the Nature Portfolio Reporting Summary linked to this article.

## Data availability

Stanford-ADRC data are available upon reasonable request to the Stanford-ADRC data release committee, https://web.stanford.edu/group/adrc/cgi-bin/web-proj/datareq.php. All Stanford-ADRC data will be made publicly available after an embargo period at https://twc-stanford.shinyapps.io/adrc/. SAMS data are available to qualified investigators upon request to principal investigators Beth Mormino (bmormino@stanford.edu) or Anthony Wagner (awagner@stanford.edu). Knight-ADRC data were generated by the laboratory of principal investigator Carlos Cruchaga (cruchagac@wustl.edu) and are available upon reasonable request to the The National Institute on Aging Genetics of Alzheimer's Disease Data Storage Site (NIAGADS) (Study ID: ng00130), https://www.niagads.org/knight-adrc-collection. Data from the Covance and LonGenity cohorts can be accessed according to the policies described in the initial study publications[54–56]. Preprocessed human heart[52] and kidney[51] scRNA-seq data were accessed from studies in the Human Cell Atlas. Preprocessed brain scRNA-seq data were accessed from ref. 53. Preprocessed human brain vasculature scRNA-seq data were accessed from Yang et. al. 2022 (ref. 42). Preprocessed human vasculature scRNA-seq data were accessed from Tabula Sapiens[41]. Differential expression statistics of proteins and RNA from Alzheimer's disease versus control brains were accessed from ref. 39. Change with age information of approximately 5,000 SomaScan v.4 plasma proteins across all five cohorts (Supplementary Fig. 8 and Supplementary Table 25) are available in a public shiny app (https://twc-stanford.shinyapps.io/aging_plasma_proteome_v2/).

## Code availability

All analyses have been carried out using freely available software packages in python and R. All aging models are available and easily accessible using the organage package in Python and the associated github repository (https://github.com/hamiltonoh/organage). The package requires v.4 or higher SomaScan data, age and sex as inputs, and outputs

estimated organ ages and age gaps. The aging models are available to download from the package, and the model coefficients are available in Supplementary Tables 6 and 17. Code for the FIBA algorithm are in the package's GitHub repository.

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

**Acknowledgements** We thank A. Keller, D. Gate, O. Leventhal, R. Vest, T. Iram, S. R. Shuken, A. Kaur, S. Shi, E. Costa, A. Shankar, A. Morningstar and other members of the Wyss-Coray laboratory for feedback and support, and D. Berdnick, H. Zhang and K. Dickey for laboratory management. This work was supported by the Stanford Alzheimer's Disease Research Center (National Institute on Aging grants P50AG047366 and P30AG066515), the National Institute on Aging (AG072255,T.W.-C; AG057909, AG061155 and AG044829, S.M. and N.B; AG066206, Z.H.), the National Institutes of Health (R01AG044546, RF1AG053303, RF1AG058501 and U01AG058922, C.C.; P01AG003991, C.C. and J.C.M.; RF1AG074007, Y.J.S.), the Michael J. Fox Foundation (L.I. and C.C.), the Alzheimer's Association Zenith Fellows Award (ZEN-22-848604, C.C.), the Milky Way Research Foundation, Nan Fung Life Sciences (T.W.-C.), the Stanford Graduate Fellowship (H.O. and J.R.), the Stanford Translational Program in Aging Research (T32AG047126, D.N.) and the National Science Foundation Graduate Research Fellowship (H.O.).

**Author contributions** T.W.-C., B.L., H.O. and J.R. conceptualized the study. J.R. and H.O. led and performed all analyses. J.R., H.O. and P.M.-L. assessed quality control and normalization methods for SomaScan plasma proteomics data. H.O. and J.R. developed the FIBA algorithm. R.P. and D.N. advised on machine learning best practices. D.N., Z.H. and S.B.M. advised on statistical methods. O.A. aided in brain MRI data analyses from the SAMS and Stanford-ADRC cohorts. D.Y.U. and T.M.M. aided in analyses. K.K. and P.M.-L. created the shiny app. D.C. led plasma collection for the Stanford-ADRC cohort. Y.J.S., L.W., J.T., D.W., M.L., P.K., J.B. and C.C. generated proteomics from the Knight-ADRC cohort. E.N.W. and K.I.A. led plasma tau data collection in the Stanford-ADRC cohort. Y.G. and M.D.G. generated Alzheimer's polygenic risk scores in the Stanford-ADRC cohort. R.P., M.H. and A.C.Y. aided in single-cell RNA-seq analyses. S.S. collected proteomics and E.F.W. led cognition tests for the LonGenity cohort. S.M. and N.B. established the LonGenity project and provided data. A.D.W. and E.M. established the SAMS cohort and provided data and insights. V.W.H. assisted in Stanford-ADRC data acquisition. V.W.H., F.M.L. and T.W.-C. lead the Stanford-ADRC cohort. H.O. assembled the figures. J.R. and H.O. wrote the manuscript. J.R. edited the manuscript. T.W.-C. supervised the study. All authors critically revised the manuscript for intellectual content. All authors read and approved the final version of the manuscript.

**Competing interests** T.W.-C., H.O., J.R., B.L. and Stanford University have filed a patent application related to this work, PCT/US2023/027896. T.W.-C., H.O. and J.R. are co-founders and scientific advisors of Teal Omics Inc. and have received equity stakes. T.W.-C. is a co-founder and scientific advisor of Alkahest Inc. and Qinotto Inc. and has received equity stakes in these companies. C.C. has received research support from GSK and EISAI. The funders of the study had no role in the collection, analysis or interpretation of data; in the writing of the report; nor in the decision to submit the paper for publication. C.C. is a member of the advisory board of Vivid Genomics and Circular Genomics and owns stocks in these companies. S.B.M is a consultant for BioMarin, MyOme and Tenaya Therapeutics. All other authors have certified they have no competing interests to declare.

**Additional information**
**Correspondence and requests for materials** should be addressed to Tony Wyss-Coray.

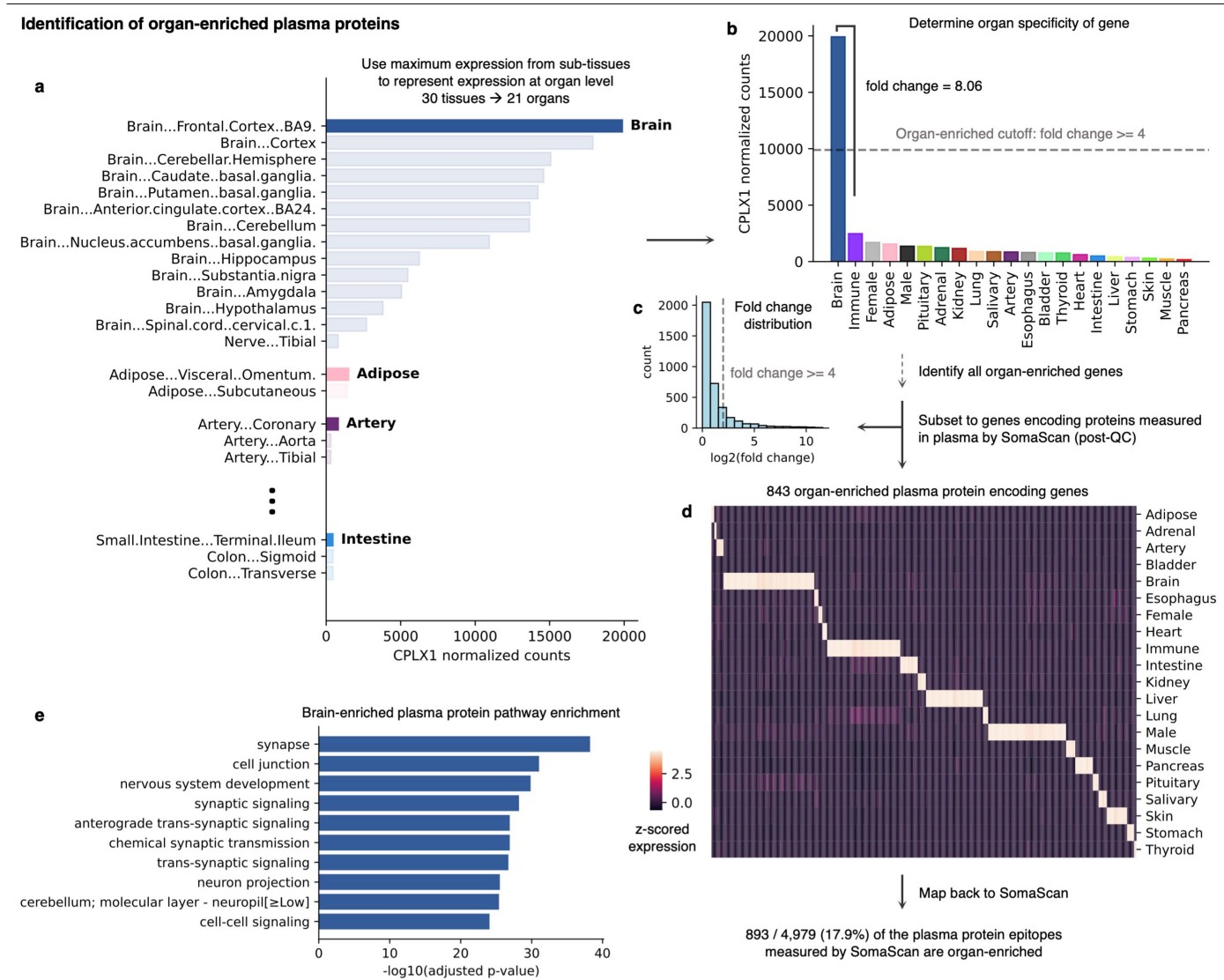

**Extended Data Fig. 1 | Identification of organ-enriched plasma proteins.** **a**, Plasma proteins for which the gene encoding the protein was expressed at least four-fold higher in one organ compared to any other organ were called "organ-enriched" in line with the definition proposed by the Human Protein Atlas. To calculate organ-level gene expression, the maximum expression of sub-tissues in the Gene Tissue Expression Atlas (GTEx) bulk RNA-seq database was used. An example of this tissue expression aggregation into organ expression CPLX1. (See ST2). **b**, Organ-wide expression for CPLX1. CPLX1 is expressed over four-fold higher in the brain compared to any other organ and is therefore defined as organ-enriched. **c**, Organ-level fold-change distribution of SomaScan plasma protein encoding genes. (See ST3). **d**, Organ-level expression of 843 organ-enriched plasma protein encoding genes. These 843 genes correspond to 893 plasma protein epitopes measured on the SomaScan assay. Some plasma proteins on the assay are quantified multiple times by different aptamers, which target different epitopes of the same protein. **e**, Top significantly enrichment biological pathways of brain-enriched plasma proteins.

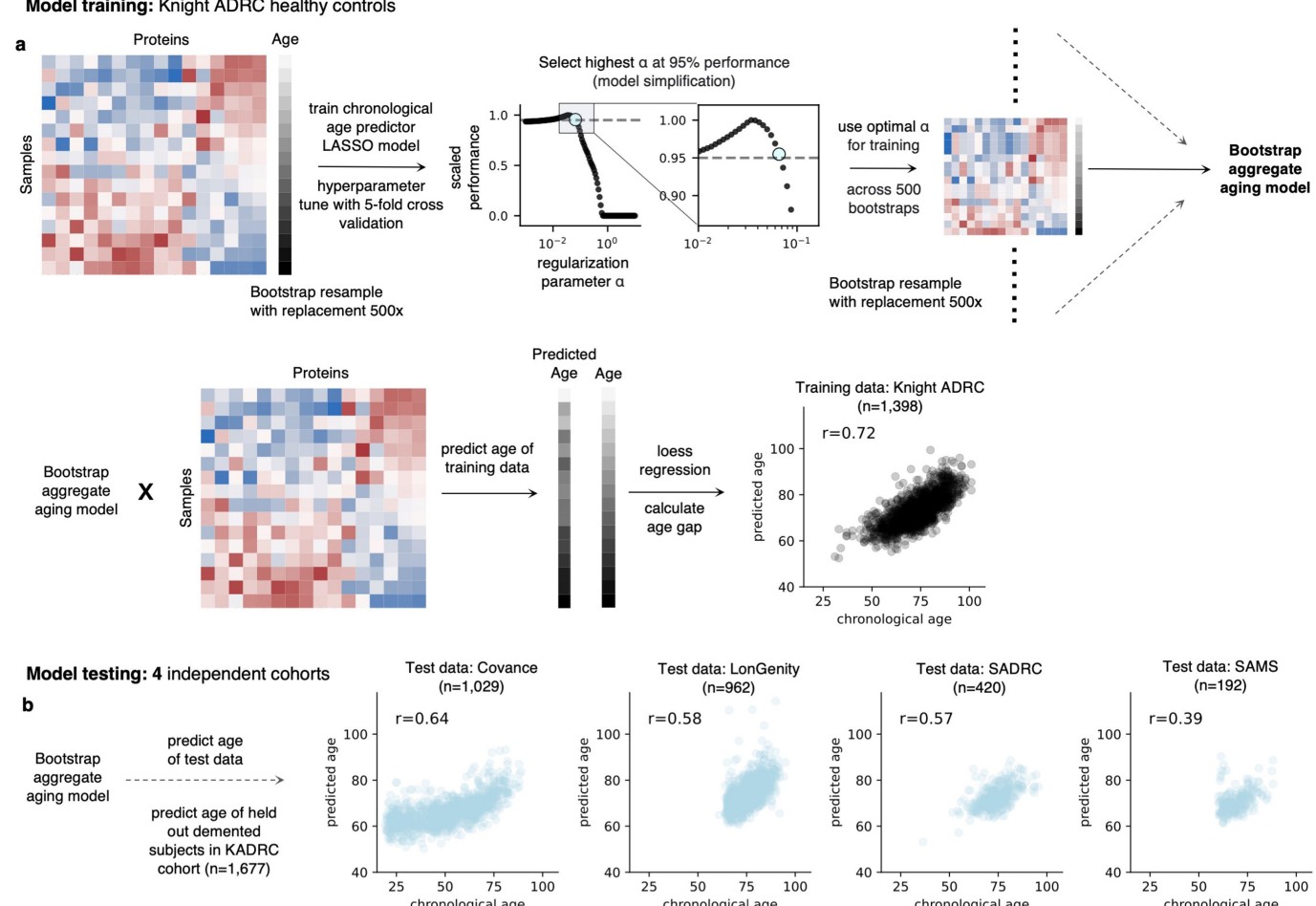

**Extended Data Fig. 2 | Aging model training and testing. a**, A bagged ensemble of least absolute shrinkage and selection operator (LASSO) aging models was trained for each of 11 major organs using the mutually exclusive organ-enriched proteins identified as inputs. An "organismal" aging model using the 3907 organ-nonspecific proteins and a "conventional" aging model using all 4778 QC'ed proteins on the SomaScan assay were also trained. Models were trained from the 1,398 healthy individuals in the Knight-ADRC cohort.

To reduce overfitting, the LASSO regularization parameter α was determined with bootstrap resampling by selecting sparser model α that provided 95% of maximum training set performance. An individual's predicted age was defined as the average predicted age across all bootstrapped models. The entire model training scheme for a single example aging model is shown. **b**, Models were tested in four independent cohorts (Covance, LonGenity, Stanford-ADRC, SAMS). Age predictions from a single example aging model across test cohorts is shown.

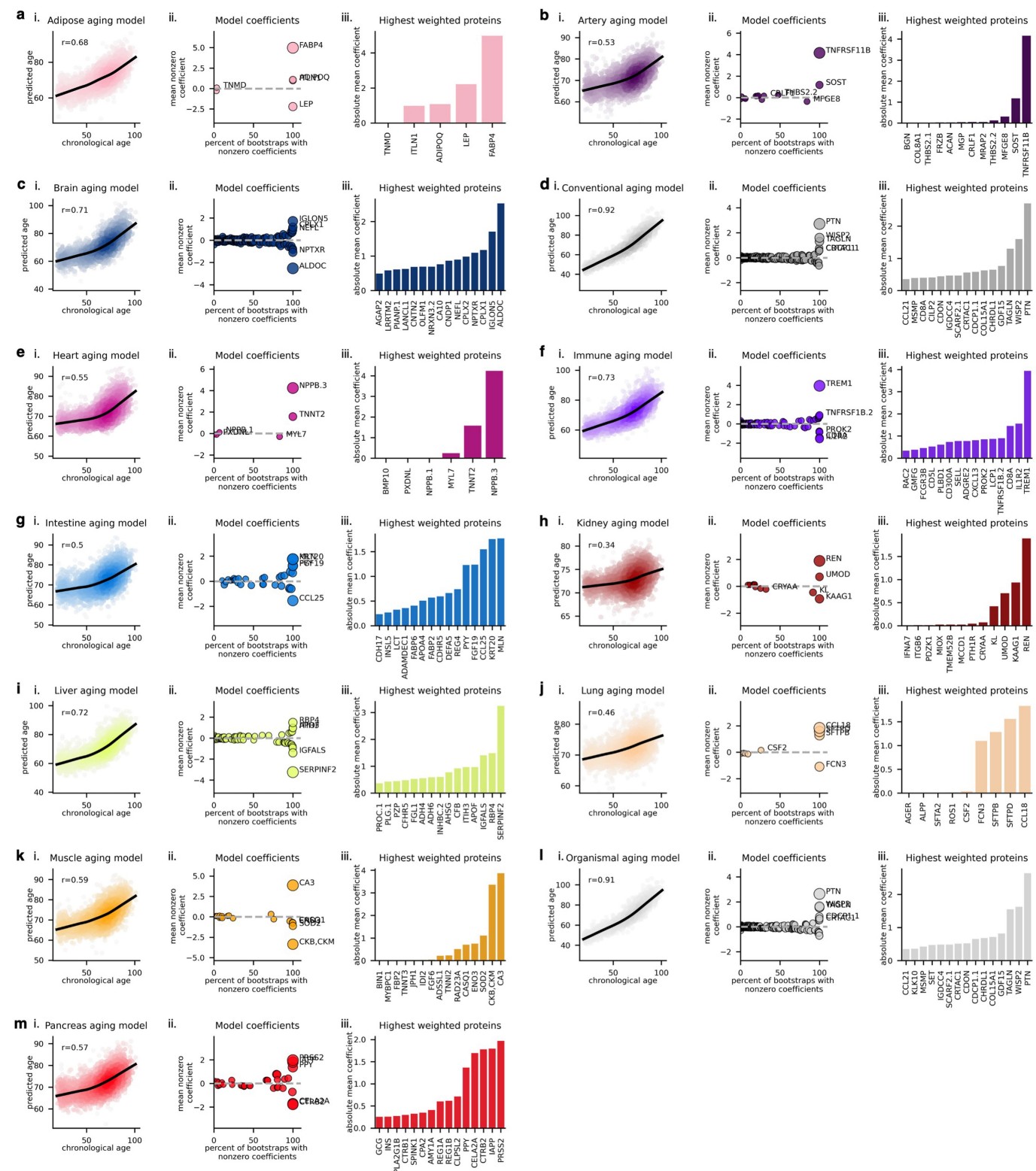

**Extended Data Fig. 3 | Aging model prediction and coefficients. a-m**, Aging model age prediction (**i**), average coefficients across bootstraps (**ii**) and top 15 coefficients (**iii**) are shown for all aging models in alphabetical order. (See ST7).

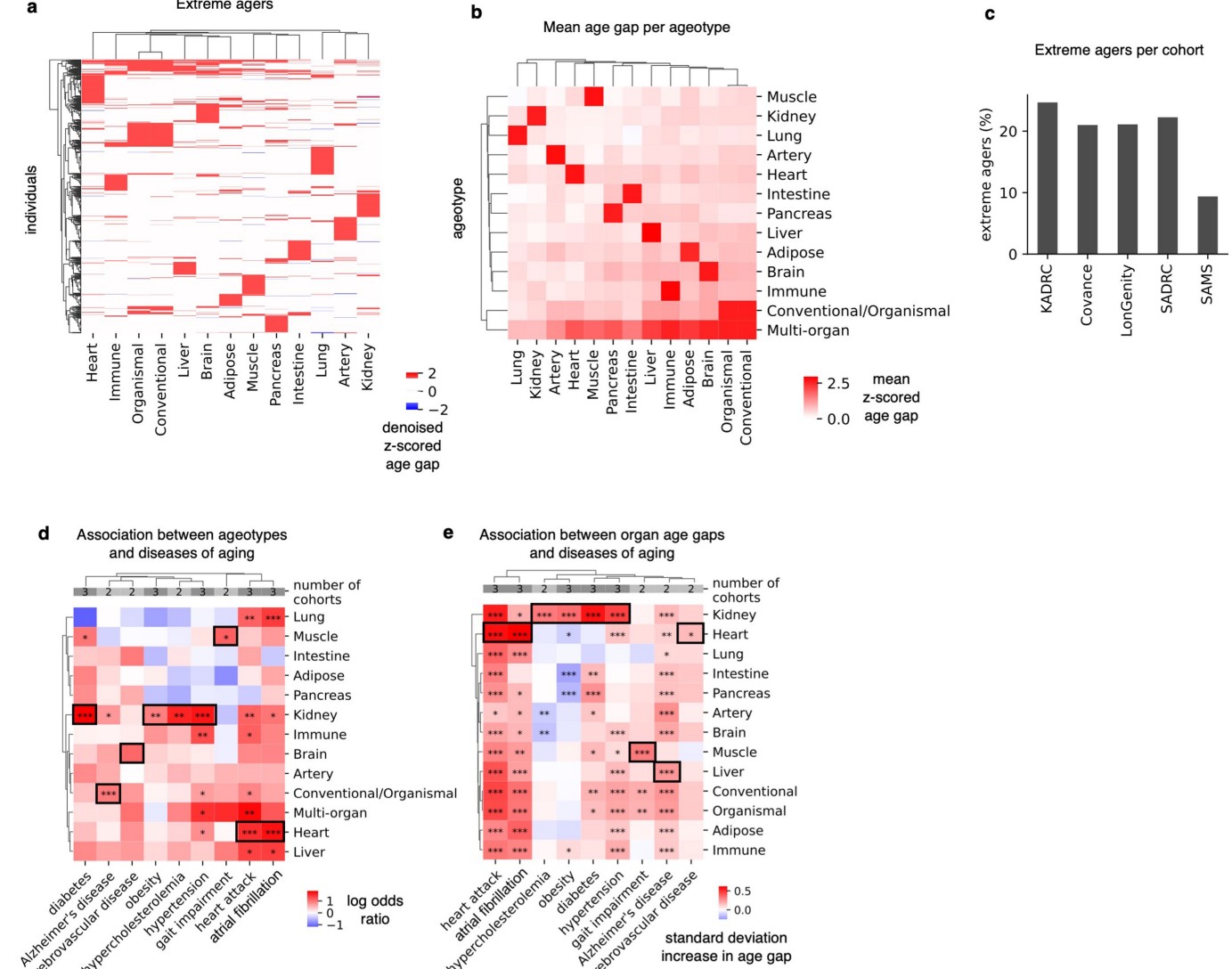

**Extended Data Fig. 4 | Extreme organ agers are widespread in the population.**
**a**, Extreme agers were defined as individuals with a 2-standard deviation increase or decrease in at least one age gap. 23% of the individuals (n = 5,676) across the four cohorts were identified as extreme agers. To visualize all extreme agers, age gaps were denoised by setting values below absolute z-score of 2 to zero. Denoised age gaps are shown in the heatmap. **b**, Extreme ageotypes were defined based on kmeans clustering of individuals based on their denoised age gaps. The mean z-scored age gap per ageotype is shown. **c**, The percentage of extreme agers is shown across all cohorts. **d**, A cross-cohort meta-analysis of associations (logistic regression) between extreme ageotypes versus diagnosis

of 9 major age-related diseases annotated in at least 2 independent cohorts. Log odds ratios and significance are shown. P-values were Benjamini-Hochberg FDR-corrected. The strongest associations per disease are highlighted with black borders. (See ST9). **e**, A cross-cohort meta-analysis of associations (linear regression) between organ age gaps versus diagnosis of 9 major age-related diseases annotated in at least 2 independent cohorts. Disease covariate effects and significance are shown. P-values were Benjamini Hochberg FDR-corrected. The strongest associations per disease are highlighted with black borders. (See ST10). Asterisks represent q-value thresholds: *q < 0.05; **q < 0.01; ***q < 0.001.

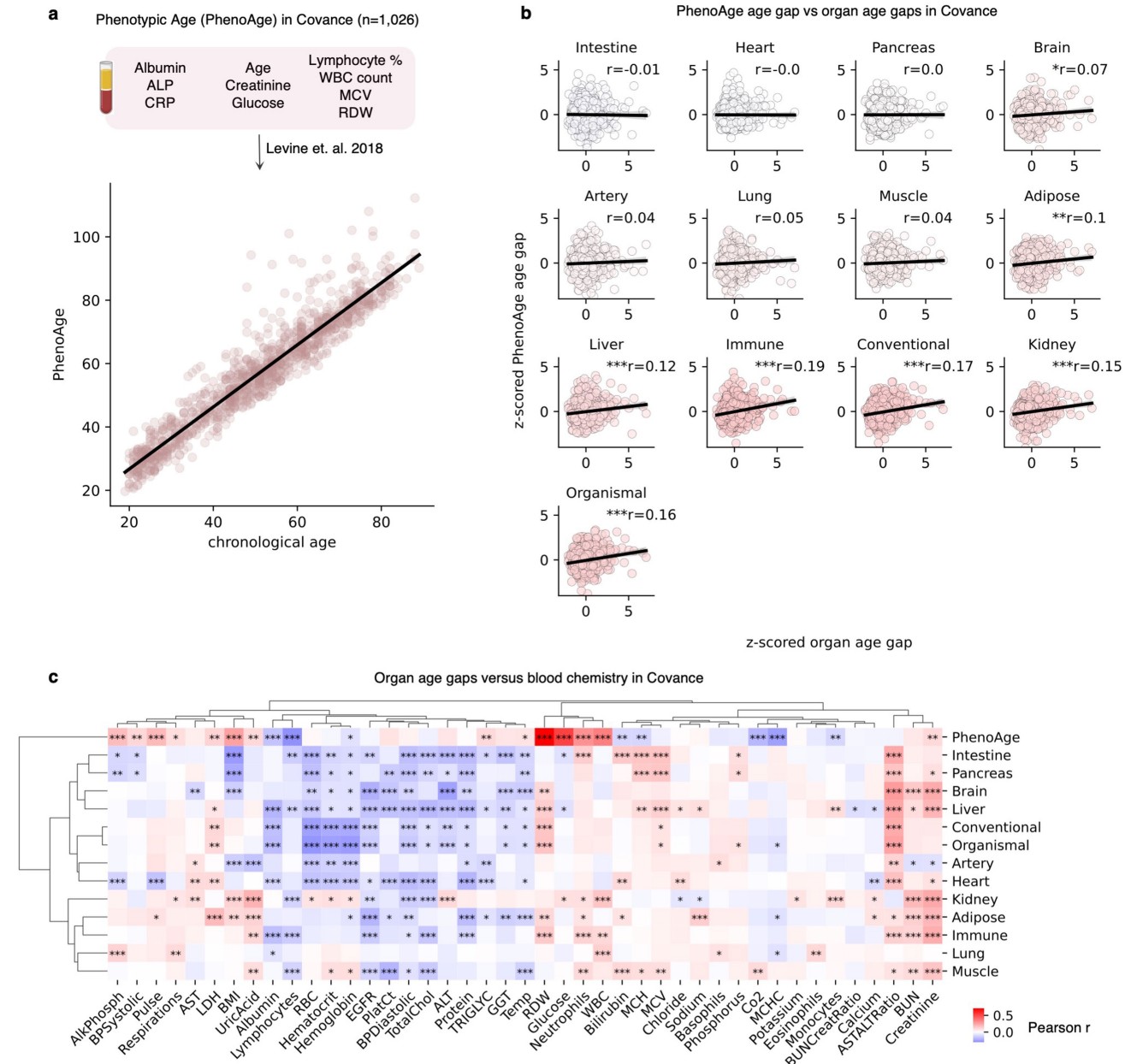

**Extended Data Fig. 5 | Plasma proteomic organ aging models versus established clinical markers of aging, health, and disease. a**, Phenotypic Age (PhenoAge, Levine et al. 2018) was calculated based on 10 clinical markers in the Covance cohort (n = 1,026). PhenoAge-based age prediction is shown. **b**, The PhenoAge age gap was calculated and correlated with plasma proteomic organ aging model age gaps. Pairwise correlations are shown. **c**, Organ age gaps and the PhenoAge age gap were associated with 43 individual clinical markers of health and disease. Phenotype covariate effect sizes and significance based on Benjamini Hochberg FDR corrected p-values for all associations are shown. Asterisks represent q-value thresholds: *q < 0.05; **q < 0.01; ***q < 0.001. (See ST14).

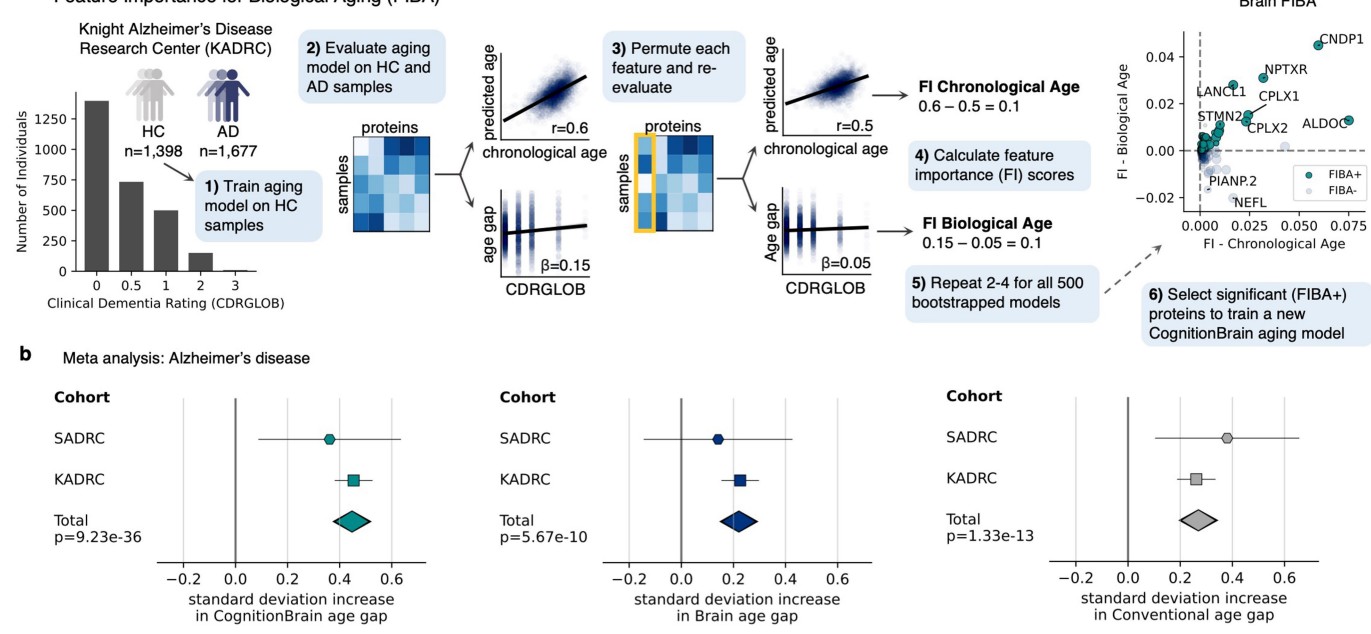

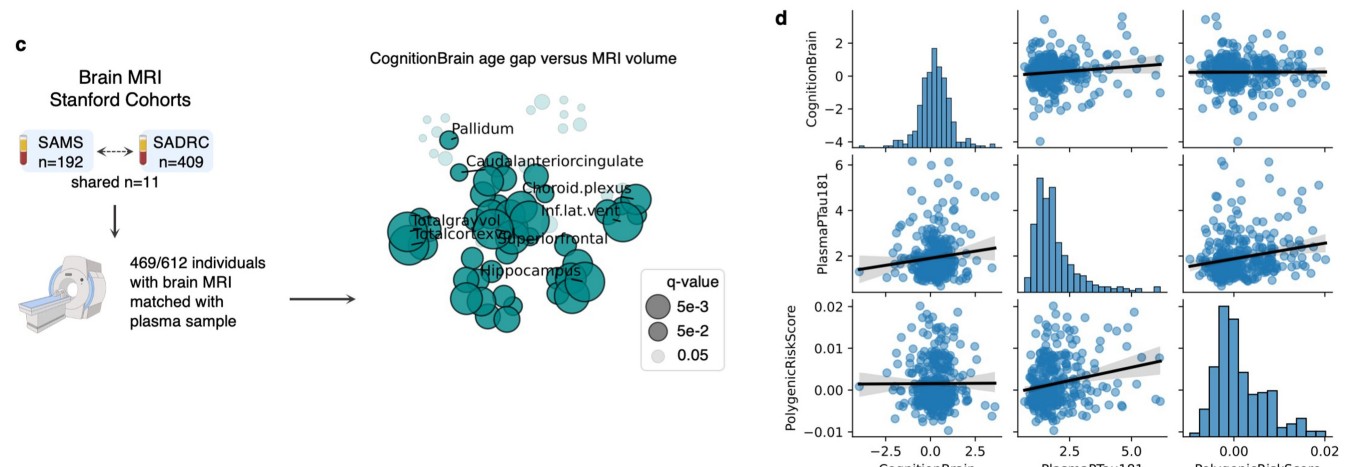

CognitionBrain age gap vs clinical markers of Alzheimer's disease

**Extended Data Fig. 6 | Feature Importance for Biological Aging (FIBA) to derive a cognition-associated brain aging model. a**, Schematic for FIBA algorithm, (see methods) an algorithm to assess brain aging model protein contributions to the brain age gap association with cognition and chronological age prediction accuracy. FIBA+ brain aging model proteins were used to train a new cognition-optimized brain aging model (CognitionBrain) from healthy individuals in the Knight-ADRC cohort. **b**, A cross-cohort meta-analysis of the association (linear regression) between the CognitionBrain, Brain, and Conventional age gaps versus Alzheimer's disease (with AD n = 1,441,

without n = 2,052). CognitionBrain age gap p-value$_{meta}$ = 9.23 × 10$^{-36}$, effect size$_{meta}$ = 0.448; Brain age gap p-value$_{meta}$ = 5.67 × 10$^{-10}$, effect size$_{meta}$ = 0.221; Conventional age gap p-value$_{meta}$ = 1.33 × 10$^{-13}$, effect size$_{meta}$ = 0.270. (See ST10, ST20). **c**, CognitionBrain age gaps were associated with brain MRI volume in the Stanford-ADRC and SAMS cohorts (n = 469). CognitionBrain associations with individual brain region volumes shown. Bubbles are sized by FDR corrected p-value. (See ST22). **d**, Pairwise-correlations between the CognitionBrain age gap, plasma pTau-181, and AD polygenic risk score. All error bars represent 95% confidence intervals.

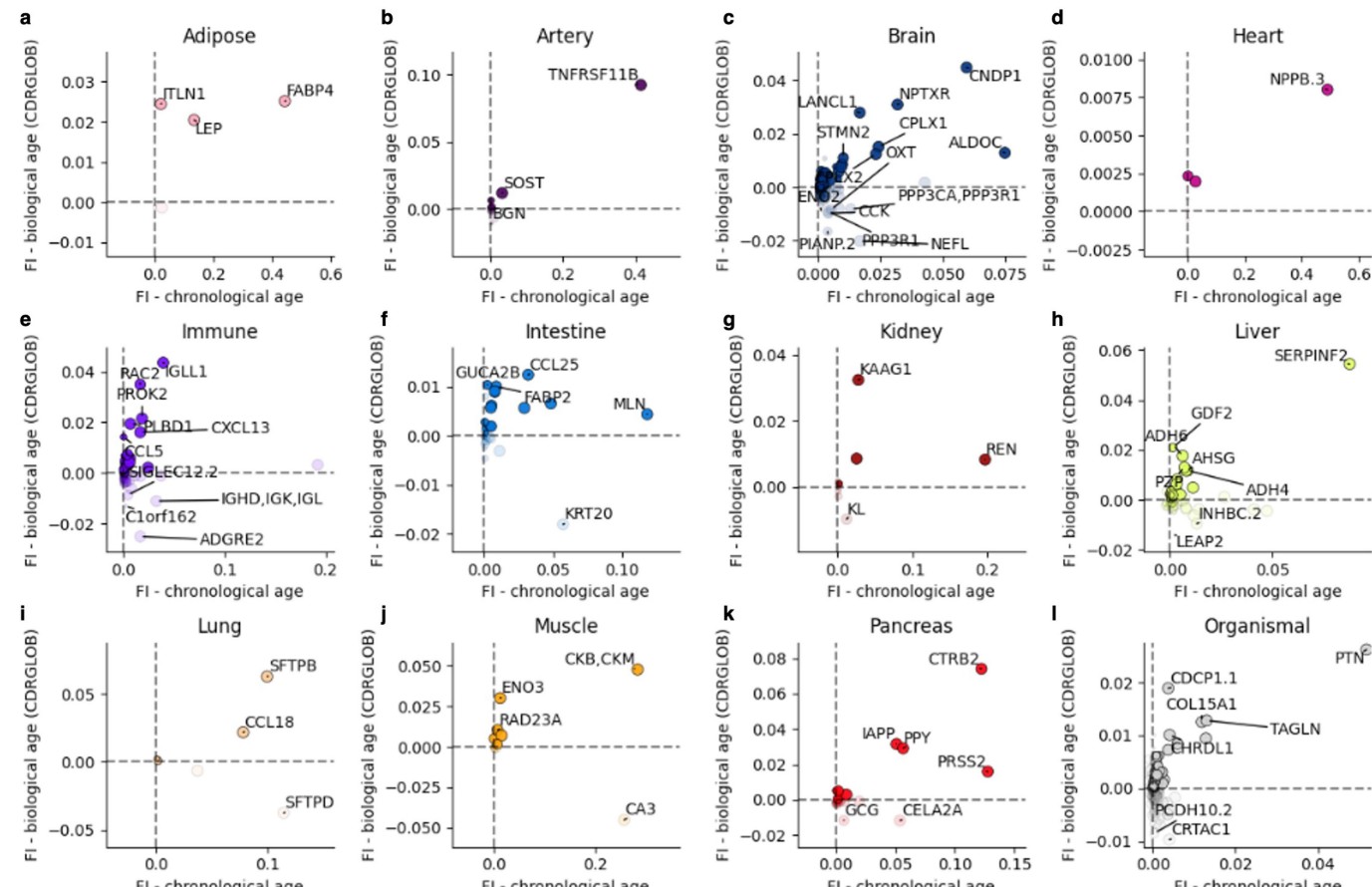

**Extended Data Fig. 7 | Feature Importance for Biological Aging (FIBA) plots for all aging models in relation to cognition. a**, FIBA was applied to all aging models to assess peripheral versus central contributions to brain aging and cognitive decline (CDR-Global dementia rating). For each aging model, proteins were assessed for their contributions to the age gap association with cognition (CDR-Global, y-axis) and chronological age prediction accuracy (x-axis). Proteins for which permutation reduces the age gap association with cognition were termed FIBA+ , while proteins for which permutation strengthens the age gap association with dementia were termed FIBA-. FIBA+ proteins were used to train new cognition-optimized aging models from healthy individuals in the Knight-ADRC cohort. FIBA results for all aging models are shown in alphabetical order. (See ST15).

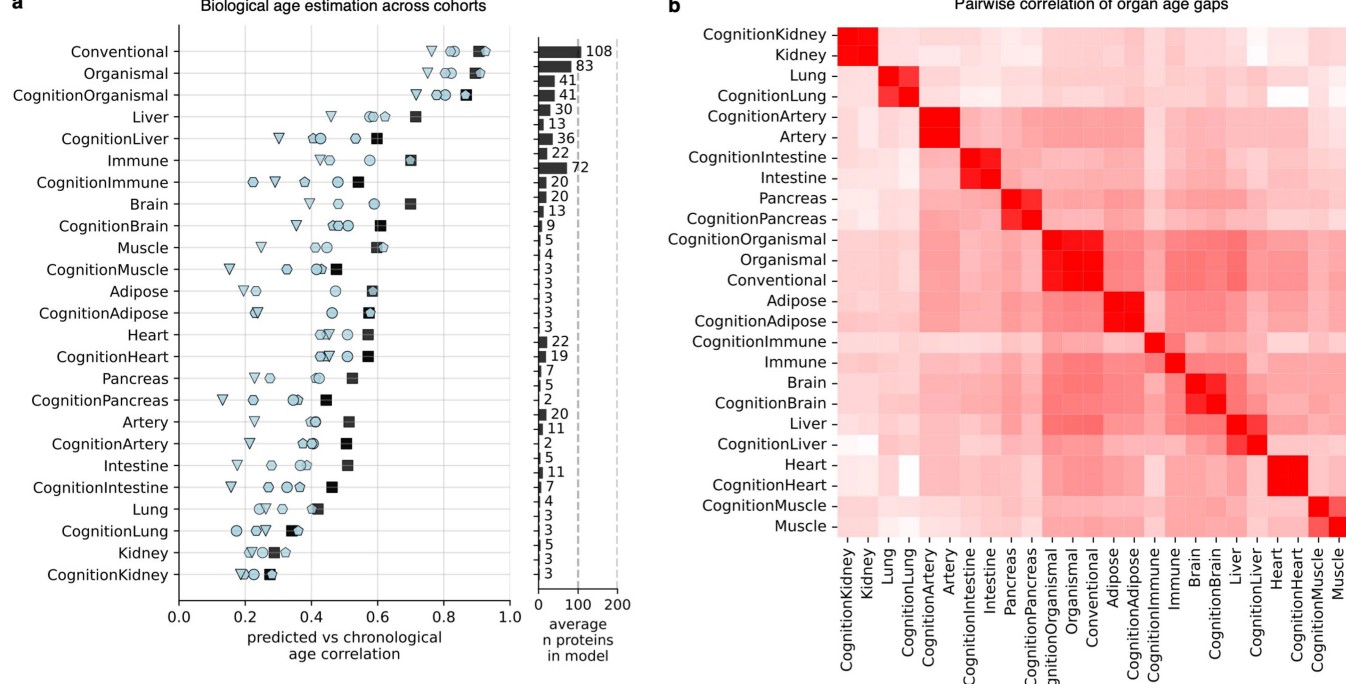

**a** Biological age estimation across cohorts

**b** Pairwise correlation of organ age gaps

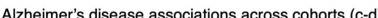

Alzheimer's disease associations across cohorts (c-d)

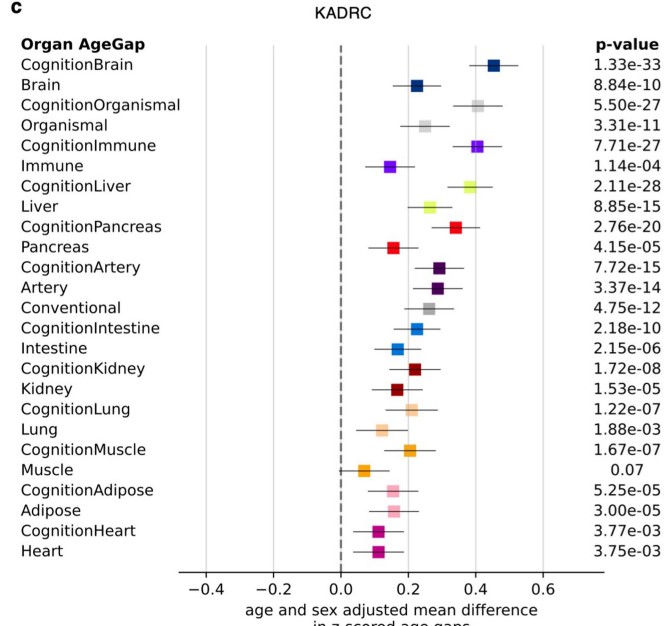

**c** KADRC

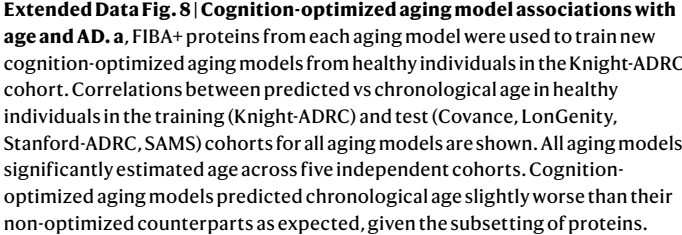

**d** SADRC

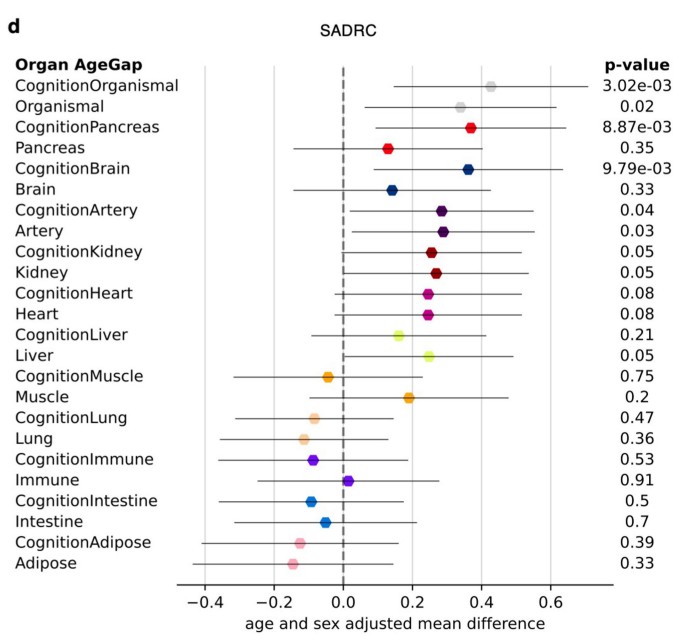

**Extended Data Fig. 8 | Cognition-optimized aging model associations with age and AD. a**, FIBA+ proteins from each aging model were used to train new cognition-optimized aging models from healthy individuals in the Knight-ADRC cohort. Correlations between predicted vs chronological age in healthy individuals in the training (Knight-ADRC) and test (Covance, LonGenity, Stanford-ADRC, SAMS) cohorts for all aging models are shown. All aging models significantly estimated age across five independent cohorts. Cognition-optimized aging models predicted chronological age slightly worse than their non-optimized counterparts as expected, given the subsetting of proteins.

(See ST19). **b**, Pairwise correlation of all model age gaps in all cohorts. Cognition-optimized aging models predicted similar age gap estimates with their non-optimized models. **c**, Model age gap associations (linear regression) with Alzheimer's disease (with AD n = 1,393, control n = 1,680) in the Knight-ADRC cohort. Effect sizes, 95% confidence intervals, and p-values for the Alzheimer's covariate are shown. Despite decreased associations with chronological age, cognition-optimized models showed substantially stronger associations with Alzheimer's disease. (See ST20). **d**, As in **c**, but in the Stanford-ADRC cohort (with AD n = 48, control n = 372). (See ST20).

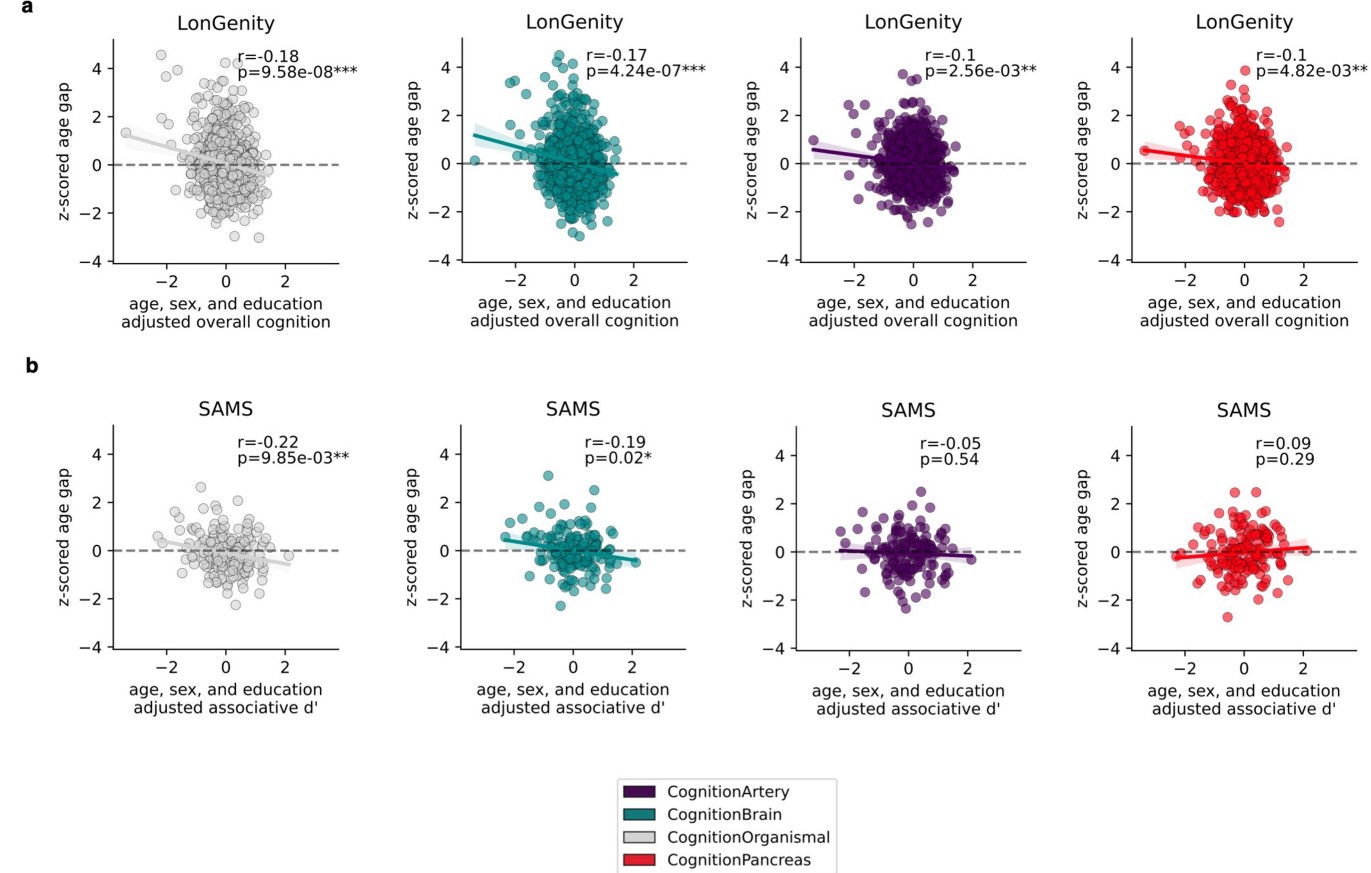

Organ age gap versus cognition in cognitively healthy individuals across cohorts

**Extended Data Fig. 9 | Cognition-optimized aging model associations with cognitive function in non-cognitively impaired individuals. a**, Associations (linear regression) between organ age gaps and a composite score of overall cognition in the LonGenity cohort (n = 888) shown. $p_{CognitionOrganismal} = 9.58 \times 10^{-8}$, $p_{CognitionBrain} = 4.24 \times 10^{-7}$, $p_{CognitionArtery} = 2.46 \times 10^{-3}$, $p_{CognitionPancreas} = 4.8 \times 10^{-3}$. (See ST23). **b**, Associations (linear regression) between organ age gaps and a memory test score in the SAMS cohort (n = 160) shown. $p_{CognitionOrganismal} = 9.85 \times 10^{-3}$, $p_{CognitionBrain} = 2.44 \times 10^{-2}$, $p_{CognitionArtery} = 0.53$, $p_{CognitionPancreas} = 0.29$. (See ST23).

**GTEX human organ bulk RNA-seq:** Top CognitionOrganismal proteins

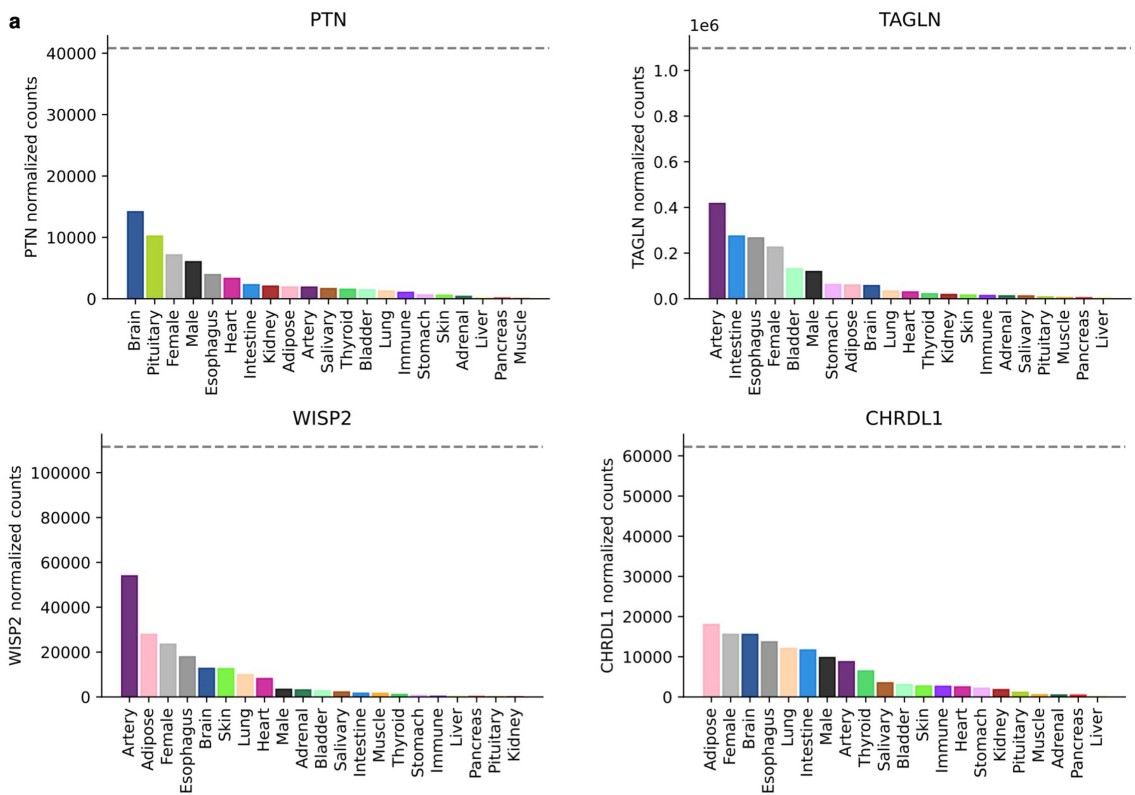

**Tabula Sapiens:** Human vasculature scRNA-seq

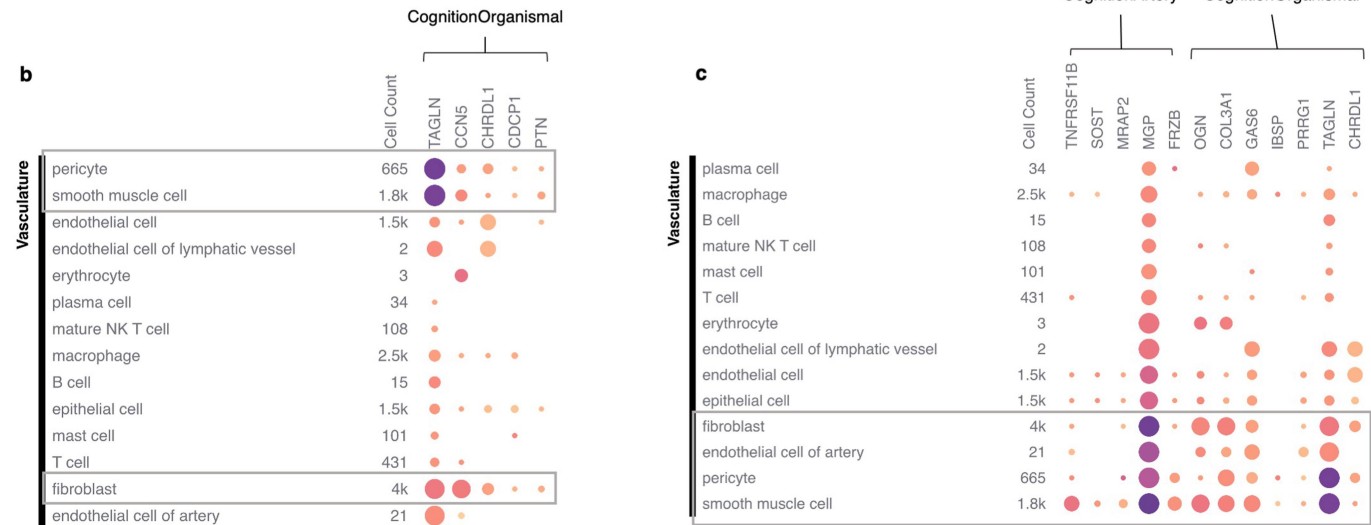

**Extended Data Fig. 10 | Mapping CognitionOrganismal and CognitionArtery proteins to human organs and cell types. a,** The organ sources of highly weighted CognitionOrganismal proteins were investigated by analyzing their expression levels in the Gene Tissue Expression Atlas (GTEx) bulk RNA-seq database. Organ-level expression of pleiotrophin (PTN), transgelin (TAGLN), WNT1 inducible signalling pathway protein 2 (WISP2), and chordin like 1 (CHRDL1) are shown. Though not organ-specific, these genes were highly expressed in the arteries and brain. **b,** Single-cell RNA expression (Tabula Sapiens) of highly weighted CognitionOrganismal proteins in human vasculature. Mean normalized expression values and fraction of cells expressing the genes are shown. **c,** Single-cell RNA expression (Tabula Sapiens) of highly weighted CognitionArtery and StringDB-based "interacting" proteins in human vasculature. Mean normalized expression values and fraction of cells expressing the genes are shown.

# Reporting Summary

## Statistics

For all statistical analyses, confirm that the following items are present in the figure legend, table legend, main text, or Methods section.

| n/a | Confirmed | |
|---|---|---|
| ☐ | ☒ | The exact sample size (*n*) for each experimental group/condition, given as a discrete number and unit of measurement |
| ☐ | ☒ | A statement on whether measurements were taken from distinct samples or whether the same sample was measured repeatedly |
| ☐ | ☒ | The statistical test(s) used AND whether they are one- or two-sided<br>*Only common tests should be described solely by name; describe more complex techniques in the Methods section.* |
| ☐ | ☒ | A description of all covariates tested |
| ☐ | ☒ | A description of any assumptions or corrections, such as tests of normality and adjustment for multiple comparisons |
| ☐ | ☒ | A full description of the statistical parameters including central tendency (e.g. means) or other basic estimates (e.g. regression coefficient) AND variation (e.g. standard deviation) or associated estimates of uncertainty (e.g. confidence intervals) |
| ☐ | ☒ | For null hypothesis testing, the test statistic (e.g. *F*, *t*, *r*) with confidence intervals, effect sizes, degrees of freedom and *P* value noted<br>*Give P values as exact values whenever suitable.* |
| ☒ | ☐ | For Bayesian analysis, information on the choice of priors and Markov chain Monte Carlo settings |
| ☒ | ☐ | For hierarchical and complex designs, identification of the appropriate level for tests and full reporting of outcomes |
| ☐ | ☒ | Estimates of effect sizes (e.g. Cohen's *d*, Pearson's *r*), indicating how they were calculated |
| | | *Our web collection on statistics for biologists contains articles on many of the points above.* |

## Software and code

Policy information about availability of computer code

Data collection    Plasma proteomics data were collected through Somalogic Inc's.

| Data analysis | Python 3.9.12<br>gprofiler-official==1.0.0<br>lifelines==0.27.3<br>matplotlib==3.5.1<br>numpy==1.21.5<br>pandas==1.4.2<br>scanpy==1.9.1<br>scikit-learn==1.0.2<br>scikit-survival==0.17.2<br>scipy==1.10.0<br>seaborn==0.12.2<br>statsmodels==0.13.5<br><br>R 4.1.2<br>metafor==4.2.0<br>DESeq2=1.38.3 |
|---|---|

For manuscripts utilizing custom algorithms or software that are central to the research but not yet described in published literature, software must be made available to editors and reviewers. We strongly encourage code deposition in a community repository (e.g. GitHub). See the Nature Portfolio guidelines for submitting code & software for further information.

# Data

Policy information about availability of data

All manuscripts must include a data availability statement. This statement should provide the following information, where applicable:

- Accession codes, unique identifiers, or web links for publicly available datasets
- A description of any restrictions on data availability
- For clinical datasets or third party data, please ensure that the statement adheres to our policy

Stanford-ADRC data are available upon reasonable request to the Stanford-ADRC data release committee, https://web.stanford.edu/group/adrc/cgi-bin/web-proj/datareq.php. All Stanford-ADRC data will be made publicly available after an embargo period at this link: https://twc-stanford.shinyapps.io/adrc/. SAMS data are available to qualified investigators upon request to principal investigators Beth Mormino (bmormino@stanford.edu) or Anthony Wagner (awagner@stanford.edu). Knight-ADRC data were generated by the lab of principal investigator Carlos Cruchaga (cruchagac@wustl.edu) and are available upon reasonable request to the NIAGADS (Study ID: ng00130), https://www.niagads.org/knight-adrc-collection. Data from the Covance and LonGenity cohorts can be accessed according to the policies described in the initial study publications51–53.

Pre-processed human heart73 and kidney74 scRNA-seq data were accessed from studies in the Human Cell Atlas. Pre-processed brain scRNA-seq data were accessed from Haney et. al. 202375. Pre-processed human brain vasculature scRNA-seq data were accessed from Yang et. al. 202242. Pre-processed human vasculature scRNA-seq data were accessed from Tabula Sapiens41. Differential expression statistics of proteins and RNA from Alzheimer's disease versus control brains were accessed from Johnson et. al. 202276.

Change with age information of ~5,000 SomaScan v4 plasma proteins across all 5 cohorts (Supplementary Fig. 8, Supplementary Table 25) are available in a public shiny app (https://twc-stanford.shinyapps.io/aging_plasma_proteome_v2/).

# Research involving human participants, their data, or biological material

Policy information about studies with human participants or human data. See also policy information about sex, gender (identity/presentation), and sexual orientation and race, ethnicity and racism.

| Reporting on sex and gender | "Models that included sex as a covariate and models trained separately on males and females showed similar age prediction performance on both sexes, so we controlled for sex to extend the generality of the findings and reduce analytic complexity (Supplementary Fig. 3 a-c)." |
|---|---|
| Reporting on race, ethnicity, or other socially relevant groupings | Biological differences between different ethnic/racial groups were not assessed in this study. Ethnicity distributions per cohort are available in Supplementary Table 1. |
| Population characteristics | Covance<br>Details of the Covance study have been previously published77. Briefly, Covance is a multi-site cross-sectional study of health across the lifespan collected at 5 hospital sites in the United States in 2008. 1028 subjects were included in analyses for this study. Cohort demographic characteristics are summarized in Supplementary Table 1. Exclusion criteria for the study included uncontrolled hypertension, self-reported treatment for a malignancy other than squamous cell or basal cell carcinoma of the skin in the last 2 years, self-reported pregnancy, self-reported chronic infection, autoimmune condition or other inflammatory condition, self-reported chronic kidney or liver disease, chronic heart failure or diagnosed with myocardial infarction in the last 3 months, self-reported diabetes (HbA1c>8% if known), self-reported acute bacterial or viral infection in the past 24 hours or a temperature > 38 C within 24 hours of enrollment, self-reported participation in any therapeutic study within 14 days prior of blood sampling, and taking more than 20 mg of prednisone or related drugs.<br><br>Clinical blood chemistry performed on the same samples, including a complete blood count and comprehensive metabolic panel, lipid panel, and liver function tests. Basic physical workup (blood pressure, pulse, respirations) was also collected. |

Lifestyle information was also collected from all participants using a survey which asked about smoking, alcohol, exercise, habits, and frequency of consumption of different meats and vegetables.

LonGenity
Details of the LonGenity cohort have been previously published78,79. Briefly, LonGenity is an ongoing longitudinal study initiated in 2008 and designed to identify biological factors that contribute to healthy aging. The LonGenity study enrolls older adults of Ashkenazi Jewish descent with age 65–94 years at baseline. Approximately half of the cohort consists of offspring of parents with exceptional longevity, defined as having at least one parent who survived to 95 years of age. The other half of the cohort includes offspring of parents with usual survival, defined as not having a parental history of exceptional longevity. 962 subjects were included in analyses for this study. Cohort characteristics are summarized in Supplementary Table 1. LonGenity participants are thoroughly characterized demographically and phenotypically at annual visits that include collection of medical history and physical and detailed neurocognitive assessments (described in detail below). The LonGenity study was approved by the institutional review board (IRB) at the Albert Einstein College of Medicine.

Subjects in the LonGenity cohort underwent extensive cognitive examination. The Overall Cognition Composite score was determined by the relative performance of the subject in the Free and Cued Selective Reminding Test, WMS-R Logical Memory I, RBANS Figure Copy, RBANS Figure Recall, WAIS-III Digit Span, WAIS-III Digit Symbol Coding, Phonemic Fluency (FAS), Categorical Fluency, Trail Making Test A, and Trail Making Test B. For each task a standardized score (z) was calculated based on the population. The z for each task is then combined to create the Overall Cognition Composite.

Stanford Alzheimer's Disease Research Center (Stanford-ADRC)
Samples were acquired through the National Institute on Aging (NIA)-funded Stanford Alzheimer's Disease Research Center (Stanford-ADRC). The Stanford-ADRC cohort is a longitudinal observational study of clinical dementia subjects and age-sex-matched non-demented subjects. The collection of plasma was approved by the Institutional Review Board of Stanford University and written consent was obtained from all subjects. Blood collection and processing were done according to a rigorous standardized protocol to minimize variation associated with blood draw and blood processing. Briefly, about 10 cc whole blood was collected in a vacutainer EDTA tube (BD Vacutainer EDTA tube) and spun at 3000RPM for 10 mins to separate out plasma, leaving 1 cm of plasma above the buffy coat and taking care not to disturb the buffy coat to circumvent cell contamination. Plasma processing times averaged approximately one hour from the time of the blood draw to the time of freezing and storage. All blood draws were done in the morning to minimize the impact of circadian rhythm on protein concentrations. Plasma pTau-181 levels were measured using the fully-automated Lumipulse G 1200 platform (Fujirebio US, Inc, Malvern, PA) by experimenters blind to diagnostic information, as previously described42.

All healthy control participants were deemed cognitively unimpaired during a clinical consensus conference that included board-certified neurologists and neuropsychologists. Cognitively impaired subjects underwent Clinical Dementia Rating and standardized neurological and neuropsychological assessments to determine cognitive and diagnostic status, including procedures of the National Alzheimer's Coordinating Center (https://naccdata.org/). Cognitive status was determined in a clinical consensus conference that included neurologists and neuropsychologists. All participants were free from acute infectious diseases and in good physical condition. 412 subjects were included in analyses for this study. Cohort demographics and clinical diagnostic categories are summarized in Supplementary Table 1.

Stanford Aging Memory Study (SAMS)
SAMS is an ongoing longitudinal study of healthy aging. Blood collection and processing were done by the same team and using the same protocol as in Stanford-ADRC. Neurological and neuropsychological assessment were done by the same team and using the same protocol as in Stanford-ADRC. All SAMS participants had CDR=0 and neuropsychological test score within normal range; all SAMS participants were deemed cognitively unimpaired during a clinical consensus conference that included neurologists and neuropsychologists. 192 cognitively SAMS participants were included in the present study, and 11 were participants in both the Stanford-ADRC and SAMS study. Cohort demographics and clinical diagnostic categories are summarized in Supplementary Table 1.

Knight Alzheimer's Disease Research Center (Knight-ADRC)
The Knight ADRC (Knight-ADRC) cohort is an NIA-funded longitudinal observational study of clinical dementia subjects and age-matched controls. Research participants at the Knight-ADRC undergo longitudinal cognitive, neuropsychologic, imaging, and biomarker assessments including Clinical Dementia Rating (CDR). Among individuals with CSF and plasma data, AD cases corresponded to those with a diagnosis of dementia of the Alzheimer's type (DAT) using criteria equivalent to the National Institute of Neurological and Communication Disorders and Stroke-Alzheimer's Disease and Related Disorders Association for probable AD80 and AD severity was determined using the Clinical Dementia Rating (CDR®)81 at the time of lumbar puncture (for CSF samples) or blood draw (for plasma samples). Controls received the same assessment as the cases but were non-demented (CDR=0). Blood samples were collected in EDTA tubes (BD Vacutainer purple top) at the visit time, immediately centrifuged at 1500g for 10 minutes, aliquoted on 2D barcoded Micronic tubes (200ul per aliquot) and stored at – 80°C. The plasma was stored in monitored -80C freezer until it was pulled and sent to Somalogic for data generation. The Institutional Review Board of Washington University School of Medicine in St. Louis approved the study and research was performed in accordance with the approved protocols. 3075 participants were included in the present study. Cohort demographics and clinical diagnostic categories are summarized in Supplementary Table 1.

| Recruitment | Participants were recruited through the NIA-funded Stanford Alzheimer's Disease Research Center and the Knight Alzheimer's Disease Research Center. Recruitment of healthy controls is biased towards significant others/partners/relatives of diseased patients, which may affect findings. |
|---|---|
| Ethics oversight | LonGenity cohort: Institutional Review Board of the Albert Einstein College of Medicine. Stanford-ADRC and SAMS cohorts: Institutional Review Board of Stanford University. Knight-ADRC cohort: Institutional Review Board of Washington University School of Medicine in St. Louis |

Note that full information on the approval of the study protocol must also be provided in the manuscript.

# Field-specific reporting

Please select the one below that is the best fit for your research. If you are not sure, read the appropriate sections before making your selection.

☒ Life sciences  ☐ Behavioural & social sciences  ☐ Ecological, evolutionary & environmental sciences

For a reference copy of the document with all sections, see nature.com/documents/nr-reporting-summary-flat.pdf

# Life sciences study design

All studies must disclose on these points even when the disclosure is negative.

| | |
|---|---|
| Sample size | No power analyses were used to predetermine sample sizes. However, sample sizes were informed by prior literature using similar experimental paradigms that yielded interpretable results and the lab's previous experience. For example in Lehallier et al. Nature Medicine 2019, we measured ~3,000 proteins from 4,263 individuals. Here we measure ~5,000 from 5,676 individuals. |
| Data exclusions | Samples flagged by Somalogic for quality control were excluded. |
| Replication | All plasma proteomic measurements were acquired from unique human samples. |
| Randomization | Samples from healthy individuals and individuals with neurodegenerative disease were age and sex matched. |
| Blinding | Training and testing of models were performed on completely independent datasets. Investigators responsible for plasma and data collection were blinded to patient demographics. |

# Reporting for specific materials, systems and methods

We require information from authors about some types of materials, experimental systems and methods used in many studies. Here, indicate whether each material, system or method listed is relevant to your study. If you are not sure if a list item applies to your research, read the appropriate section before selecting a response.

## Materials & experimental systems

| n/a | Involved in the study |
|---|---|
| ☒ ☐ | Antibodies |
| ☒ ☐ | Eukaryotic cell lines |
| ☒ ☐ | Palaeontology and archaeology |
| ☒ ☐ | Animals and other organisms |
| ☒ ☐ | Clinical data |
| ☒ ☐ | Dual use research of concern |
| ☒ ☐ | Plants |

## Methods

| n/a | Involved in the study |
|---|---|
| ☒ ☐ | ChIP-seq |
| ☒ ☐ | Flow cytometry |
| ☐ ☒ | MRI-based neuroimaging |

# Magnetic resonance imaging

## Experimental design

| | |
|---|---|
| Design type | Resting state |
| Design specifications | NA |
| Behavioral performance measures | NA |

## Acquisition

| | |
|---|---|
| Imaging type(s) | structural |
| Field strength | 3T |
| Sequence & imaging parameters | Whole-brain MRI scans were collected from all subjects in the Stanford-ADRC and SAMS cohorts. All MRI data was collected at the Stanford Richard M. Lucas Center for Imaging.  271 subjects underwent MRI scanning on a 3T MRI scanner (GE Discovery MR750). T1-weighted SPGR scans were collected (TR/TE/TI=8.2/3.2/900ms, flip angle=9, 1x1x1mm) and used to define gray matter volumes. 134 subjects underwent MRI scanning on hybrid PET/MRI scanner (Signa 3 tesla, GE Healthcare).  T1-weighted SPGR scan were collected (TR/TE/TI=7.7/3.1/400ms, flip angle=11, 1.2x1.1x1.1mm) and used to define gray matter volumes. |
| Area of acquisition | whole brain |

Diffusion MRI      ☐ Used      ☒ Not used

## Preprocessing

| | |
|---|---|
| Preprocessing software | Region of interest (ROI) labeling was implemented using the FreeSurfer100 software package version 7 (http://surfer.nmr.mgh.harvard.edu). In brief, structural images were bias field corrected, intensity normalized, and skull stripped using a watershed algorithm. These images underwent a white matter-based segmentation, grey/white matter and pial surfaces were defined, and topology correction was applied to these reconstructed surfaces. Subcortical and cortical ROIs spanning the entire brain were defined in each subject's native space, using the aparc+aseg atlas in FreeSurfer. |
| Normalization | T1-w SPGR scans were normalized according to FreeSurfer's recon-all command |
| Normalization template | MNI305 |
| Noise and artifact removal | NA |
| Volume censoring | NA |

## Statistical modeling & inference

| | |
|---|---|
| Model type and settings | basic multivariate linear models were used to associate brain ROI volumes with plasma proteomic aging signatures, while accounting for age and sex. |
| Effect(s) tested | only brain ROI volumes were assessed. |

Specify type of analysis:      ☐ Whole brain      ☒ ROI-based      ☐ Both

| | |
|---|---|
| Anatomical location(s) | Subcortical and cortical ROIs spanning the entire brain were defined in each subject's native space, using the aparc+aseg atlas in FreeSurfer. |

Statistic type for inference

(See Eklund et al. 2016)

*Specify voxel-wise or cluster-wise and report all relevant parameters for cluster-wise methods.*

Correction      *Describe the type of correction and how it is obtained for multiple comparisons (e.g. FWE, FDR, permutation or Monte Carlo).*

## Models & analysis

| n/a | Involved in the study |
|---|---|
| ☒ | ☐ Functional and/or effective connectivity |
| ☒ | ☐ Graph analysis |
| ☐ | ☒ Multivariate modeling or predictive analysis |

| | |
|---|---|
| Multivariate modeling and predictive analysis | Using matched brain MRI and plasma proteomic data from n=541 samples in SAMS and Stanford-ADRC, we compared our plasma proteomic organ clocks with established brain MRI based-clocks, brainageR and BARACUS Brain-Age.<br><br>We used a pre-trained machine learning algorithm (https://github.com/james-cole/brainageR) and raw T1-weighted MRI scans to estimate brain age. This software uses SPM12 (https://www.fil.ion.ucl.ac.uk/spm/software/spm12/) to perform tissue segmentation and normalization of individual scans to Montreal Neurological Institute (MNI) template space. The software relies on a model that used Gaussian process regression to predict brain age on 3,777 participants from seven publicly available datasets (mean age = 40.1, range = 18-90 years). It applies the results of this training to predict brain age in any new T1-w data, utilizing the RNifti (version 1.4.5) and kernlab (version 0.9-32) packages within R version 4.2. |

We also used another pre-trained algorithm, BARACUS (https://github.com/bids-apps/baracus; Liem et al. 2017), to estimate brain age from FreeSurfer version 5.3 processed T1-w scans. The vertex-wise cortical thickness and surface area values (transformed from subject space to fsaverage4 standard space), along with the subcortical volumetric statistics, were used as input to BARACUS's linear support vector machine model. This model was trained on 1,166 participants with no objective cognitive impairment (566 female, mean age = 59.1, range = 20-80 years). It returns a "stacked-anatomy" prediction among its results, which we used as the estimate of brain age for this method.

