## [Peer Review File · Nature]

Manuscript Title: Organ aging signatures in the plasma proteome track health and disease

Reviewer Comments & Author Rebuttals

Reviewer Reports on the Initial Version:

Referees' comments:

Referee #1 (Remarks to the Author):

Manuscript background information

While various attempts have been made to define “biological age”, the key problem with “clock” models - trained to mimic chronologic age - has been in the (sometimes spurious) attribution of meaning to the “age gap” - an individual’s deviation from a model’s prediction, and subsequent confusion about what kind of health intervention might help compensate for adverse deviations. The authors of this study have addressed these problems in a very interesting and novel manner.

From a larger study of 1,029 participants (the Covance study), for model training they focused on 353 healthy participants’ plasma, aged from 60-97, measuring ~5000 proteins in each sample using the SomaScan modified-aptamer platform. They pre-defined organ-related subsets of these proteins (19 subsets involving 793 of the measured proteins) as “organ-enriched” if they had at least 4-fold increased RNA expression over any other organ documented in the Genotype-Tissue Expression project. They used machine learning (the LASSO algorithm) to select the most important protein features and build age-predictive models based on age-associated subsets each organ’s “enriched” candidate proteins, as well as an “organismal” age-predictor model using all available proteins as candidates without prior enrichment. They then applied these models to plasma samples measured using the same proteomic platform to 962 participants in the LonGenity study and 412 participants in the S-ARDC study. The relation of the protein models’ organ-predicted age-gaps with adverse outcomes related to each organ, and to all-cause mortality, was observed as validating the meaningfulness of such gaps. The authors also evaluated the degree of similarity of age-gaps across organs in order to evaluate whether there was evidence of differential organ aging. The exception to this process was for neurodegeneration; apparently because brain age-gaps derived from the training cohort using the gene-expression organ-enrichment did not associate with Alzheimer or Parkinson initially, a new selection process was applied to identify proteins within the LonGenity cohort related to cognition, and then these proteins were re-trained against age in the Covance study.

The degree of association between chronologic age and each model’s predicted age was variable with r -values of 0.2 to 0.7, whereas the organismal model was rather better as a simple “clock”. Additionally, the organismal model was the best at predicting overall mortality, with a hazard ratio of ~2 compared with the other models’ 1.0-1.5.

When evaluating the relation of age-gap to organ-related phenotypes, there were significant associations between the heart age-gap and arrhythmia, coronary artery disease and heart failure. For the kidney model age-gap, there were significant associations with estimated glomerular filtration rate (eGFR). There was also face-validity in the declared association of diabetes with

pancreas and kidney age-gaps, motor issues with muscle age-gaps but these associations were not shown in any of the main figures or charts. Key proteins involved in these models such as NT-proBNP and troponin in the heart model were also plausible. The initial neurodegeneration model's age-gap failed to relate to Alzheimer (AD) or Parkinson (PD), but the re-selected and re-trained model did so – individuals with these conditions appeared to have older brains, and correlated to pTau-181, a known marker of Alzheimer. For pituitary, esophagus and intestine, there was “insufficient data” to evaluate the age-gaps' relation to phenotype.

Figure 1 has 7 diagrams and plots, and shows the overall process and the degree of correlation between predicted and actual age for the organismal model (r -values of ~ 0.7 - 0.9) and the organ-specific models (r -values of 0.2 to 0.7). The hazard ratios for the age-models' predictions to 15-year mortality ranging from 1.0 to 1.8 , and the correlation of age-gaps across different models was modest at best, which the authors infer to represent differential organ aging.

Figure 2 has 19 diagrams and plots, and describes the results for the heart-aging and kidney-aging models, their relation to adverse phenotypes and to protein networks, and Figure 3, with 13 diagrams and plots shows the different approach used and results for the brain age model.

The extended data shows the origin of the organ-enriched candidate proteins and the proportion measured by SomaScan (Figure 1), the proportion of these proteins with a relation to age (figure 2), the correlations of each model to chronological age (Figure 3) and the features selected for each model (Figure 4). Figure 5 shows the age-gap relation for 8 models with approximately 100 measures of phenotype. Figure 6 shows the special process applied to derive a brain aging model that did relate to AD and PD and Figure 7 the timing of evaluations for the neurodegeneration program.

In the methods section, the three cohorts are described (with Covance, the training dataset second, should be first), the proteomic platform overview, and imaging methods in S-ADRC. The process for defining organ-enriched proteins is described, with a new complexity of the pair-enriched process. The LASSO machine-learning process and model-building. The method of relating organ-age gap to phenotype was briefly related. Cox proportional hazards models were used to assess the relation of age-gaps with mortality, CHF and dementia with age and sex as covariates.

The supplement contains technical notes on the proteomic assay platform from the manufacturer.

The premise of this study is very original and creative – that the key weakness of organismal “clock” models, namely the attribution of meaning to deviations, can be overcome through the development of organ specific models made from a-priori enrichment of a wide selection of proteins, and then trained against age. The inclusion of three different data sets of reasonable size, and the design using one for model training and two for validation of age-gaps' relation to phenotypes is also rigorous. I think that the program is ultimately publishable, but there are a number of major issues that would need to be addressed:

- 1) The most important evaluation in this study that actually addresses the thesis of whether the organ-enriched age model approach actually advances the field is the relation between age-gaps and phenotypes. But the data presented is quite incomplete: there were apparently 19 original organ-specific models created, and then a new brain model, and the organismal model. That's 21 models. Maybe three more than that if the Cox models were additional. Age-gap to phenotype results are only shown for three of them in the main manuscript (heart, kidney and the new brain model). While 8 models are shown in extended data Figure 5, and three were declared as having insufficient data in the text, by my calculation this still leaves 7 models' age-gap results completely missing. While 16

models' results vs. age and mortality are shown in Figure 1, this is not the same as an age-gap and again is an inconsistent number vs. the whole. Given the importance of this component, a chart in the main text is needed that includes a) all 21 models b) how many candidate phenotypes was each model's age-gap evaluated against in total and c) of the evaluated phenotypes, how many were statistically significantly associated with age-gap when corrected for multiple comparisons d) some measure of the effect size for the subset that were significant, perhaps borrowed from Extended Data Figure 5. Without that data, the selected age gap results in the text and figures seem like anecdotes that have been picked because they worked.

2) Statistical issues:

a. I am worried about false discovery rates. There are a couple of warning flags that might simply need clarification, or alternatively might represent a lack of control, which would make this entire program a hypothesis-generating study rather than a validation study. First, it is not clear how many phenotypes were evaluated – there are 100 in Extended Data Figure 5. Given this is a nice round number, I suspect that is only the most illustrative subset of the whole number of possibilities. Any p-values for relations to age-gaps should be corrected for the total number multiple comparisons evaluated. This may well be more than 100. It is also not clear from any of the figures whether p-values were corrected for the multiple comparisons shown, and I didn't see it in the text. And finally, I didn't see a mention of whether statistical analysis plans and phenotype comparisons for each model were defined a-priori. A statistical methods section which addresses the false discovery issue should be added.

b. For the relations of age-gaps to mortality, heart failure and dementia, it appears that age and sex were used as covariates in the Cox models (Methods line 11107). Including chronologic age as a covariate in the prediction that was itself trained against age seems rather circular, and does not help us address the key hypothesis: that organ-specific proteins can relate to phenotype better than age alone or an organismal model. If the heart-failure survival curves in figure 2 and the hazard ratios quoted use age as a covariate in the Cox model, I think the prognostic claims about organ-specific proteins for those survival metrics are rather optimistic.

c. The minimally acceptable performance characteristics of a successful model do not appear to have been pre-defined or, aside from Tau, post-hoc related to available alternatives (e.g. measuring troponin, NTproBNP, eGFR). Some definition of "how good is good enough" should be mentioned in the new statistics section.

3) My inference is that the development of the second brain model was post-hoc, when the original approach failed. While I like the approach, it is an entirely different method of selecting candidate features: rather than using the organ-related gene expression data to do so, the features were selected for their relation to cognition. Candidate protein feature selection against a functional measure or an outcome is a well-recognized approach, but it doesn't match that of the other models and therefore seems like a separate experiment rather than an extension of the others; this method could have been used for all models (selecting phenotype-related proteins rather than organ-related proteins). If it is included for one model, then the natural question is why not try both approaches for all the models and compare them systematically?

Minor issues

1) The relation of kidney age-gap to eGFR is on the one hand a nice validation, but it raises the

question that if you adjusted the age-gap model for eGFR, is there any extra value over and above eGFR itself?

2) LASSO based models are impacted by penalization for feature number and are not fully parametric or quantitative. State of the art seems to be to make fully parametric models after LASSO. Also, elastic-nets, with its ability to regularize both feature number and the degree of correlation allowed might be viewed as more advanced if these models are meant to be final. But if these models are archetypes or proof of concept with finalization still in development this is OK, but that should be made clear.

3) The number of tiny plots and diagrams within a figure makes it hard to read – Figure 2 is the worst culprit with 19 components but figure 1 is not far behind. Not all of those plots are equally important, and especially with the omission of a decent age-gap phenotype chart as in the first major issue described above, they seem more trivial than what is currently left out.

4) The comments about plausibility and the pathway analyses would be more convincing and useful if an agnostic selection process had been used. Given that the features were selected from known organ related proteins (aside from the second brain model), they are all to some extent plausible already. I wonder if the plausibility comments and even the string network diagrams. could be sacrificed to make more space/emphasis for truly novel findings

5) The numbers of participants for the validation cohorts should be included in the text for the main section; similarly, the numbers of models developed, evaluated and successful should be clear and consistent in the text and in the figures.

6) In the main text, the “organ enriched” definition seems clear and absolute – RNA expression at least four-fold above any other organ. In the methods section there is also mention of a slightly different approach where pairs of organs could be closer. Which is it?

7) In the discussion, an argument for causality is made that includes TNN2 (troponin). Troponin is a downstream marker of damage, released from injured myocytes. It cannot be causal. Renin is also rather suspect of not usually being primary causal. Including these markers weakens the argument – and there’s nothing wrong with downstream markers!

8) In the discussion, I think the advantages of these organ-specific models vs. the more typical organismal single “clock” model could be substantially expanded to include the likely upside of making more specific inferences about health interventions (drug and/or lifestyle) that would be applicable to the individual’s organs (which are aging differently as shown in this paper) rather than guessing what to do from a single deviation from a clock model.

Referee #2 (Remarks to the Author):

The paper titled "1 Organ-specific aging signatures in the plasma proteome track health and disease" by Hamilton Oh et al out of the Tony Wyss-Coray lab presents a study of blood plasma proteome to measure organ-specific aging. Wyss-Coray laboratory is a reputable highly-regarded research group.

The authors used the human bulk RNASeq data from the different organs and SomaLogic protein array with the Covance healthy aging cohort (~353 subjects) to predict "organ-specific" aging.

Major comments:

1. Line 282-284. The discussion concludes with a statement, "This framework could serve as a blueprint to monitor organ health and aging throughout life, test the effects of rejuvenation approaches, and identify novel targets for aging interventions in humans". Unfortunately, the paper does not show any examples of target identification, nor it demonstrates a case of testing a rejuvenation or aging intervention in humans.

The authors should consider using the methodology to identify promising protein targets that may modulate the aging process and/or disease, provide a protein-level causality hypothesis, and, preferably, demonstrate the role of these genes or protein targets in aging and disease.

2. The number of subjects in the Covance aging cohort, LongGenity and Stanford ADRC cohorts is very low, and the R values for the models are very low. Some journals would not publish an aging model with r of 0.37 for the kidney, 0.47 for the heart, and 0.39 for the brain. It would be great to see a larger sample size and a much better machine learning model and better statistical significance.

Also, the participants in the Covance study are 60 years and older and do not cover the younger aging adults.

3. The subjects in the human healthy cohorts should have basic blood biochemistry data, methylation, and imaging data. Each one of these data types has several biological aging clocks associated with it. How do the proteomic and organ-specific aging biomarkers presented in the study correlate with the more established aging clocks such as DNAm, GrimAge, Levine's PheoAge, transcriptomic and proteomic clocks?

4. The statement on line 54 "Current methods to measure molecular aging in living humans have largely measured blood cell age, yet do not provide direct information on the biological age of less accessible internal organs such as the heart, kidney, or brain" is not true. There are aging models that utilize blood biochemistry data that is linked directly to organ function and is interpretable by a physician.

5. Again, since the subjects in the co-horts should have basic blood biochemistry data (e.g. glucose, albumin, AST, ALT, urea...), which can be related to specific organ functions, I would expect a Nature-level study to should provide a link between the proteomic aging models with blood biochemistry and physician-level interpretation of the models.

6. More research should be performed and reported on the role of CPLX1, CPLX2, NRXN3, and PPP3R1 in brain aging and disease. Can these proteins be good targets or point to the possible protein targets? These were previously implicated in brain aging. Any new findings from the model? Any other important features that could be reported and used for target discovery? This question is valid for other organs.

7. Feature importance diagrams for each organ and for the plasma proteomics model should be analyzed, interpreted, and presented. Can we derive any new biological hypotheses from these models?

Minor comments:

1. Only a few authors disclosed competing interests. Usually, it is a good idea to disclose all of the commercial affiliations of all authors.

2. Seminal literature on organismal and tissue-specific biological aging biomarkers is missing.

Building machine learning models on transcriptomic and proteomic data is not a novel concept, and there are even granted patents on multiple technologies in the field with priority dating back to 2017.

Referee #3 (Remarks to the Author):

Major:

They used 3 different cohorts and a powerful longitudinal approach in which disease was characterized. Close to 5000 proteins were assessed to track “organ” aging in ~1500 subject. Statistical analysis heavily depends on the healthy aging group. Where people on a structured exercise training regime excluded from all the cohorts? This could have provided different results. The most challenging aspect is that all plasma proteome is dependent on all tissues. It hard from this paper to ensure that the tissue expression (GTEx) project to classify genes as “organ enriched” is valid. After proteins expressed at least 4 times higher in one organ compared to any other organ is their target of change. I cannot judge if this is a valid approach. Thus the identification of the organ enriched plasma proteins to “non-invasively” quantify organ specific biological age is challenging by GTEx human tissue bulk-RNA seq data, but it is what is currently available. This is the major limitation, but might be minor if this is an accepted standard. In other more statistical wording is that they used plasma proteomic markers to predict organ-specific aging signatures (based on GTex) using a generic machine learning (LASSO) algorithm and this might not be sufficient. There is only one omic marker, not multiomics used in the approach. The prediction performance and potential overfitting of the model were not provided and/or fully evaluated.

Minor:

Line 91 “training” cohort...what is training? Exercise? Or simply control healthy aging cohort. Word “trained” throughout is hard to understand exactly L 127.

Line 47 change molecularly? With age

Line 51 transformative medicine bit vague

L188 and “trained”? in healthy aged individuals....

L267 reporters or causes of aging.....likely both and or reflect necrosis and or tissue and brain/blood barrier damage. Some biomarkers have beneficial roles and consider myokines (originating from muscle) or from other tissues.

L278 non-invasively ...should be less invisibly....or just state by blood sample?

Referee #4 (Remarks to the Author):

Summary

This manuscript by Oh et al. describes the creation of 19 organ-specific ageing clocks, of which 15 were able to predict chronological age in two independent testing cohorts. The authors also create an organismal ageing clock and compare its performance with those that are organ-specific. Oh et al. test the association of the age gaps derived from each of their clocks with all-cause mortality and age-related phenotypes relating to the relevant organs. For example, pancreas and kidney clock age

gaps are associated with diabetes and muscle age gap was associated with gait speed.

The authors also develop a novel algorithm to determine feature importance and use it to create a brain age clock (CognitionBrain model) that is associated with risk of dementia progression.

While the idea of creating organ-specific ageing clocks has been posited before and indeed, these authors have previously explored the performance of clocks built from different subsets of plasma proteins on a pathway specific basis (<https://doi.org/10.1111/ace.13256>). It is exciting to see for the first time how organ-specific clocks perform, particularly as the authors have assessed their performance using organ-specific follow up outcomes.

With this approach they have been able to observe the following: the low correlation of age gap measures between different clocks, that their organismal age gap is more strongly associated with all-cause mortality than organ-specific age gaps, that individuals with a history of heart attacks appear biologically older in many organs not just the heart when, in contrast individuals with hypertension showed no abnormal ageing in other major organs. Together, these results suggest that different ageing phenotypes have different patterns of organ ageing which may be captured by different sets of plasma proteins.

The organ-specific approach and the observations presented would be a step forward for the ageing clock field however, revisions need to be made to make the findings reproducible and to ensure the validity of the results and conclusions drawn before the manuscript is ready for publication.

Data & Methodology

The authors use their three independent cohorts well, having one for model training and two for testing allowing them to demonstrate the validity of their models. Additionally, the authors showed that sex is not affecting their results by finding that there was similar performance between models trained separately within each sex and those that included sex as a covariate. Oh et al. further included both chronological age and sex as covariates for all linear association analyses of model age gaps with outcome phenotypes thus reporting only the association of biological age independent of chronological age.

However, further reporting is required to make the work reproducible and demonstrate the validity of the results and conclusions drawn.

On the point of validity, there does not appear to be adjustment of p-values to account for multiple testing. Given that in several analyses there are multiple clock age gaps being tested against multiple outcomes, it would be expected to report: the number of tests performed, the method of p-value adjustment (Bonferroni, FDR, Benjamini-Hochberg etc.), the threshold for statistical significance of the adjusted p-values (5%, 1% etc.) and the number of tests that pass these criteria and are statistically significant clearly for each analysis in the main text. As at present it is not clear which of the reported associations are statistically significant.

For example, multiple times in the manuscript the authors state that “several” age gaps are

significantly associated, however they should state exactly how many tests were statistically significant and at what threshold.

On a related note, the authors highlight “strong” associations of the heart age gap with multiple heart-related phenotypes, both in this instance and in general it would be helpful to the reader to report how strong in the main text, by explicitly stating the effect size or hazard ratio and its associated standard error or confidence intervals along with a p-value. Even if effect sizes are indicated in figures, it is helpful to understand the context of the result presented. Further, reporting of effect sizes in the main text makes it easier for others to reproduce and replicate the authors results.

Additionally, it would improve transparency to include full results from all the analysis as supplementary or extended data tables. In the manuscript the authors highlight key findings for specific organs of interest in figures. Given the number of additional associations run that are not explored in the main text, including the full association results of every clock age gap against every outcome (effect sizes/hazard ratios, standard errors/confidence intervals, p-values and adjusted p-values) for each analysis would give readers the opportunity to investigate results of specific interest not only those that were significant in this analysis/highlighted by the authors. Further, it would aid repeatability and reproducibility if the authors included for each ageing clock the proteins included and their model coefficients in supplementary tables as done by many ageing clocks papers.

The authors describe the creation of their CognitionBrain model using their FIBA algorithm, report its age gap’s association with risk of dementia progression and go on to comment on its comparison with the gold standard biomarker plasma pTau-181. Clarification is needed in the first sentence of this paragraph as to exactly how this was done. As far as I understand the individuals were ranked and those in the top 25% based on CognitionBrain age gap who were also in the top 25% when ranked based on plasma pTau-181 levels were compared to those individuals who appeared in the bottom 25% of both lists? If so, I am unsure if this analysis supports the conclusions stated: “These findings indicate that CognitionBrain age gap uncovers novel biology related to AD progression not captured by current pathology-based markers.”. In order to evidence this statement, the authors could run Cox proportional hazard models for the AD outcome (2-point increase in CDR-SB within 5 years) with pTau-181 level as the predictor (along with sex and chronological age as covariates) and compare it to a model run with both pTau-181 level and CognitionBrain age gap as predictors. If the hazard ratio for the CognitionBrain age gap is significant in the combined model, that would indicate that their novel marker was capturing additional information over and above what is being provided by the current gold standard marker.

While the authors have multiple independent cohorts and large numbers of plasma proteins measured with the SomaLogic assay, there are two limitations of this data which should be caveated in the discussion.

First, compared to previous ageing clock papers the sample size of the cohort used for training (n=353) is modest, meaning that there is a greater chance of overfitting of the models (there are lower correlations between predicted age and chronological age in the testing cohorts compared to the training cohort) and them being less generalisable than models trained with larger sample sizes.

Second, is the restricted age range, due to the age range in the testing cohorts, the authors were limited to training models in individuals over 60 years of age. This does mean that the while the performance of the models presented has been demonstrated in older individuals it has not in an age range that spans the general population. It has been shown previously that ageing clocks trained in an age restricted cohort do not replicate well in wider range of ages (<https://doi.org/10.1186/s12864-020-07168-8>) and given that many ageing clocks are trained in and have been successfully replicated in cohorts spanning a much wider age range, this caveat should be highlighted in the discussion, or the authors could seek to replicate their clocks in an additional cohort with a wider age range.

Suggested Improvements

The manuscript shows association between the novel organ-specific age gaps and organ-specific outcomes as well as all-cause mortality. If the data were available, it would be nice to see if organ-specific age gaps were prognostic of (organ-relevant) disease-specific mortality.

References

In addition to the groups own previous paper which presents novel plasma proteomics ageing clocks as mentioned above, if the authors are keeping to the scope of plasma proteomics rather than specifically the SomaLogic platform, they could add the additional two citations for papers which present novel ageing clocks built with plasma proteomics from the Olink assay: Enroth et al., 2015 (<https://doi.org/10.1038/srep17282>) and Macdonald-Dunlop et al., 2022 (<https://doi.org/10.18632/aging.203847>). Both of these papers trained novel proteomic ageing clocks that predicted chronological age in both training and testing samples (as well as additional independent cohorts in the case of Macdonald-Dunlop et al.) as well as demonstrating that models built using a much smaller subset of proteins were as predictive as those containing a much larger numbers of proteins.

In the introduction, the sentence “Current methods to measure molecular ageing in living humans have largely measured blood cell age, yet do not provide direct information on the biological age of less accessible internal organs”, is the authors point that most ageing clock papers calculate one single biological age measure for the whole organism from blood-based measures – equivalent to their organismal clock – rather than separate ones for each organ system? If so, this could be clearer as it currently reads as though the point is that most clocks are made from blood-based measures which of course includes their own plasma proteomics-based clocks. The two DNA methylation-based clocks papers cited do not help clarify this, as there have been many papers that use a variety of different blood-based omics assays (including plasma proteomics) to derive measures of biological age.

Referee #1

While various attempts have been made to define “biological age”, the key problem with “clock” models - trained to mimic chronologic age - has been in the (sometimes spurious) attribution of meaning to the “age gap” - an individual’s deviation from a model’s prediction, and subsequent confusion about what kind of health intervention might help compensate for adverse deviations. The authors of this study have addressed these problems in a very interesting and novel manner.

The premise of this study is very original and creative – that the key weakness of organismal “clock” models, namely the attribution of meaning to deviations, can be overcome through the development of organ specific models made from a-priori enrichment of a wide selection of proteins, and then trained against age. The inclusion of three different data sets of reasonable size, and the design using one for model training and two for validation of age-gaps’ relation to phenotypes is also rigorous. I think that the program is ultimately publishable, but there are a number of major issues that would need to be addressed

Thank you for this thorough and balanced review. It has helped us strengthen the presentation of our results and expand the scope of the work. We believe we have now addressed all your concerns.

- We now present a comprehensive list of all statistical analyses performed for the paper, and compare all models to all phenotypes where the comparison is warranted. We report the number of statistical tests in each section and the associated FDR-adjusted q values. We also now provide all results for all tests with associated FDR control in supplementary tables, and a supplementary flowchart of statistical testing in the study.
- We’ve added an additional cohort with 3,075 individuals and several disease phenotypes. We are now using this cohort for model training, which has further strengthened all our claims and addresses concerns about sample size.
- We are now making all aging models from this study publicly available and easily accessible through a python package called *organage* (<https://github.com/hamiltonoh/organage>). Users can input sample metadata (Age and Sex) and sample protein expression (SomaScan data) to output predicted organ ages and age gaps. Coefficients for all bootstrapped aging models are accessible both through the package and listed in Supplementary Tables 6 and 16. Additional summary statistics of the models are provided in Supplementary Table 7-8,17-18.
- We provide statistics and visualizations of the changes with age and sex for all ~5,000 proteins on a shinyapp: https://twc-stanford.shinyapps.io/aging_plasma_proteome_v2/

Major

1) The most important evaluation in this study that actually addresses the thesis of whether the organ-enriched age model approach actually advances the field is the relation between age-gaps and phenotypes. But the data presented is quite incomplete: there were apparently 19 original organ-specific models created, and then a new brain model, and the organismal model. That’s 21 models. Maybe three more than that if the Cox models were additional. Age-gap to phenotype results are only shown for three of them in the main manuscript (heart, kidney and the new brain model). While 8 models are shown in extended data Figure 5, and three were declared as having insufficient data in the text, by my calculation this still leaves 7 models’ age-gap results completely missing. While 16 models’

results vs. age and mortality are shown in Figure 1, this is not the same as an age-gap and again is an inconsistent number vs. the whole. Given the importance of this component, a chart in the main text is needed that includes a) all 21 models b) how many candidate phenotypes was each model's age-gap evaluated against in total and c) of the evaluated phenotypes, how many were statistically significantly associated with age-gap when corrected for multiple comparisons d) some measure of the effect size for the subset that were significant, perhaps borrowed from Extended Data Figure 5. Without that data, the selected age gap results in the text and figures seem like anecdotes that have been picked because they worked.

We apologize for the lack of clarity on the main analysis plan regarding age gaps versus phenotypes. For the revision, we present a systematic evaluation of organ age gap associations with age-related phenotypes and provide statistics for all results in the main figures, extended data figures, and supplementary tables. In combination with this reviewer's second point about systematic and FDR-controlled evaluation of all associations tested in the manuscript, this is a substantial update to the figures and text, which we felt was warranted given these were concerns shared by multiple reviewers. The heart, kidney, and brain results we highlighted in the original manuscript remain highly significant and relevant in the context of the systematic analyses now presented. We now also present several other interesting significant associations in the revised manuscript for additional completeness. Please note that because we have changed the training cohort to address other comments from this and other reviewers, the exact values previously reported have all changed, but previously made general claims are supported. Below are the major revisions related to this reviewer concern.

1) Study design flowchart. It is difficult to also incorporate effect sizes here due to the large number of tests, but effect sizes are reported in tables and relative strength of highlighted examples is now noted in main text.

Supplementary Figure 2. Study design flow chart

- 2) We have decided to restrict our analysis to 11 organ aging models which we have higher confidence in evaluating, as well as an organismal model and conventional model, for a total of 13. The meaning of many organ-specific models, such as esophagus, adrenal, and male, remain somewhat unclear at this time because we do not have data on the phenotypic aging of these organs to test if the age gaps for these models is associated with the hypothesized phenotypic organ age. We believe these models are still interesting and valid, some surprisingly so, but they are best left for more detailed follow-up work. We have made adjustments to the text and figures to clarify the models that are tested.
- 3) Lines 96-106: “Based on this concept, we trained a bagged ensemble of least absolute shrinkage and selection operator (LASSO) aging models for 11 major organs using the mutually exclusive organ-enriched proteins we identified as inputs (Fig. 1a, Extended Data Fig. 2a-b, Supplementary Fig. 3, Supplementary Table 6-8). We chose to restrict our analyses to adipose tissue, artery, brain, heart, immune tissue, intestine, kidney, liver, lung, muscle, and pancreas because of their relatively well-understood contributions to diseases of aging and the availability of relevant age-related phenotype data in the tested cohorts. We also trained an “organismal” aging model using the 3,907 organ-nonspecific plasma proteins as inputs to compare the contribution of specific organs to an organ-shared aging signature, and a “conventional” proteomic aging model using all 4,778 proteins to compare the organ aging models to a global plasma proteomic aging signature as previously reported^{32,33}.”
- 4) All statistical tests and the number of statistically significant results after FDR correction are now clearly listed in the text, figures, and supplementary tables.
 - a) Figure 1b-e, 2i-j, and Extended Data Fig. 4, now show all 13 aging models.
 - b) Supplementary tables 9-13 contain all statistics for all statistical tests performed on these 13 models with FDR correction across each table.
 - c) Lines 109-111: “All 11 organ aging models and the organismal model significantly estimated age in all five cohorts after multiple test correction (Supplementary Fig. 3b)”
 - d) Lines 131-134 and Lines 143-146: “To assess the relationship between organ age and biological aging, we tested whether organ e-ageotypes were associated with nine age-related disease states for which we had sufficient data in at least two independent cohorts; Alzheimer's disease, atrial fibrillation, cerebrovascular disease, diabetes, heart attack, hypercholesterolemia, hypertension, obesity, and gait impairment... At the whole population level, the relationships between organ age gaps and disease showed the same trends as ageotypes, but more diseases were significantly associated with age gaps due to higher statistical power (65/117, 56%, statistically significant after multiple test correction, Extended Data Fig. 4e, Supplementary Table 10).”
 - e) Lines 173-180: “we used longitudinal follow-up among healthy participants in the LonGenity cohort to test if organ age was significantly associated with future heart failure risk (Fig. 2i, Supplementary Table 11). We found that among people with no active disease or clinically abnormal biomarkers at baseline, every 4.1 years of additional heart age (1 standard deviation) conferred an almost 2.5 fold increased risk of heart failure over a 15 year follow-up (23% increased risk per year of heart aging, Fig. 2i). Age gaps from multiple other tissues, but not the conventional aging model, also trended towards significance.”
 - f) Lines 181-184: “We next tested the associations between organ age gaps and all-cause mortality. We found that the age gaps from 10 out of 11 organs, the organismal, and the conventional model were significantly associated with future risk of all-cause mortality after multiple test correction in the LonGenity cohort over 15 years of follow-up (Fig. 2j, Supplementary Table 12).”

g) Lines 192-194 and Lines 200-202: “we tested the associations between organ age gaps and 43 clinical biochemistry and cell count markers in the t2est cohort Covance (Extended Data Fig. 5, Supplementary Fig. 4, see Supplement for additional discussion)... We found 226 out of 559 (40%) associations between organ age gaps and clinical biochemistry markers were significant after multiple testing correction (Extended Data Fig. 5c, Supplementary Table 13).”

5) The heart and kidney associations we noted in Figure 2 remain the most statistically significant associations when reporting the full unbiased assessment of disease associations. However, to reflect the expanded analysis, Figure 2 has changed.

a) Extended Data Fig. 4e now contains associations between population wide organ age gaps and 9 morbidities of aging (f). Full statistics are in Supplementary Table 10.

Extended Data Figure 4.

f, A cross-cohort meta-analysis of associations between organ age gaps versus diagnosis of 9 major age-related diseases annotated in at least 2 independent cohorts, controlling for age and sex (linear model: $AgeGap \sim Disease + Age + Sex$). Disease covariate effects and significance are shown. P-values were Benjamini Hochberg corrected. Asterisks represent q-value thresholds: *q < 0.05; **q < 0.01; ***q < 0.001. The strongest associations per disease are highlighted with black borders.

b) Visualization of original Figure 2c,d,k,i,m has now been absorbed into Figure 2a-d. Figure 2a-d show all cohorts which we have data for the four most significant associations, and give a more detailed statistical presentation of the data than the previous version using a meta-analysis framework.

Figure 2

a, Forest plot displaying results from a cross-cohort meta-analysis of the association between the kidney age gap and hypertension history, controlling for age and sex (linear model: $AgeGap \sim Disease + Age + Sex$). Per cohort and total effect sizes, 95% confidence intervals, Benjamini Hochberg total p-value (q-values) are shown. P-values were corrected based on 117 tests of organ age gap associations with major comorbidities (Extended Data Fig. 4e).

b, As in **a**, but for kidney age gap versus diabetes history.

c, As in **a**, but for heart age gap versus atrial fibrillation or pacemaker history.

d, As in **a**, but for heart age gap versus heart attack history.

c) Original Fig. 2e has been changed to current Fig. 2i and shows heart failure risk for all models. Full statistics are in Supplementary Table 11. Original Fig. 1d was moved to Figure 2j and has been updated to FDR-corrected q-values across the 13 models we focus on. Full statistics are in Supplementary Table 12.

Figure 2

i, Cox proportional hazard regression analysis in congestive heart failure risk, controlling for age and sex ($\text{HeartFailureRisk} \sim \text{AgeGap} + \text{Age} + \text{Sex}$), within 15 years in the LonGenity cohort. 27 events out of 812 individuals. Hazard ratios, 95% confidence intervals, and Benjamini Hochberg corrected p-values (q-values) for z-scored organ age gaps are shown.

j, Cox proportional hazard regression analysis in mortality risk, controlling for age and sex ($\text{MortalityRisk} \sim \text{AgeGap} + \text{Age} + \text{Sex}$), within 15 years in the LonGenity cohort. 190 events out of 903 individuals. Hazard ratios, 95% confidence intervals, and Benjamini Hochberg corrected p-values (q-values) for z-scored organ age gaps are shown.

6) In response to another major concern from this reviewer, analysis of the cognition-optimized models has been expanded to include all organs, and FDR-corrected q-values and complete statistics are now reported. This point will be expanded on further below.

2) Statistical issues:

a. I am worried about false discovery rates. There are a couple of warning flags that might simply need clarification, or alternatively might represent a lack of control, which would make this entire program a hypothesis-generating study rather than a validation study. First, it is not clear how many phenotypes were evaluated – there are 100 in Extended Data Figure 5. Given this is a nice round number, I suspect that is only the most illustrative subset of the whole number of possibilities. Any p-values for relations to age-gaps should be corrected for the total number multiple comparisons evaluated. This may well be more than 100. It is also not clear from any of the figures whether p-values were corrected for the multiple comparisons shown, and I didn't see it in the text. And finally, I didn't see a mention of whether statistical analysis plans and phenotype comparisons for each model were defined a-priori. A statistical methods section which addresses the false discovery issue should be added.

We apologize for this oversight and miscommunication. To avoid over-complicating the manuscript submission, we did select an illustrative subset of statistical tests to showcase. We also did not report FDR-corrected p-values in all cases, which was an honest oversight and not meant to obscure or artificially inflate results. We now report all tests for all models in this revised manuscript and do appropriate false discovery rate control using the Benjamini-Hochberg method. We have made this clear in the text, figures, supplementary tables, and methods, as also highlighted above. The major revisions related to this point are summarized below:

1. We have added a detailed statistical methods section specifically to address false discovery rate control.
 - a. Lines 596-645:
2. All statistical tests and the number of statistically significant results after FDR correction are now clearly listed in the text, figures, and supplementary tables.
 - a) Supplementary tables 9-13 contain all statistics for all statistical tests performed on these 13 models with FDR correction across each table.
 - b) All figures have clearly labeled q or p values showing the FDR control
 - c) All text is now clear on the total testing burden

b. For the relations of age-gaps to mortality, heart failure and dementia, it appears that age and sex were used as covariates in the Cox models (Methods line 11107). Including chronologic age as a

covariate in the prediction that was itself trained against age seems rather circular, and does not help us address the key hypothesis: that organ-specific proteins can relate to phenotype better than age alone or an organismal model. If the heart-failure survival curves in figure 2 and the hazard ratios quoted use age as a covariate in the Cox model, I think the prognostic claims about organ-specific proteins for those survival metrics are rather optimistic.

We apologize for not explaining our methodology more carefully and attribute some of the misunderstanding perhaps to the ways different sub-fields use language. The statistics we report are actually more conservative than those requested by this comment. It seems that the reviewer believes we reported a metric of combined prediction accuracy, like r^2 , from a multivariate model that includes age. But this is not the case. We reported the AgeGap covariate-specific model coefficient and associated q-value. Correcting the model for age in this way is both standard and more conservative than reporting the statistics from the univariate model.

To assess age gap associations with disease/mortality risk we tested the cox proportional hazards model: Risk \sim AgeGap + Age + Sex. The AgeGap coefficient from this model, which is the coefficient we report in the text and figures along with the q-value and 95% confidence interval, is the hazard ratio of the age gap in relation to disease risk, independent of age and sex. Put another way, this is testing the exact hypothesis we want to know about, which is whether differences in organ age gap represent differences in biological aging between two individuals of the same chronological age. This cox proportional hazard model with age adjustment is the same model used in published landmark aging clock studies¹⁻³.

Levine et. al. *Aging* 2018

Validation data for phenotypic age came from NHANES IV, and included up to 12 years of mortality follow-up for n=6,209 national representative US adults. In this population, phenotypic age is correlated with chronological age at $r=0.94$. Results from all-cause and cause-specific (competing risk) mortality predictions, **adjusting for chronological age** (Table 2), show that a one-year increase in phenotypic age is associated with a 9% increase in the risk of all-cause mortality (HR=1.09, $p=3.8E-49$), a 9% increase in the risk of mortality

Lu et. al. *Aging* 2019

Unless indicated otherwise, we used AgeAccelGrim (rather than DNAm GrimAge) in association tests of age-related conditions because age was a confounder in these analyses. For the same reason, we also used age-adjusted versions of our DNA-based surrogate markers (for smoking pack-years and the seven plasma protein levels). In general, all association tests were **adjusted for chronological age** and, when required, other confounders as well (such as sex, Methods).

Belsky et. al. *eLife* 2022

Effect-sizes for analysis of mortality in the Normative Aging Study and the Framingham Heart Study Offspring cohort.

Effect-sizes are reported as hazard ratios (HR) per standard deviation increment of the aging measures estimated from Cox proportional hazard regression. All models included **covariates for chronological age and sex.**

In theory, this should be very similar to the univariate model, because the AgeGap is calculated as the residual off an age prediction model, so it is by design independent from Age. However, to further ease this reviewer's concerns, here we provide an example where we compare the AgeGap hazard ratios between unadjusted versus

adjusted models for mortality risk. As shown below, adjusting for age and sex provides a slightly more conservative estimate of the hazard ratio, though they are overall very similar. We chose to report estimates from adjusted models for all analyses where age is confounding to be more scientifically rigorous and to conform with the age modeling field more broadly.

Reviewer Figure

a, Cox proportional hazard regression analysis in mortality risk ($MortalityRisk \sim AgeGap$), within 15 years in the LonGenity cohort. 190 events out of 903 individuals. Hazard ratios, 95% confidence intervals, and Benjamini Hochberg corrected p-values (q-values) for z-scored organ age gaps are shown.

b, Cox proportional hazard regression analysis in mortality risk, controlling for age and sex ($MortalityRisk \sim AgeGap + Age + Sex$), within 15 years in the LonGenity cohort. 190 events out of 903 individuals. Hazard ratios, 95% confidence intervals, and Benjamini Hochberg corrected p-values (q-values) for z-scored organ age gaps are shown.

c. The minimally acceptable performance characteristics of a successful model do not appear to have been pre-defined or, aside from Tau, post-hoc related to available alternatives (e.g. measuring troponin, NTproBNP, eGFR). Some definition of “how good is good enough” should be mentioned in the new statistics section.

We appreciate the suggestion, and we have added some commentary in our statistics and discussion sections, however we are open to additional feedback on this point. It is unclear if the reviewer is referring to performance on future disease prediction tasks, disease association, or some other element.

The primary focus of the study is to demonstrate that our approach to model aging of different organs using plasma proteins is able to capture variance in different organ aging outcomes not captured by other methods, and this can be used to understand aging biology in a new way. Since our models are general aging models, we do not claim that they are better predictors of specific disease outcomes, for example heart failure risk, than training a model to instead directly predict heart failure risk. Since biomarkers for every disease will have different performance goals, we have not attempted to provide performance characteristics comprehensively for all diseases.

These general aging models do, however, show what we consider to be strong performance in disease risk prediction, which underscores the important contribution of aging to disease. To provide additional context for reviewers, we benchmark against a couple models here, however, in the absence of direct comparison data, we do not claim our approach is better.

In the case of future risk of heart failure in an independent cohort, the heart age gap is a similar strength predictor to the best reported recent models designed specifically to predict heart disease risk. When applied to independent aging cohorts, the heart aging clock (HR = 2.40, CI = 1.73-3.33) performs comparably to a 9-protein model of cardiovascular outcomes risk (HR = 2.1, CI = 1.86-2.33)⁴, the adjusted Framingham model (HR = 1.7, CI = 1.53-1.97), and a multi-protein model of cardiovascular stress which includes NTproBNP and troponin (HR = 1.83, CI = 1.55-2.16)⁵, at predicting future heart disease. These statistics are all for 1 standard deviation difference in a normalized model. We show these comparisons not to make claims of superiority, as these risk models were trained and evaluated on different cohorts with different follow-up times, but to show the relevance of the heart aging model in heart disease.

In the case of mortality, the organ age gaps predict mortality in an independent cohort with comparable effect size to age gaps from methylation aging clocks which are trained specifically to predict mortality and rate of aging, such as DNAmPhenoAge (HR = 1.21)¹, DNAmGrimAge (HR = 1.31)², and DunedinPACE (HR = 1.24)³. The heart, adipose, artery, pancreas, brain, liver, muscle, and immune age gaps all predict mortality with similar hazard ratios (provided in Supplementary Table 9). These statistics are all for 1 standard deviation difference in a normalized model. Because we cannot evaluate these models in the same cohorts, claims of superiority cannot be made. However, mortality data in our independent test cohort is collected over a comparable follow-up time (ours is 15 years, DNAmPhenoAge is 12 years, DNAmGrimAge is 13.7 years, DuendinPace is 14 years) in an American cohort of similar demographic character.

We have added a statement to the discussion section to emphasize these points. Lines 360-364: “Although we were unable to perform direct comparisons, our models predict mortality with comparable effect sizes to models trained specifically to predict mortality and heart disease in independent cohorts^{12,15,39,67,68}. We demonstrated that our approach added increased value to established biomarkers of Alzheimer’s disease, and we expect that multimodal aging and disease prediction models may have similar impacts in other diseases”

3) My inference is that the development of the second brain model was post-hoc, when the original approach failed. While I like the approach, it is an entirely different method of selecting candidate features: rather than using the organ-related gene expression data to do so, the features were selected for their relation to cognition. Candidate protein feature selection against a functional measure or an outcome is a well-recognized approach, but it doesn’t match that of the other models and therefore seems like a separate experiment rather than an extension of the others; this method could have been used for all models (selecting phenotype-related proteins rather than organ-related proteins). If it is included for one model, then the natural question is why not try both approaches for all the models and compare them systematically?

Thank you for the balanced critique. It is true that the method was applied post-hoc because we were specifically interested in better understanding brain aging and dementia. It is a reasonable critique that it is unfair to compare this model to the previous models since the underlying method is different. We have now expanded the scope of the analysis to apply this approach to all organs and compare the results systematically. This is a substantial

new set of analyses which has resulted in many new additional interesting results, which we have expanded into an additional figure 4 and multiple new extended data figures. The brain aging model remains the most significant/largest effect model in the context of this larger analysis. We outline the expanded approach and new results below:

1. New training cohort

- a. We leveraged the additional cohort we acquired, the Knight ADRC, to do feature selection for these second-generation models on dementia status, our primary outcome of interest, as opposed to the composite cognition measure. We initially opted to use composite cognition in the LonGenity cohort for feature selection because we only had one cohort with dementia participants, the SADRC, so we did not have a fully independent test cohort if we used dementia status. We have changed the text in Lines 223-226 to reflect this. All major findings using this new approach remain consistent with the cognition-optimized brain model in the original manuscript, but we believe the strength of our claims is increased now.
 - b. Due to the larger sample size, the use of the KADRC cohort as the new training cohort actually led to an association between the first-generation brain aging model and Alzheimer's disease (now shown in Extended Data Fig. 4e), which is further strengthened through our FIBA feature selection approach. We have thus changed the textual motivation for the FIBA approach to more accurately reflect our thinking in Lines 216-223.
2. As requested, we have expanded this approach to all organs, which now constitutes Figure 4, Extended Data Figures 7-10, and Supplementary Tables 14-19,22. Textually, this encompasses the new section we have added, from Lines 291-344. To summarize the major discoveries, we found the following.
- a. CognitionBrain, CognitionOrganismal, CognitionPancreas, and CognitionArtery age gaps were significantly associated with the presence of/a diagnosis of Alzheimer's disease in both the KADRC and SADRC. The CognitionBrain model had the largest effect size in the larger KADRC cohort, followed by CognitionOrganismal. Full effect sizes and statistics for optimized models are in Supplementary Table 19 and Extended Data Figure 8.
 - b. The CognitionArtery and CognitionOrganismal models are specifically associated with future conversion from cognitively normal to mild cognitive impairment (MCI) in healthy participants.
 - c. We discuss multiple biological processes and proteins which we believe underlie these relationships. The data implicate vascular aging, specifically blood brain barrier dysregulation, vascular calcification, and extracellular matrix changes as important biological processes underlying cognitive decline and dementia onset very early in the temporal sequence of brain aging and Alzheimer's disease.

Minor

1) The relation of kidney age-gap to eGFR is on the one hand a nice validation, but it raises the question that if you adjusted the age-gap model for eGFR, is there any extra value over and above eGFR itself?

We have addressed this by adjusting the kidney age gap for eGFR in the LonGenity cohort, the only independent test cohort where we have both eGFR and the measures of diabetes status and hypertension status. We corrected the kidney age gap for eGFR using a linear model in R. We calculated eGFR from the CKD-EPI equation using serum creatinine measured in the cohort. The eGFR-adjusted kidney age gap remained significantly associated with both diabetes status and hypertension status, as shown below. We have added this as Supplementary Fig. 5 and made a note in the text on Lines 207-211.

Kidney age gap versus disease with EGFR adjustment in the LonGenity cohort

Supplementary Figure 5

- a, Kidney age gap associations with hypertension, adjusted for EGFR in the LonGenity cohort.
- b, Kidney age gap associations with diabetes, adjusted for EGFR in the LonGenity cohort.

2) LASSO based models are impacted by penalization for feature number and are not fully parametric or quantitative. State of the art seems to be to make fully parametric models after LASSO. Also, elastic-nets, with its ability to regularize both feature number and the degree of correlation allowed might be viewed as more advanced if these models are meant to be final. But if these models are archetypes or proof of concept with finalization still in development this is OK, but that should be made clear.

We are grateful for the reviewer's attention to modeling decisions, and we agree that there could be additional performance improvements to the models by extensively testing alternative algorithms and training schemes. However, we believe the paper is better focused on the biology discovered, so the models presented are meant as archetypes and to illustrate the underlying rationale and robustness of the methods described in the paper. We have made this clear in the text and discuss limitations of the current approach in the discussion section, Lines 388-389.

We also note here that Elastic Net models perform very similar to our bagged LASSO model approach, as shown here summarized across all models. Additionally, for the majority of organ aging models which start from a small set of possible features (less than 100), nearly identical features are selected because a LASSO-like mixing coefficient is optimal. Lastly, LASSO is both fully parametric and fully quantitative. Unlike two-stage methylation aging clocks such as DNAmPhenoAge and DNAmGrimAge, which are first trained on mortality data using a nonparametric cox model for the purposes of feature selection, and then are re-parameterized as Gompertz models our approach is simpler and just trains on chronological age.

Reviewer Figure. Lasso vs ElasticNet age gap correlation for all models and all samples.

3) The number of tiny plots and diagrams within a figure makes it hard to read – Figure 2 is the worst culprit with 19 components but figure 1 is not far behind. Not all of those plots are equally important, and especially with the omission of a decent age-gap phenotype chart as in the first major issue described above, they seem more trivial than what is currently left out.

Thank you for the comments. We have re-designed figure 2 with less panels and a stronger focus on the truly novel results.

4) The comments about plausibility and the pathway analyses would be more convincing and useful if an agnostic selection process had been used. Given that the features were selected from known organ related proteins (aside from the second brain model), they are all to some extent plausible already. I wonder if the plausibility comments and even the string network diagrams. could be sacrificed to make more space/emphasis for truly novel findings

We have removed the string diagrams in favor of more analysis on which proteins in our models have the strongest relationships to different traits. We are unsure which specific plausibility comments the reviewer refers to, but we are open to adjusting the text here. We will note though, that it was not previously obvious before our study that organ-specific proteins can be markers of organ aging, though we agree they are all plausibly involved the biology of their respective tissues which is part of the appeal of our approach.

5) The numbers of participants for the validation cohorts should be included in the text for the main section; similarly, the numbers of models developed, evaluated and successful should be clear and consistent in the text and in the figures.

We have fully addressed this as noted for other comments from this reviewer. Main Figure 1a shows the sample size of all cohorts. This has been added to the text as well at line 146.

6) In the main text, the “organ enriched” definition seems clear and absolute – RNA expression at least four-fold above any other organ. In the methods section there is also mention of a slightly different approach where pairs of organs could be closer. Which is it?

We apologize for the confusion. The “organ enriched” definition is clear and absolute – RNA expression at least four-fold above any other organ. We have adjusted the methods to be consistent with the text, lines 80-83. Methods section Lines 542-552.

7) In the discussion, an argument for causality is made that includes TNN2 (troponin). Troponin is a downstream marker of damage, released from injured myocytes. It cannot be causal. Renin is also rather suspect of not usually being primary causal. Including these markers weakens the argument – and there’s nothing wrong with downstream markers!

We see the reviewer’s point and have removed these two proteins from this statement now.

8) In the discussion, I think the advantages of these organ-specific models vs. the more typical organismal single “clock” model could be substantially expanded to include the likely upside of making more specific inferences about health interventions (drug and/or lifestyle) that would be applicable to the individual’s organs (which are aging differently as shown in this paper) rather than guessing what to do from a single deviation from a clock model.

Thank you for the encouragement. We have expanded this slightly in the discussion section, lines 349-351.

Referee #2

The paper titled "1 Organ-specific aging signatures in the plasma proteome track health and disease" by Hamilton Oh et al out of the Tony Wyss-Coray lab presents a study of blood plasma proteome to measure organ-specific aging. Wyss-Coray laboratory is a reputable highly-regarded research group. The authors used the human bulk RNASeq data from the different organs and SomaLogic protein array with the Covance healthy aging cohort (~353 subjects) to predict "organ-specific" aging.

We thank this reviewer for their thorough review. We believe the emphasis on biological interpretation and comparison to other benchmarks has greatly improved our manuscript and increased our own understanding of this approach.

Major

1. Line 282-284. The discussion concludes with a statement, "This framework could serve as a blueprint to monitor organ health and aging throughout life, test the effects of rejuvenation approaches, and identify novel targets for aging interventions in humans". Unfortunately, the paper does not show any examples of target identification, nor it demonstrates a case of testing a rejuvenation or aging intervention in humans. The authors should consider using the methodology to identify promising protein targets that may modulate the aging process and/or disease, provide a protein-level causality hypothesis, and, preferably, demonstrate the role of these genes or protein targets in aging and disease.

We apologize if our concluding statement came across as too strong and we did not mean to suggest that the manuscript provided applications of the framework already. The sentence was meant to provide an outlook for the future and encourage our peers to test it. Nevertheless, we have now toned down the concluding sentence as follows:

1. "Altogether, we show that large-scale plasma proteomics and machine learning can be leveraged to non-invasively measure organ health and aging in living people. We show that biologically motivated modeling, in which we use sets of organ-specific proteins and the FIBA algorithm to further subset to physiological age-related proteins, enables deconvolution of the different rates of aging within an individual and measurement of aging at organ-level resolution."

We also discuss additional interesting biological observations based on our new methodology, but our study is unable to infer causality for any aging model. In this regard it is similar to other published "aging clocks" which can provide insight into the biology of aging and help in the generation of new hypothesis but are not able to infer causality as we recently discussed in a review article on the subject⁶. Other changes to the text and figures related to this claim are summarized below.

- a. When considering the biology of the proteins in the heart aging model, we note the prominent role of MYL7, a component of cardiac myosin and its recent link to heart disease in the literature, lines 170-172: "MYL7 is expressed by atrial cardiomyocytes and has recently become a promising target for hypertrophic cardiomyopathy³⁸, suggesting that this could be a repurposing target for heart aging more generally."
- b. We have added a new section, based on this and other reviewer comments, in which we propose aging mechanisms and proteins involved in the earliest phases of cognitive decline and future dementia onset. Our data are consistent with a model in which changes in the brain vasculature and extracellular matrix, particularly vascular calcification and alterations to pericyte function and related blood-brain-barrier integrity,

may be early contributors to cognitive decline based on the temporal changes observed during aging. Relevant section in lines 291-344.

2. The number of subjects in the Covance aging cohort, LongGenity and Stanford ADRC cohorts is very low, and the R values for the models are very low. Some journals would not publish an aging model with r of 0.37 for the kidney, 0.47 for the heart, and 0.39 for the brain. It would be great to see a larger sample size and a much better machine learning model and better statistical significance. Also, the participants in the Covance study are 60 years and older and do not cover the younger aging adults.

We will address these concerns in order.

A. "The number of subjects in the Covance aging cohort, LongGenity and Stanford ADRC cohorts is very low"

To address this concern, we have added two new cohorts to our study: the Knight Alzheimer's Disease Research Center (KADRC) cohort (n=3,075) and the Stanford Aging and Memory Study (SAMS) cohort (n=192). This means we have increased our number of independent cohorts from three to five and have increased our total sample size from 1,727 to 5,678, making our revised study the largest plasma proteomic aging clock study to date⁷⁻¹¹, and the largest plasma proteomics study of clinically assessed Alzheimer's disease participants to our knowledge^{12,13}.

Figure 1

a, Schematic illustrating the study design of estimating and validating organ-specific biological age (See Methods).

We find that this cohort number and sample size is sufficient to discover highly significant and reproducible associations between age gaps and chronic diseases in individual cohorts and cross-cohort meta-analyses. This is now reflected explicitly in the text.

1. Lines 131-134 and Lines 143-146: "To assess the relationship between organ age and biological aging, we tested whether organ e-ageotypes were associated with nine age-related disease states for which we had sufficient data in at least two independent cohorts; Alzheimer's disease, atrial fibrillation, cerebrovascular disease, diabetes, heart attack, hypercholesterolemia, hypertension, obesity, and gait impairment... At the whole population level, the relationships between organ age gaps and disease showed the same trends as ageotypes, but more diseases were significantly associated with age gaps due to higher statistical power (65/117, 56%, statistically significant after multiple test correction, Extended Data Fig. 4e, Supplementary Table 10)."

We have also changed the visualization and statistical reporting of results to meta-analysis forest plots, which demonstrate the high reproducibility of our results across multiple independent cohorts. Full results of this analysis are now in Supplemental Table 10.

Figure 2

a, Forest plot displaying results from a cross-cohort meta-analysis of the association between the kidney age gap and hypertension history, controlling for age and sex (linear model: $AgeGap \sim Disease + Age + Sex$). Per cohort and total effect sizes, 95% confidence intervals, Benjamini Hochberg total p-value (q -values) are shown. P-values were corrected based on 117 tests of organ age gap associations with major comorbidities (Extended Data Fig. 4e).

b, As in **a**, but for kidney age gap versus diabetes history.

c, As in **a**, but for heart age gap versus atrial fibrillation or pacemaker history.

As sample size concerns are most often concerns about reproducibility/generalizability, we hope that our substantial increase in sample size and cross-cohort statistical analyses in the revision, together, sufficiently demonstrate that our results are reproducible.

Naturally, our approach will become even more powerful if applied to larger-scale proteomics resources that are being generated, such as the UK biobank, and we have now made a note of this in the discussion section, Lines 365-367: “We present one of the largest studies of plasma proteome aging to date, but as larger plasma proteomics resources are being generated⁶⁹⁻⁷² the power of this approach will further increase.”

B. “the R values for the models are very low. Some journals would not publish an aging model with r of 0.37 for the kidney, 0.47 for the heart, and 0.39 for the brain. It would be great to see a larger sample size and a much better machine learning model and better statistical significance.”

This is a very interesting comment the field has increasingly grappled with. What is an “ideal” prediction accuracy in a model that is designed to estimate biological age or uncover information about biological aging.

While there clearly must be robust statistically significant correlation between a molecular aging model and chronological age, the optimal correlation strength is unknown (and very likely differs for different systems). Consider that if an aging model predicted chronological age perfectly, then it would *by definition* contain no extra information about biological aging beyond chronological age itself and would thus be useless as a biomarker of biological aging. There is a direct statistical trade-off between chronological age prediction accuracy and the ability to detect biological differences in aging between people of the same chronological age, which has been repeatedly noted in the field of aging modeling¹⁴ as a core challenge and contradiction. In fact, the field is rapidly moving away from this way of thinking as most new research uses methods which

are more informed by the biological factors we care about more than chronological age itself. Two of the most recent methylation aging clocks, DunedinPoAm and DunedinPACE, which are among the best-performing methylation aging clocks for predicting aging phenotypes, have correlations with chronological age of just $r = 0.11$ ¹⁵ and $r = 0.32$ ³, respectively. Having recently written a comprehensive review on the field of aging clocks⁶, we noted that the majority of recent papers is moving away from chronological age prediction accuracy as a justification for the scientific interest or utility of a molecular aging model. Instead, the focus in the field is on how well and how generally such models can predict aging phenotypes.

To evaluate our results in this realm, in addition to the robust disease associations we see, we can use the hazard ratio to predict e.g., mortality as an independent measure of the strength of various models. We find multiple organ age gaps predict mortality in an independent cohort as good or better than the age gaps from methylation aging clocks which are trained specifically to predict mortality and rate of aging, such as DNAmPhenoAge (HR = 1.21)¹, DNAmGrimAge (HR = 1.31)², and DunedinPACE (HR = 1.24)³. The heart, adipose, artery, pancreas, brain, liver, muscle, and immune age gaps all predict mortality with larger hazard ratios (provided in Supplementary Table 12). These statistics are all for 1 standard deviation difference in a normalized model and are directly comparable. We have chosen not to focus on this in the manuscript, because this is not a direct comparison and we are unable to do a direct comparison because we do not have any cohorts with proteomics and methylation data. However, mortality data in our independent test cohort is collected over a comparable follow-up time (ours is 15 years, DNAmPhenoAge is 12 years, DNAmGrimAge is 13.7 years, DuendinPace is 14 years) in an American cohort of similar demographic character.

Furthermore, in our analyses associating organ age gaps with age-related phenotypes, organ aging models with moderate chronological age correlation generally outperform the control and organismal models which have high age correlation. This is the case for mortality, all 9 diseases we tested, future heart failure, 39/42 clinical biochemistry markers, and future Alzheimer's disease risk. Here we show some examples where we compare the kidney model (average $r = 0.33$) to the conventional model (average test $r = 0.91$). Clearly, the kidney aging model has a much stronger association with kidney disease than the conventional model, despite having a lower correlation with chronological age.

Reviewer Figure

a, Forest plot displaying results from a cross-cohort meta-analysis of the association between the kidney age gap and diabetes history, controlling for age and sex (linear model: $\text{AgeGap} \sim \text{Disease} + \text{Age} + \text{Sex}$). Per cohort and total effect sizes, 95% confidence intervals, Benjamini Hochberg total p-value (q -values) are shown. P-values were corrected based on 117 tests of organ age gap associations with major comorbidities (Extended Data Fig. 4e).

b, As in **a**, but for conventional age gap versus diabetes history.

We do agree with the reviewer that the machine learning could be further refined, and it's possible, that doing so may also lead to a higher chronological age prediction accuracy. Reviewer 1 has also requested that we acknowledge that these models are in some sense prototypes and they could possibly be further refined by using different algorithms, training schemes, or data. Given that many proteins change with age in nonlinear ways, we think a particularly fruitful area for future work will be nonlinear modeling using tree-based methods or neural networks. We have now addressed this in the discussion section, Lines 388-389: "More sophisticated non-linear machine learning methods such as neural networks or random forests may further improve the accuracy and generalizability of this approach in the future".

C. "the participants in the Covance study are 60 years and older and do not cover the younger aging adults". This is a valid concern which we have attempted to address with substantial additional training data. It is also a complex issue that warrants a detailed response with additional literature context, which we now provide here and in the revised manuscript.

a. First and foremost, in our revised study design we use the newly added KADRC cohort as our training cohort, which expands the sample size and age range of our training (age 27-104). We now use the entire Covance cohort (age 18-90) as one of the test cohorts (Fig. 1a, Supplementary Fig. 3b), and we can evaluate the degree to which our models may be "overfit" to aged individuals. Overall, we found that our models are overfit to some degree in all test cohorts, but this was not driven by the different age ranges in the cohorts. Some degree of overfitting is expected given technical variation in omics assays. In many cases, Covance showed the best fit (ie. Organismal, Liver, Immune, etc) – suggesting that the models selected proteins that change consistently across the lifespan, making them reproducible in both younger and older individuals.

Supplemental Figure 3

b, Correlations between predicted vs chronological age in healthy individuals in the training (Knight-ADRC) and test (Covance, LonGenity, Stanford-ADRC, SAMS) cohorts for all aging models. All aging models significantly estimated age across five independent cohorts.

b. However, there is still a larger representation of older adults than younger adults in our training data, as the mean age in the KADRC is 75, so we have added an additional statement in the discussion urging caution when applying our models to younger populations, lines 386-387. "While our current models serve as a proof of principle for this approach, since they are trained and evaluated largely on older adults caution should be used when applying them to young people."

There are clear tradeoffs between optimizing the aging measure for young adults or the elderly. Given the test cohorts we analyze in the manuscript are all older cohorts, as are most human datasets for diseases of aging, we believe it is important for the training cohort to match the test distributions better as this allows for more statistically sound and biologically correct inference.

There are many biomarkers of health which have a nonlinear relationship to aging outcomes, and in the elderly many relationships between biomarkers and health/mortality/frailty reverse direction. We believe

this is very important to understand, especially when evaluating the relationship between training and test cohorts, and noticed that this is largely not discussed or accounted for in models of molecular aging. We have now used our analysis of the clinical biochemistry data in the Covance cohort to illustrate the nuance of this point. We believe this is an important contribution, and it supports our use of a primarily older training cohort to evaluate disease outcomes in older test cohorts which is the primary focus of the manuscript. We emphasize this in the newly added Supplemental Text, and in Supplementary Fig. 4, where we show U-shaped trait relationships for 3 important traits in the Covance cohort with age that are discussed in the text.

Supplemental Figure 4

b, U-shaped relationship between age and certain traits, including diastolic blood pressure, BMI, and alanine transaminase are shown.

Supplemental text lines 38-62:

“There are many biomarkers of health which have a nonlinear relationship to aging outcomes, and in the elderly many relationships between biomarkers and health/mortality/frailty reverse direction compared to young and middle-aged adults. The distribution and mean age of the population that an aging model is trained on will thus impact associations with traits. This is not frequently discussed or accounted for in models of molecular aging.

Such a case is illustrated by diastolic blood pressure, where the strongest association was with heart aging (adjusted Pearson $r=-0.18$, $q=2.62e-10$). Nine organ age gaps (adipose, brain, control, heart, intestine, kidney, liver, muscle, organismal, pancreas) were significantly associated with decreases in diastolic blood pressure, while the opposite association was seen with the PhenoAge age gap (Supplementary Fig. 5a, Supplementary Table 13). Diastolic blood pressure was one of many traits with a U-shaped relationship to aging outcomes (Supplementary Fig. 5b). While high blood pressure in young and middle-aged adults is indicative of cardiometabolic dysfunction, in the elderly low blood pressure is common and more strongly associated with mortality and frailty³⁻⁵, though high blood pressure is also detrimental⁶. The differences between PhenoAge and the organ age models could be due to differences in the age distribution of the underlying training cohorts for the models. Our models were trained in the KADRC, which has a greater proportion of elderly individuals, while PhenoAge was trained in NHANES, which has a greater proportion of young individuals.

This kind of U-shaped relationship with age and aging outcomes is quite common and is also seen with BMI⁷. Prospective studies in older adults have shown that while obesity slightly increases mortality and cardiovascular disease risk, the highest risk groups are those with a BMI under 23. Interestingly, the intestine and pancreas age gaps show a negative association with

BMI and obesity but a positive association with mortality risk, while the kidney age gap shows a positive association with BMI, suggesting that the full picture of organ health in aging and disease may be more complex than currently understood.”

3. The subjects in the human healthy cohorts should have basic blood biochemistry data, methylation, and imaging data. Each one of these data types has several biological aging clocks associated with it. How do the proteomic and organ-specific aging biomarkers presented in the study correlate with the more established aging clocks such as DNAm, GrimAge, Levine's PhenoAge, transcriptomic and proteomic clocks?

This is a great suggestion and we have now used any available biochemistry data in our current cohorts to model phenotypic aging. The subjects in the Covance cohort have proteomics and biochemistry, while the subjects in the SADRC and SAMS have proteomics and neuroimaging. We have therefore compared the organ-specific clocks to previous proteomics, Levine's PhenoAge, and MRI-based brain age. Overall, we find statistically significant but small correlations between the organ aging models and these other clocks. We have also added a section to the discussion highlighting the need to compare these new models to additional approaches, such as DNA methylation aging clocks. Results and changes to the manuscript are summarized below:

A. We have now made our comparison to previous proteomic clock methods explicit by adding a model labeled “conventional” to Figure 1 and Figure 2. The conventional model is trained with a LASSO model and without using additional feature selection, the same method used to train previous proteomic aging clocks^{8,10}. Since the version of the SomaScan assay is different, this is a more rigorous test than directly applying previously developed clocks which perform slightly worse on the new assay version. As highlighted in other comments, we find that the control model is the best predictor of chronological age, but is not the best predictor of any other aging phenotype. We now distinguish the Conventional model from the “Organismal” model, which is trained using only organ non-specific proteins. We have changed all figures to include these models.

Figure 1

b, Comparison of conventional age gap versus organ age gap profile. Examples are shown where three individuals with the same conventional age gap, have vastly different distributions of organ age gaps.

c, Pairwise correlation of organ age gaps in all cohorts. Correlations were generally weak, given that all models were trained to predict age, suggesting that though organ aging models capture distinct information. Distribution of all pairwise correlations is shown in inset histogram. The conventional age gap was highly correlated with the organismal age gap ($r=0.98$).

B. Comparison to PhenoAge is covered in Extended Data Figure 5a-b, in the text from Lines 194-199, and in the supplemental text lines 3-62. The strongest correlation was with the immune age gap, followed by the conventional, organismal, kidney, and liver. These correlations make sense, as 6/9 clinical markers that underlie PhenoAge are liver (albumin, CRP, ALP), kidney (creatinine), and immune (% lymphocytes, white blood cell count, CRP) derived or related. Despite the statistically significant correlations, the majority of variance in all organ age gaps is unexplained by PhenoAge, and some organs, such as the heart aging model, were not at all correlated with PhenoAge but were predictive of all-cause mortality and aging phenotypes (Fig. 2).

Extended Data Figure 5

a, Phenotypic Age (PhenoAge), a gold-standard clinical marker based aging model, was calculated based on the 10 requisite clinical markers in the Covance cohort (n=1,026). PhenoAge-based age prediction is shown.

b, The PhenoAge age gap was calculated and correlated with plasma proteomic organ aging model age gaps. Pairwise correlations are shown.

We also find that associations between clinical biochemistry markers, PhenoAge, and organ age are complementary and interpretable within the context of the model development and scientific literature on aging biomarkers. This point is expanded upon in the next reviewer comment.

C. We used matched brain MRI and plasma proteomic data from n=541 samples in SAMS and SADRC to compare our plasma proteomics-based brain age to two MRI brain aging clocks. Based on its established publication record, we started with the BARACUS model¹⁶, a linear support vector machine based aging clock trained on brain MRI-based volumetric data from 1,166 cognitively normal individuals aged 20-80. However, when assessing predicted versus chronological age correlation, we noticed an odd technical artifact: the predicted age had a ceiling near 75, even for individuals with chronological age above 90. Looking more closely at the original publication, we found the same issue of an upper ceiling, and also a lower ceiling, to predicted age. This leads us to believe that the BARACUS algorithm cannot accommodate all ages in our cohort.

Due to this technical limitation of BARACUS, we also assessed brainageR¹⁷, a Gaussian Processes based aging clock trained on brain MRI-based volumetric data from n=3,377 cognitively healthy individuals aged 18-92, and which has shown better performance than BARACUS in other studies¹⁸. The CognitionBrain age gap was positively correlated with the brainageR age gap ($r=0.16$, $p=7.51e-4$) (Extended Data Fig. 6h), but not as strongly as the correlation between CognitionBrain age gap and individual brain volumes (ie. hippocampus: adjusted $r=-0.21$, $q=1.36e-4$). This is now reflected in the text from lines 244-248 and Supplemental text lines 64-93.

Supplementary Figure 6

a, CognitionBrain age gaps were associated with brain MRI volume in the Stanford-ADRC and SAMS cohorts (n=469).

b, The publicly available brain MRI based aging model, BARACUS Brain-Age, was tested in the Stanford-ADRC and SAMS cohorts. The age prediction image from the original publication Liem et. al. 2017 is shown. An age prediction ceiling can be observed.

c, The age prediction using BARACUS in the Stanford-ADRC and SAMS cohorts is shown. An age prediction ceiling can be observed in both the original publication and when tested in Stanford cohorts.

d, The publicly available brain MRI based aging model, brainageR, was tested in the Stanford-ADRC and SAMS cohorts. The age prediction image from the github is shown. No age prediction ceiling can be observed.

e, The age prediction using brainageR in the Stanford-ADRC and SAMS cohorts is shown. No age prediction ceiling can be observed.

f, The correlation between the CognitionBrain age gap and brainageR age gap was assessed and is shown.

4. The statement on line 54 "Current methods to measure molecular aging in living humans have largely measured blood cell age, yet do not provide direct information on the biological age of less accessible internal organs such as the heart, kidney, or brain" is not true. There are aging models that utilize blood biochemistry data that is linked directly to organ function and is interpretable by a physician.

Figure 1

d, Identification of extreme agers, defined by a 2-standard deviation increase or decrease in at least one age gap. A kidney ager, heart ager, and multi-organ ager are shown as examples.

We apologize for this omission and have adjusted the text on line 64-65 in our manuscript to acknowledge aging models which have been trained on clinical chemistry data, such as Phenotypic Age¹⁹, Pace of Aging²⁰, and the deep blood chemistry aging clock²¹ from the Zhavoronkov group. What we wished to emphasize was that like other models, these predict a single composite aging score for the whole body. To the extent they contain organ-specific information, this is limited largely to inflammation, liver, and metabolic markers. An individual could have a high age gap for multiple different reasons that impact these more nonspecific markers, and in the absence of one or two particular underlying extreme values, it would be unclear what organs were driving the age prediction even if a feature importance plot was provided. This is now highlighted more clearly in revised Figure 1d-e, where we illustrate "extreme organ agers", those individuals who show large age gaps for one organ compared to the rest of the body. It would also not be a complete picture of organ aging, as markers from many tissues, such as the brain, are absent from these models. Other groups have developed imaging-based aging clocks for specific organs, mainly for the brain, which we also did not acknowledge in the first version, but now acknowledge in the revised version in lines 67-69. We believe our method has many advantages over imaging-based approaches and is complementary to them. Hopefully, this is clearer in the text now and is not

dismissing other important contributions in the field, which was not our intention. The new text appears at Lines 61-69:

“While many methods to measure molecular aging in humans have been developed^{10–18}, most of them provide just a single measure of aging for the whole body. This is difficult to interpret given the complexity of human aging trajectories, and no single method so far predicts aging outcomes in all organs. Some recent methods have used clinical chemistry markers which include some markers of organ function^{10,16,19}. However, these methods still generate a single composite score for the whole body and contain many markers with low organ specificity, making them difficult to interpret for organ-specific aging. Methods to measure brain aging have used MRI-based brain volume and functional connectivity measurements, but they are costly, time-consuming, and do not provide molecular insights.”

5. Again, since the subjects in the cohorts should have basic blood biochemistry data (e.g. glucose, albumin, AST, ALT, urea...), which can be related to specific organ functions, I would expect a Nature-level study to should provide a link between the proteomic aging models with blood biochemistry and physician-level interpretation of the models.

We have addressed this comment with additional analyses of clinical biochemistry in our test cohort Covance, which is the only cohort in which we have both clinical biochemistry and proteomics. This analysis is now included in Extended Data Fig. 5, Supplementary Table 13, Supplementary Fig. 4. It appears from Lines 191-209:

“Finally, to better understand the relationship between organ age and additional markers of health and disease, we tested the associations between organ age gaps and 43 clinical biochemistry and cell count markers in the test cohort Covance (Extended Data Fig. 5, Supplementary Fig. 4, see Supplement for additional discussion). We also used these markers to calculate Phenotypic age^{12,19} (PhenoAge), a clinical biochemistry-based aging clock which predicts mortality and morbidity risk, for all participants in Covance (Extended Data Fig. 5a). We found that the PhenoAge age gap was significantly correlated with multiple organ age gaps, but only a small portion of the variance in any model was explained by another (Extended Data Fig. 5b).

We found 226 out of 559 (40%) associations between organ age gaps and clinical biochemistry markers were significant after multiple testing correction (Extended Data Fig. 5c, Supplementary Table 13). The strongest associations included associations between liver age gap and blood AST:ALT ratio, a clinical marker of liver health and function that is known to change with age (adjusted Pearson $r=0.25$, $q=6.13e-17$), and between kidney age gap and serum creatinine, the standard clinical marker of kidney function (adjusted Pearson $r=0.23$, $q=1.65e-16$). While these results are highly significant, they only partially explain the relationship between organ age gaps and disease phenotypes. Even after correcting for estimated glomerular filtration rate (eGFR), the kidney age gap is still significantly associated with hypertension and diabetes (Supplementary Fig. 5).”

a. It is further expanded upon in the Supplemental Text to address the reviewer comment, lines 4-62:

“While a full analysis of all clinical biochemistry markers is challenging, there are a number of additional interesting relationships in the data.

- BUN: Kidney, adipose, brain, immune, and muscle age gaps are significantly positively associated with BUN, artery age gap is significantly negatively associated. The strongest association is with the kidney age gap. While BUN is not specific, it is often considered a marker of kidney function clinically.
- AST: Kidney, heart, and artery age gaps are positively significantly associated with AST, while brain is significantly negatively associated. AST variation within the normal range is difficult to interpret clinically. Abnormally high AST is often a sign of liver or heart disease, and moderately high AST is most often noted as a sign of elevated cardiovascular risk in middle aged and elderly populations.

- ALT: The brain, control, liver, intestine, kidney, organismal, and pancreas age gaps are significantly negatively associated with ALT, while the kidney age gap is significantly positively associated with ALT. PhenoAge gap is positive but not significant. As discussed in the text, this is yet another U-shaped aging biomarker. Low ALT in the elderly is associated with increased frailty and reduced survival and has been previously suggested as a biomarker of aging¹. Abnormally high ALT can be a marker of acute liver damage, although it is also produced by other tissues and is not specific.
- Albumin: The immune, heart, liver, organismal, control, and PhenoAge gaps are significantly negatively associated with albumin levels. The strongest association is with the liver age gap. Albumin is produced by the liver, although it is not detected by the SomaScan assay so it is not a protein in the liver aging model. Clinically, lower albumin could be considered as a sign of worse health, and it can be low in a number of liver, kidney, and digestive diseases as well as in malnutrition/undernutrition.
- Plasma glucose is significantly positively associated with PhenoAge age gap and kidney age gap, while intestine and liver age gap are significantly negatively associated. The strongest association is with PhenoAge, which is unsurprising since plasma glucose is the highest weighted input biomarker in the PhenoAge model. Both kidney and intestine age gap are positively associated with diabetes incidence but have differential associations with plasma glucose. This further supports the hypothesis that different organ models could be measuring different aspects of aging, in this case metabolic aging. Insulin resistance, glucose response, and glucose levels are all known to degrade with age, but insulin levels and glucose response have been noted to change more dramatically than fasting blood glucose level².

There are many biomarkers of health which have a nonlinear relationship to aging outcomes, and in the elderly many relationships between biomarkers and health/mortality/frailty reverse direction compared to young and middle-aged adults. The distribution and mean age of the population that an aging model is trained on will thus impact associations with traits. This is not frequently discussed or accounted for in models of molecular aging.

Such a case is illustrated by diastolic blood pressure, where the strongest association was with heart aging (adjusted Pearson $r=-0.16$, $q=2.38e-8$). Nine organ age gaps (adipose, brain, control, heart, intestine, kidney, liver, muscle, organismal, pancreas) were significantly associated with decreases in diastolic blood pressure, while the opposite association was seen with the PhenoAge age gap (Supplementary Fig. 5a, Supplementary Table 13). Diastolic blood pressure was one of many traits with a U-shaped relationship to aging outcomes (Supplementary Fig. 5b). While high blood pressure in young and middle-aged adults is indicative of cardiometabolic dysfunction, in the elderly low blood pressure is common and more strongly associated with mortality and frailty³⁻⁵, though high blood pressure is also detrimental⁶. The differences between PhenoAge and the org5. an age models could be due to differences in the age distribution of the underlying training cohorts for the models. Our models were trained in the KADRC, which has a greater proportion of elderly individuals, while PhenoAge was trained in NHANES, which has a greater proportion of young individuals.

This kind of U-shaped relationship with age and aging outcomes is quite common and is also seen with BMI⁷. Prospective studies in older adults have shown that while obesity slightly increases mortality and cardiovascular disease risk, the highest risk groups are those with a BMI under 23. Interestingly, the intestine and pancreas age gaps show a negative association with BMI and obesity but a positive association with mortality risk, while the kidney age gap shows a positive association with BMI, suggesting that the full picture of organ health in aging and disease may be more complex than currently understood.”

6. More research should be performed and reported on the role of CPLX1, CPLX2, NRXN3, and PPP3R1 in brain aging and disease. Can these proteins be good targets or point to the possible protein targets? These were previously implicated in brain aging. Any new findings from the model? Any other important features that could be reported and used for target discovery? This question is valid for other organs.

We entirely agree with the reviewer that one aspect of our organ specific age modeling is to derive biologically and therapeutically relevant information from the proteins in a model. At the same time, as discussed above, any such proteins will need to be validated experimentally in future studies. Nevertheless, we have now performed additional research on brain model proteins and include here A) a review of existing literature on some of the highest weighted CognitionBrain model proteins and B) expanded our computational analyses using publicly available datasets to better understand how brain-derived protein levels in plasma may reflect the biology of the brain tissue, as these insights would inform their use as biomarkers and drug targets.

A) Literature review

We surveyed the literature to better understand the roles of highly weighted brain model proteins and their relation to brain aging and neurodegeneration (Table 1). In addition to identifying astrocyte marker ALDOC and age-associated synaptic proteins CPLX1, CPLX2, and NRXN3, our model also identified several new brain aging proteins including olfactomedin 1 (OLFM1), neuronal pentraxin receptor (NPTXR), stathmin 2 (STMN2), LanC like glutathione S-transferase 1 (LANCL1), fibroblast growth factor 4 (FGF4), and tenascin R (TNR), all of which been reported to be causal regulators of neuronal health or cognitive function in mice and humans (Table 1). LANCL1 is especially interesting in that brain-specific overexpression extends lifespan and decelerates disease progression in a mouse model of ALS²². The model also identified carnosine dipeptidase 1 (CNDP1), whose function in the brain is not known. Collectively, these studies nominate several new proteins with intriguing links to brain aging and neurodegeneration for future investigation. We discuss these potential future directions in the discussion, lines 375-380. The table below is also included in the Supplemental Text, line 96.

Reviewer Figure
Top 15 CognitionBrain model coefficients by absolute value.

Table 1: Literature review of top CognitionBrain model proteins and their relation to brain aging and cognitive function

Protein	Reference	Title	Organism
CPLX1	Yu, et. al., JAMA Psychiatry (2020)	Cortical Proteins Associated with Cognitive Resilience in Community-Dwelling Older Persons	Human
CPLX1	Glynn, et. al., Human Molecular Genetics (2005)	Profound ataxia in complexin I knockout mice masks a complex phenotype that includes exploratory and habituation deficits	Mouse
CPLX1,CPLX2	Tannenber, et. al., Neurochemistry International (2006)	Selective loss of synaptic proteins in Alzheimer's disease: Evidence for an increased severity with APOE ε4	Human
CPLX2	Begemann, et al. Arch. Gen. Psychiatry	Modification of Cognitive Performance in	Human

Protein	Reference	Title	Organism
	(2010)	Schizophrenia by Complexin 2 Gene Polymorphisms	
CPLX2	Glynn, et. al., Human Molecular Genetics (2003)	Complexin II is essential for normal neurological function in mice	Mouse
CPLX2	Li, et. al., Oxidative Medicine and Cellular Longevity (2020)	Proteomic Profile of Mouse Brain Aging Contributions to Mitochondrial Dysfunction, DNA Oxidative Damage, Loss of Neurotrophic Factor, and Synaptic and Ribosomal Proteins	Mouse
FGF4	Konijnenberg, et. al., Alzheimer's Research & Therapy (2020)	APOE ϵ 4 genotype-dependent cerebrospinal fluid proteomic signatures in Alzheimer's disease	Human
FGF4	Kosaka, et. al., Federation of American Societies for Experimental Biology (2006)	FGF-4 regulates neural progenitor cell proliferation and neuronal differentiation	Mouse
LANCL1	Drummond, et. al., Brain (2020)	Phosphorylated tau interactome in the human Alzheimer's disease brain	Human
LANCL1	Huang, Chen, and Peng, et. al., Developmental Cell (2014)	Developmental and Activity-Dependent Expression of LanCL1 Confers Antioxidant Activity Required for Neuronal Survival	Mouse
LANCL1	Tan and Chen, et. al., Cell Death & Differentiation (2020)	LanCL1 promotes motor neuron survival and extends the lifespan of amyotrophic lateral sclerosis mice	Mouse
NPTXR	Begcevic, et. al., F1000Research (2018)	Neuronal pentraxin receptor-1 is a new cerebrospinal fluid biomarker of Alzheimer's disease progression	Human
NPTXR	Pelkey, et. al. Neuron (2015)	Pentraxins Coordinate Excitatory Synapse Maturation and Circuit Integration of Parvalbumin Interneurons	Mouse
NRXN3	Hishimoto, et. al., Alzheimer's Research & Therapy (2019)	Neurexin 3 transmembrane and soluble isoform expression and splicing haplotype are associated with neuron inflammasome and Alzheimer's disease	Human
NRXN3	Zheng, et. al., Medicine (2018)	Low expression of aging-related NRXN3 is associated with Alzheimer disease	Human
NRXN3	Martinez-Mir, et. al., J Alz Disease (2013)	Genetic study of neurexin and neuroligin genes in Alzheimer's disease	Human
NRXN3	Aoto, et. al. Nature Neuroscience (2015)	Distinct circuit-dependent functions of presynaptic neurexin-3 at GABAergic and glutamatergic synapses	Mouse
OLFM1	Nakaya, et. al., Experimental Neurology (2013)	Deletion in the N-terminal half of olfactomedin 1 modifies its interaction with synaptic proteins and causes brain dystrophy and abnormal behavior in mice	Mouse
OLFM1	Nakaya, et. al., Journal of Biological Chemistry (2012)	Olfactomedin 1 Interacts with the Nogo A Receptor Complex to Regulate Axon Growth	Mouse
STMN2	Theunissen, et. al., Frontiers in Aging Neuroscience (2021)	Novel STMN2 Variant Linked to Amyotrophic Lateral Sclerosis Risk and Clinical Phenotype	Human
STMN2	Krus, et. al., Cell Reports (2022)	Loss of Stathmin-2, a hallmark of TDP-43-associated ALS, causes motor neuropathy	Human
STMN2	San Juan and Nash, et. al., Neuron (2022)	Loss of mouse Stmn2 function causes motor neuropathy	Mouse

Protein	Reference	Title	Organism
TNR	Wagner, et. al., Genetics in Medicine (2020)	Loss of TNR causes a nonprogressive neurodevelopmental disorder with spasticity and transient opisthotonus	Human
TNR	Dankovich, et. al., Nature Communications (2021)	Extracellular matrix remodeling through endocytosis and resurfacing of Tenascin-R	Mouse
TNR	Bauch and Faissner, Cells (2022)	The Extracellular Matrix Proteins Tenascin-C and Tenascin-R Retard Oligodendrocyte Precursor Maturation and Myelin Regeneration in a Cuprizone-Induced Long-Term Demyelination Animal Model	Mouse

B) Plasma protein levels vs tissue protein levels

Figure 3

g, Changes with age and AD of top CognitionBrain proteins across tissues (plasma, brain) and molecular layers (protein, bulk RNA, single-cell RNA). Proteins with significant changes in both fluid and solid tissues shown. Plasma changes with age and AD were assessed using the linear model: Protein ~ Age + AD + Sex in the Knight-ADRC cohort. Brain protein and bulk RNA changes with AD were derived from Johnson et. al. 2022 supplementary tables.. Brain single-cell RNA changes with AD were from Haney et. al. 2023.

We also assessed how the highest weighted CognitionBrain proteins change with age and AD in plasma (KADRC and SADRC cohorts) and in brain tissue using publicly available datasets. We report our findings in lines 281-287: “We assessed the highest weighted CognitionBrain proteins for their changes with age and AD across plasma in the Knight-ADRC and Stanford-ADRC cohorts, as well as their changes with AD in brain tissue at the protein⁵³, bulk RNA⁵³, and single-cell RNA levels from publicly available datasets (Fig. 3g). We observed a consistent pattern of decreases in AD brain tissue and increases in the blood with age and AD. This suggests that the increase of synapse and neurite growth related protein levels in the blood could reflect a loss or alteration in protein processing and subsequent shedding of these crucial factors in the brain. ”

Lastly, we performed further analyses to identify and understand peripheral contributions to cognitive decline and dementia which are summarized in a new Figure 4 and Extended Data Figures 7-10. This section discusses additional proteins of interest, lines 291-344.

Figure 4

f, Changes with age of cognition-optimized aging model proteins (linear model: Protein ~ Age + Sex). Age effect and negative \log_{10} Benjamini Hochberg corrected p-values (q-values) are shown. Size of bubbles are scaled by the absolute value of the average model weight.

g, Single-cell RNA expression of top 5 CognitionOrganismal protein encoding genes in human brain vasculature. Mean normalized expression values and fraction of cells expressing the genes are shown. Also shown are cell composition differences in the human brain vasculature between Alzheimer's and control, based on the same single-cell RNA-sequencing dataset.

h, Model of age-related cellular degradation of the human brain vasculature reflected in the plasma proteome.

i, StringDB protein-protein-interaction network of CognitionArtery and interacting proteins, as well as pathway enrichment of these proteins.

j, Model of age-related vascular calcification and extracellular matrix alterations reflected in the plasma proteome.

Lines 313-344: "To understand the biological processes and proteins involved in early cognitive decline, we plotted the aging trajectory of all model proteins and found that highly weighted CognitionOrganismal and CognitionArtery proteins changed with age earlier and at a faster rate than CognitionBrain and CognitionPancreas proteins (Fig. 4e). The earliest changes occurred in a highly correlated cluster of CognitionOrganismal proteins: pleiotrophin (PTN), transgelin (TAGLN), WNT1 Inducible Signaling Pathway Protein 2 (WISP2), CUB Domain Containing Protein 1 (CDCP1), and chordin like 1 (CHRDL1; Fig. 4f). Though not organ-specific, these genes were all highly expressed in the arteries and brain (Extended Data Fig. 10a). Single-cell expression of these genes in human vasculature^{55,56}, indicated these genes are expressed primarily by smooth muscle cells, pericytes, and fibroblasts (Fig. 4g; Extended Data Fig. 10b). Loss of brain pericytes, smooth muscle cells, and perivascular fibroblasts is associated with age and AD⁵⁶⁻⁵⁸ (Fig. 4g), and pericyte-specific deletion of PTN renders neurons prone to ischemic and excitotoxic injury⁵⁹. This early-changing signature in the CognitionOrganismal model may thus represent degenerative changes to the cellular integrity of the brain vasculature and the loss of its neuroprotective functions with aging (Fig. 4h).

The five proteins composing the CognitionArtery model, TNF receptor superfamily member 11b (TNFRSF11B), sclerostin (SOST), melanocortin 2 receptor accessory protein (MRAP2), frizzled related protein (FRZB), and matrix gla protein (MGP) were also primarily expressed in vascular smooth muscle cells, pericytes and fibroblasts⁵⁵ (Extended Data Fig. 10c) and are all strongly implicated in vascular calcification. TNFRSF11B/APOE double knockout mice show increased calcium deposition by vascular smooth muscle cells⁶⁰, MGP deficiency causing mutations in humans leads to Keutel syndrome, a disease characterized by soft tissue calcification⁶¹, and SOST and

FRZB are negative regulators of WNT signaling that drive calcification and are increased in the plasma of people with vascular calcification^{62,63}. We found that CognitionArtery proteins and the vascular signature in the CognitionOrganismal proteins form an interaction network using StringDB (Fig. 4i). Additional model proteins in this interaction network included integrin binding sialoprotein (IBSP), osteoglycin (OGN), collagen type III alpha 1 chain (COL3A1), proline rich and gla domain 1 (PRRG1), and growth arrest specific 6 (GAS6). Together these proteins are enriched in extracellular matrix, cartilage development, and osteoblast signaling pathways and implicate vascular calcification and extracellular matrix alterations as a major component of aging that underlies the early phases of cognitive decline and neurodegenerative disease (Fig. 4i-j).”

6b. Any other important features that could be reported and used for target discovery?

As our model weights and associations with diseases may help drug target prioritization efforts in the community, we have extensively updated our extended data figures and supplementary tables to provide all results from the study. Feature importance plots for all models are in Extended Data Figure 3. We have also made all aging models from this study publicly available and easily accessible through a python package called organage (<https://github.com/hamiltonoh/organage>). Users can input sample metadata (Age and Sex) and sample protein expression (SomaScan data) to output predicted organ ages and age gaps. Coefficients for all bootstrapped aging models are accessible both through the package and listed in Supplementary Tables 6,16. Additional summary statistics of the models are provided in Supplementary Table 7-8,17-18. Lastly, we provide statistics and visualizations of the changes with age and sex for all ~5,000 proteins on a shinyapp: https://twc-stanford.shinyapps.io/aging_plasma_proteome_v2/

6c. This question is valid for other organs.

We chose to focus the majority of our new target identification analyses to brain aging and cognitive decline, where these discoveries are novel, of high interest to the aging and neuroscience communities, and where we have extensive functional validation of the models from the Knight and Stanford ADRC cohorts. We also now investigate organismal and artery proteins more deeply in the context of cognitive decline, as discussed above. We hope future studies perform additional in-depth analyses for other organs and diseases.

7. Feature importance diagrams for each organ and for the plasma proteomics model should be analyzed, interpreted, and presented. Can we derive any new biological hypotheses from these models?

We now present feature importance diagrams for all organ aging models in Extended Data Figure 3 and their provide values in Supplementary Tables 6-8,16-18. We analyze and interpret proteins from the kidney and heart in Figure 2, the brain in Figure 3, and the brain, organismal, and artery models in Figure 4. As discussed above, the revised manuscript now includes multiple new hypotheses for future studies.

Minor

1. Only a few authors disclosed competing interests. Usually, it is a good idea to disclose all of the commercial affiliations of all authors.

All authors have declared their competing interests relevant to this manuscript in accordance with Nature publishing policies.

2. Seminal literature on organismal and tissue-specific biological aging biomarkers is missing. Building machine learning models on transcriptomic and proteomic data is not a novel concept, and there are even granted patents on multiple technologies in the field with priority dating back to 2017.

Thank you for the additional literature suggestions, we are open to specific suggestions on papers the reviewer believes are relevant and should be cited. Based on the commentary on patents, we now cite studies, for example, from Dr. Alex Zhavoronkov's group, which are linked to granted patents mentioned by the reviewer. If they believe we are missing other important literature, we are open to citing additional sources.

Referee #4

This manuscript by Oh et al. describes the creation of 19 organ-specific ageing clocks, of which 15 were able to predict chronological age in two independent testing cohorts. The authors also create an organismal ageing clock and compare its performance with those that are organ-specific. Oh et al. test the association of the age gaps derived from each of their clocks with all-cause mortality and age-related phenotypes relating to the relevant organs.

It is exciting to see for the first time how organ-specific clocks perform, particularly as the authors have assessed their performance using organ-specific follow up outcomes. The organ-specific approach and the observations presented would be a step forward for the ageing clock field however, revisions need to be made to make the findings reproducible and to ensure the validity of the results and conclusions drawn before the manuscript is ready for publication.

We appreciate this reviewer's excitement for our work and thank them for their thorough review. Their critique on the statistical validity and reproducibility of the results were valid and well taken. We believe we have addressed these concerns and greatly improved the manuscript with the help of this reviewer's suggestions.

Major

1. On the point of validity, there does not appear to be adjustment of p-values to account for multiple testing. Given that in several analyses there are multiple clock age gaps being tested against multiple outcomes, it would be expected to report: the number of tests performed, the method of p-value adjustment (Bonferroni, FDR, Benjamini-Hochberg etc.), the threshold for statistical significance of the adjusted p-values (5%, 1% etc.) and the number of tests that pass these criteria and are statistically significant clearly for each analysis in the main text. As at present it is not clear which of the reported associations are statistically significant. For example, multiple times in the manuscript the authors state that "several" age gaps are significantly associated, however they should state exactly how many tests were statistically significant and at what threshold.

We apologize for this oversight and miscommunication. We now report all tests for all models in this revised manuscript and report false discovery rate control using the Benjamini Hochberg method. We have made this clear in the text by removing vague language such as "several" age gaps and now explicitly enumerate which models are significant on which tests. We have also made this clear in the figures and provide more comprehensive supplementary tables. This is also highlighted in response to Reviewer 1. The major revisions related to this point are summarized below.

1. We include here a study design flowchart detailing on statistical testing for the reviewers.

Supplementary Figure 2. Study design flow chart

2. We have added a detailed statistical methods section specifically to address false discovery rate control. Lines 590-638.
3. All statistical tests and the number of statistically significant results after FDR correction are now clearly listed in the text, figures, and supplementary tables.
 - a. Figure 1b-e, 2i-j, and Extended Data Fig. 4, now show all 13 aging models.
 - b. Supplementary tables 9-13 contain all statistics for all statistical tests performed on these 13 models with FDR correction across each table.
 - c. Lines 109-111: "All 11 organ aging models and the organismal model significantly estimated age in all five cohorts after multiple test correction (Supplementary Fig. 3b)"
 - d. Lines 131-134 and Lines 143-146: "To assess the relationship between organ age and biological aging, we tested whether organ e-ageotypes were associated with nine age-related disease states for which we had sufficient data in at least two independent cohorts; Alzheimer's disease, atrial fibrillation, cerebrovascular disease, diabetes, heart attack, hypercholesterolemia, hypertension, obesity, and gait impairment... At the whole population level, the relationships between organ age gaps and disease showed the same trends as ageotypes, but more diseases were significantly associated with age gaps due to higher statistical power (65/117, 56%, statistically significant after multiple test correction, Extended Data Fig. 4e, Supplementary Table 10)."
 - e. Lines 173-180: "we used longitudinal follow-up among healthy participants in the LonGenity cohort to test if organ age was significantly associated with future heart failure risk (Fig. 2i, Supplementary Table

11). We found that among people with no active disease or clinically abnormal biomarkers at baseline, every 4.1 years of additional heart age (1 standard deviation) conferred an almost 2.5 fold increased risk of heart failure over a 15 year follow-up (23% increased risk per year of heart aging, Fig. 2i). Age gaps from multiple other tissues, but not the conventional aging model, also trended towards significance.”

- f. Lines 181-184: “We next tested the associations between organ age gaps and all-cause mortality. We found that the age gaps from 10 out of 11 organs, the organismal, and the conventional model were significantly associated with future risk of all-cause mortality after multiple test correction in the LonGenity cohort over 15 years of follow-up (Fig. 2j, Supplementary Table 12). ”
- g. Lines 187-189 and Lines 195-197: “we tested the associations between organ age gaps and 43 clinical biochemistry and cell count markers in the test cohort Covance (Extended Data Fig. 5, Supplementary Fig. 4, see Supplement for additional discussion)... We found 226 out of 559 (40%) associations between organ age gaps and clinical biochemistry markers were significant after multiple testing correction (Extended Data Fig. 5c, Supplementary Table 13).”

2. On a related note, the authors highlight “strong” associations of the heart age gap with multiple heart-related phenotypes, both in this instance and in general it would be helpful to the reader to report how strong in the main text, by explicitly stating the effect size or hazard ratio and its associated standard error or confidence intervals along with a p-value. Even if effect sizes are indicated in figures, it is helpful to understand the context of the result presented. Further, reporting of effect sizes in the main text makes it easier for others to reproduce and replicate the authors results.

We agree with the reviewer, and we have now removed all such subjective language on the “strength” of associations. Instead, we have now included detailed statistics for all tests in supplementary tables. It is our understanding that it is not in the Nature style guidelines to include effect and p-value stats in the main text. However, we are certainly open to doing so in consultation with the editor.

3. Additionally, it would improve transparency to include full results from all the analysis as supplementary or extended data tables. In the manuscript the authors highlight key findings for specific organs of interest in figures. Given the number of additional associations run that are not explored in the main text, including the full association results of every clock age gap against every outcome (effect sizes/hazard ratios, standard errors/confidence intervals, p-values and adjusted p-values) for each analysis would give readers the opportunity to investigate results of specific interest not only those that were significant in this analysis/highlighted by the authors.

This is a valid point and we have now extensively expanded our supplementary tables to include effect sizes/hazard ratios, confidence intervals, p-values, and adjusted p-values from all analyses. We now provide 23 supplementary tables for this purpose.

4. Further, it would aid repeatability and reproducibility if the authors included for each ageing clock the proteins included and their model coefficients in supplementary tables as done by many ageing clocks papers.

We now provide all this information. Coefficients for all bootstrapped aging models are accessible both through the package and listed in Supplementary Tables 6,16. Additional summary statistics of the models are provided in Supplementary Table 7-8,17-18. We have also made all aging models from this study publicly available and

easily accessible through a python package called organage (<https://github.com/hamiltonoh/organage>). Users can input sample metadata (Age and Sex) and sample protein expression (SomaScan data) to output predicted organ ages and age gaps.

5. The authors describe the creation of their CognitionBrain model using their FIBA algorithm, report its age gap’s association with risk of dementia progression and go on to comment on its comparison with the gold standard biomarker plasma pTau-181. Clarification is needed in the first sentence of this paragraph as to exactly how this was done. As far as I understand the individuals were ranked and those in the top 25% based on CognitionBrain age gap who were also in the top 25% when ranked based on plasma pTau-181 levels were compared to those individuals who appeared in the bottom 25% of both lists? If so, I am unsure if this analysis supports the conclusions stated: “These findings indicate that CognitionBrain age gap uncovers novel biology related to AD progression not captured by current pathology-based markers.”. In order to evidence this statement, the authors could run Cox proportional hazard models for the AD outcome (2-point increase in CDR-SB within 5 years) with pTau-181 level as the predictor (along with sex and chronological age as covariates) and compare it to a model run with both pTau-181 level and CognitionBrain age gap as predictors. If the hazard ratio for the CognitionBrain age gap is significant in the combined model, that would indicate that their novel marker was capturing additional information over and above what is being provided by the current gold standard marker.

We apologize for the confusion, and we appreciate the suggestion of testing a multivariate model with “pTau-181 level and CognitionBrain age gap as predictors”. For the revised manuscript, we have implemented this reviewer’s suggestion into the main figure (Fig. 3d-e). We tested a multivariate cox model of dementia progression risk with plasma pTau181 and the CognitionBrain (renamed since optimized on dementia status instead of cognition) age gap as covariates. To be even more rigorous, we also added an AD polygenic risk score, baseline dementia status, and age as additional covariates. As shown, the CognitionBrain age gap covariate is significant and has a slightly larger adjusted hazard ratio than plasma pTau181, indicating that our model captures additional information over and above what is provided by the current gold standard proteomic and genomic AD biomarkers.

Figure 3

d, Cox proportional hazard model of dementia progression risk, including biomarkers of AD and predictors of cognitive decline as covariates (2pt increase in CDR-Sum of Boxes ~ CognitionBrainAgeGap + CDR-Global + PlasmaPTau181 + ADPolygenicRiskScoreAD + Age) within 5 years in the Stanford-ADRC cohort. 48 events out of 325 individuals. Hazard ratios, 95% confidence intervals, and p-values for all covariates are shown.

e, Cumulative incidence plot derived from hazard model in **f**. Shown are predicted trajectories based on changing levels of the two significant biomarker covariates, CognitionBrainAgeGap and PlasmaPTau181, while all other covariates remain constant. Individuals with biomarker levels 2 standard deviations above average had over a 75% probability of dementia progression, while individuals with levels 2 standard deviations below average had a 10% probability of dementia progression within 5 years in the Stanford-ADRC cohort.

6. First, compared to previous ageing clock papers the sample size of the cohort used for training (n=353) is modest, meaning that there is a greater chance of overfitting of the models (there are lower correlations between predicted age and chronological age in the testing cohorts compared to the training cohort) and them being less generalisable than models trained with larger sample sizes.

To address this concern, we have added two new cohorts to our study: the Knight Alzheimer's Disease Research Center (KADRC) cohort (n=3,075) and the Stanford Aging and Memory Study (SAMS) cohort (n=192). We also changed our training cohort from Covance aged 60+ (n=353) to the healthy individuals in the KADRC cohort (n=1,398). This means we have increased our number of independent cohorts from three to five, nearly quadrupled our training data sample size, and increased our total sample size from 1,727 to 5,678, making our revised study the largest plasma proteomic aging clock study to date and the largest plasma proteomics study of clinically assessed Alzheimer's disease participants to our knowledge.

Figure 1

a, Schematic illustrating the study design of estimating and validating organ-specific biological age (See Methods).

With the addition of these cohorts, we saw improvements in model accuracy and fit across the board (Fig. 1b). More importantly, we find that this cohort number and sample size is sufficient to discover highly significant and

reproducible associations between age gaps and chronic diseases in multiple individual cohorts and cross-cohort meta-analyses. This is reflected in the text in Lines 130-213 (lines 131-146 excerpted here):

a. “To assess the relationship between organ age and biological aging, we tested whether organ e-ageotypes were associated with nine age-related disease states for which we had sufficient data in at least two independent cohorts; Alzheimer’s disease, atrial fibrillation, cerebrovascular disease, diabetes, heart attack, hypercholesterolemia, hypertension, obesity, and gait impairment. Organ e-ageotypes were associated with specific disease states with known high impact on their respective organs (23/117, 20%, associations significant in a meta-analysis after multiple testing correction, Extended Data Fig. 4d, Supplementary Table 9). The kidney ageotype was the most significantly associated with metabolic diseases (diabetes, obesity, hypercholesterolemia, hypertension), the heart ageotype was the most significantly associated with heart diseases (atrial fibrillation, heart attack), the muscle ageotype was the most significantly associated with gait impairment, the brain ageotype was the most significantly associated with cerebrovascular disease, and the organismal ageotype was the most significantly associated with Alzheimer’s disease. At the whole population level, the relationships between organ age gaps and disease showed the same trends as ageotypes, but more diseases were significantly associated with age gaps due to higher statistical power (65/117, 56%, statistically significant after multiple test correction, Extended Data Fig. 4e, Supplementary Table 10).”

We have also changed the visualization and statistical reporting of results to meta-analysis forest plots, which demonstrate the high reproducibility of our results across multiple independent cohorts. Full results of this analysis are now in Supplemental Table 10.

Figure 2

a, Forest plot displaying results from a cross-cohort meta-analysis of the association between the kidney age gap and hypertension history, controlling for age and sex (linear model: $AgeGap \sim Disease + Age + Sex$). Per cohort and total effect sizes, 95% confidence intervals, Benjamini Hochberg total p-value (q-values) are shown. P-values were corrected based on 117 tests of organ age gap associations with major comorbidities (Extended Data Fig. 4e).

b, As in **a**, but for kidney age gap versus diabetes history.

c, As in **a**, but for heart age gap versus atrial fibrillation or pacemaker history.

d, As in **a**, but for heart age gap versus heart attack history.

7. Second, is the restricted age range, due to the age range in the testing cohorts, the authors were limited to training models in individuals over 60 years of age. This does mean that while the performance of the models presented has been demonstrated in older individuals it has not in an age range that spans the general population. It has been shown previously that ageing clocks trained in an age restricted cohort do not replicate well in wider range of ages (<https://doi.org/10.1186/s12864-020->

07168-8) and given that many ageing clocks are trained in and have been successfully replicated in cohorts spanning a much wider age range, this caveat should be highlighted in the discussion, or the authors could seek to replicate their clocks in an additional cohort with a wider age range.

This is a valid concern which we have attempted to address with substantial additional training data. It is also a complex issue that warrants a detailed response with additional literature context, which we now provide here and in the revised manuscript.

a. First and foremost, in our revised study design we use the newly added KADRC cohort as our training cohort, which expands the sample size and age range of our training (age 27-104). We now use the entire Covance cohort (age 18-90) as one of the test cohorts (Fig. 1a, Supplementary Fig. 3b), and we can evaluate the degree to which our models may be “overfit” to aged individuals. Overall, we found that our models are overfit to some degree in all test cohorts, but this was not driven by the different age ranges in the cohorts. Some degree of overfitting is expected given technical variation in omics assays. In many cases, Covance showed the best fit (ie. Organismal, Liver, Immune, etc) – suggesting that the models selected proteins that change consistently across the lifespan, making them reproducible in both younger and older individuals.

b. However, there is still a larger representation of older adults than younger adults in our training data, as the mean age in the KADRC is 75, so out of an abundance of caution we have added an additional statement in the discussion urging caution when applying our models to younger populations, lines 386-387: “While our current models serve as a proof of principle for this approach, since they are trained and evaluated largely on older adults caution should be used when applying them to young people.”

There are clear tradeoffs between optimizing the aging measure for young adults or the elderly. Given the test cohorts we analyze in the manuscript are all older cohorts, as are most human datasets for diseases of aging, we believe it is important for the training cohort to match the test distributions better as this allows for more statistically sound and biologically correct inference.

There are many biomarkers of health which have a nonlinear relationship to aging outcomes, and in the elderly many relationships between biomarkers and health/mortality/frailty reverse direction. We believe this is very important to understand, especially when evaluating the relationship between training and test cohorts, and noticed that this is largely not discussed or accounted for in models of molecular aging. We have now used our analysis of the clinical biochemistry data in the Covance cohort to illustrate the nuance of this point. We believe this is an important contribution, and it supports our use of a primarily older training cohort to evaluate disease outcomes in older test cohorts which is the primary focus of the manuscript. We emphasize this in the newly added Supplemental Text, and in Supplementary Fig. 4, where we show U-shaped trait relationships for 3 important traits in the Covance cohort with age that are discussed in the text.

Supplemental Figure 3

b, Correlations between predicted vs chronological age in healthy individuals in the training (Knight-ADRC) and test (Covance, LonGenity, Stanford-ADRC, SAMS) cohorts for all aging models. All aging models significantly estimated age across five independent cohorts.

Supplemental Figure 4

b, U-shaped relationship between age and certain traits, including diastolic blood pressure, BMI, and alanine transaminase are shown.

1. Supplemental text lines 38-62: “There are many biomarkers of health which have a nonlinear relationship to aging outcomes, and in the elderly many relationships between biomarkers and health/mortality/frailty reverse direction compared to young and middle-aged adults. The distribution and mean age of the population that an aging model is trained on will thus impact associations with traits. This is not frequently discussed or accounted for in models of molecular aging.

Such a case is illustrated by diastolic blood pressure, where the strongest association was with heart aging (adjusted Pearson $r=-0.18$, $q=2.62e-10$). Nine organ age gaps (adipose, brain, control, heart, intestine, kidney, liver, muscle, organismal, pancreas) were significantly associated with decreases in diastolic blood pressure, while the opposite association was seen with the PhenoAge age gap (Supplementary Fig. 5a, Supplementary Table 13). Diastolic blood pressure was one of many traits with a U-shaped relationship to aging outcomes (Supplementary Fig. 5b). While high blood pressure in young and middle-aged adults is indicative of cardiometabolic dysfunction, in the elderly low blood pressure is common and more strongly associated with mortality and frailty³⁻⁵, though high blood pressure is also detrimental⁶. The differences between PhenoAge and the organ age models could be due to differences in the age distribution of the underlying training cohorts for the models. Our models were trained in the KADRC, which has a greater proportion of elderly individuals, while PhenoAge was trained in NHANES, which has a greater proportion of young individuals.

This kind of U-shaped relationship with age and aging outcomes is quite common and is also seen with BMI⁷. Prospective studies in older adults have shown that while obesity slightly increases mortality and cardiovascular disease risk, the highest risk groups are those with a BMI under 23. Interestingly, the intestine and pancreas age gaps show a negative association with BMI and obesity but a positive association with mortality risk, while the kidney age gap shows a positive association with BMI, suggesting that the full picture of organ health in aging and disease may be more complex than currently understood.”

Suggested Improvements

The manuscript shows association between the novel organ-specific age gaps and organ-specific outcomes as well as all-cause mortality. If the data were available, it would be nice to see if organ-specific age gaps were prognostic of (organ-relevant) disease-specific mortality.

This is a great suggestion that we would have loved to implement if we had the data, but we unfortunately do not have access to disease-specific mortality information in any cohort.

References

In addition to the groups own previous paper which presents novel plasma proteomics ageing clocks as mentioned above, if the authors are keeping to the scope of plasma proteomics rather than specifically the SomaLogic platform, they could add the additional two citations for papers which present novel ageing clocks built with plasma proteomics from the Olink assay: Enroth et al., 2015 (<https://doi.org/10.1038/srep17282>) and Macdonald-Dunlop et al., 2022 (<https://doi.org/10.18632/aging.203847>). Both of these papers trained novel proteomic ageing clocks that predicted chronological age in both training and testing samples (as well as additional independent cohorts in the case of Macdonald-Dunlop et al.) as well as demonstrating that models built using a much smaller subset of proteins were as predictive as those containing a much larger numbers of proteins.

We have added the citations to the text.

In the introduction, the sentence “Current methods to measure molecular ageing in living humans have largely measured blood cell age, yet do not provide direct information on the biological age of less accessible internal organs”, is the authors point that most ageing clock papers calculate one single biological age measure for the whole organism from blood-based measures – equivalent to their organismal clock – rather than separate ones for each organ system? If so, this could be clearer as it currently reads as though the point is that most clocks are made from blood-based measures which of course includes their own plasma proteomics-based clocks. The two DNA methylation-based clocks papers cited do not help clarify this, as there have been many papers that use a variety of different blood-based omics assays (including plasma proteomics) to derive measures of biological age.

We are sorry this statement was unclear to multiple reviewers, and we have changed the language in the introduction to be clearer. We have also expanded the citations to more accurately reflect the prior work in the field which motivated our approach. Lines 61-73:

“While many methods to measure molecular aging in humans have been developed^{10–18}, most of them provide just a single measure of aging for the whole body. This is difficult to interpret given the complexity of human aging trajectories, and no single method so far predicts aging outcomes in all organs. Some recent methods have used clinical chemistry markers which include some markers of organ function^{10,16,19}. However, these methods still generate a single composite score for the whole body and contain many markers with low organ specificity, making them difficult to interpret for organ-specific aging. Methods to measure brain aging have used MRI-based brain volume and functional connectivity measurements, but they are costly, time-consuming, and do not provide molecular insights^{20,21}. Building off the wealth of literature and clinical practice that uses

certain organ-specific plasma proteins to non-invasively assess aspects of organ health, such as troponin T for heart damage²² and alanine transaminase for liver damage²³, we hypothesized that comprehensive quantification of organ-specific proteins in plasma could enable minimally invasive assessment and tracking of human aging for any organ.”

References

1. Levine, M. E. *et al.* An epigenetic biomarker of aging for lifespan and healthspan. *Aging* **10**, 573–591 (2018).
2. Lu, A. T. *et al.* DNA methylation GrimAge strongly predicts lifespan and healthspan. *Aging* **11**, 303–327 (2019).
3. Belsky, D. W. *et al.* DunedinPACE, a DNA methylation biomarker of the pace of aging. *eLife* **11**, e73420 (2022).
4. Ganz, P. *et al.* Development and Validation of a Protein-Based Risk Score for Cardiovascular Outcomes Among Patients With Stable Coronary Heart Disease. *JAMA* **315**, 2532–2541 (2016).
5. Wang, T. J. *et al.* Prognostic Utility of Novel Biomarkers of Cardiovascular Stress. *Circulation* **126**, 1596–1604 (2012).
6. Rutledge, J., Oh, H. & Wyss-Coray, T. Measuring biological age using omics data. *Nat. Rev. Genet.* 1–13 (2022) doi:10.1038/s41576-022-00511-7.
7. Lehallier, B., Shokhirev, M. N., Wyss-Coray, T. & Johnson, A. A. Data mining of human plasma proteins generates a multitude of highly predictive aging clocks that reflect different aspects of aging. *Aging Cell* **19**, e13256 (2020).
8. Lehallier, B. *et al.* Undulating changes in human plasma proteome profiles across the lifespan. *Nat. Med.* **25**, 1843–1850 (2019).
9. Tanaka, T. *et al.* Plasma proteomic biomarker signature of age predicts health and life span. *eLife* **9**, e61073 (2020).
10. Tanaka, T. *et al.* Plasma proteomic signature of age in healthy humans. *Aging Cell* **17**, e12799 (2018).
11. Enroth, S., Enroth, S. B., Johansson, Å. & Gyllenstein, U. Protein profiling reveals consequences of lifestyle choices on predicted biological aging. *Sci. Rep.* **5**, 17282 (2015).
12. Yang, C. *et al.* Genomic atlas of the proteome from brain, CSF and plasma prioritizes proteins implicated in neurological disorders. *Nat. Neurosci.* **24**, 1302–1312 (2021).
13. Walker, K. A. *et al.* Large-scale plasma proteomic analysis identifies proteins and pathways associated with dementia risk. *Nat. Aging* **1**, 473–489 (2021).
14. Bell, C. G. *et al.* DNA methylation aging clocks: challenges and recommendations. *Genome Biol.* **20**, 249 (2019).
15. Belsky, D. W. *et al.* Quantification of the pace of biological aging in humans through a blood test, the DunedinPoAm DNA methylation algorithm. *eLife* **9**, e54870 (2020).
16. Liem, F. *et al.* Predicting brain-age from multimodal imaging data captures cognitive impairment. *NeuroImage* **148**, 179–188 (2017).
17. Cole, J. H. *et al.* Brain age predicts mortality. *Mol. Psychiatry* **23**, 1385–1392 (2018).
18. Clausen, A. N. *et al.* Assessment of brain age in posttraumatic stress disorder: Findings from the ENIGMA PTSD and brain age working groups. *Brain Behav.* **12**, e2413 (2022).
19. Levine, M. E. Modeling the Rate of Senescence: Can Estimated Biological Age Predict Mortality More Accurately Than Chronological Age? *J. Gerontol. A. Biol. Sci. Med. Sci.* **68**, 667–674 (2013).
20. Belsky, D. W. *et al.* Quantification of biological aging in young adults. *Proc. Natl. Acad. Sci.* **112**, E4104–E4110 (2015).
21. Putin, E. *et al.* Deep biomarkers of human aging: Application of deep neural networks to biomarker development. *Aging* **8**, 1021–1030 (2016).
22. Tan, H. *et al.* LanCL1 promotes motor neuron survival and extends the lifespan of amyotrophic lateral sclerosis mice. *Cell Death Differ.* **27**, 1369–1382 (2020).

Reviewer Reports on the First Revision:

Referees' comments:

Referee #1 (Remarks to the Author):

The authors have gone to extraordinary lengths to address my earlier comments; they have clarified the text and figures, corrected for multiple comparisons and included some new and stronger analyses.

I congratulate them on a really exciting paper that deserves to be published.

Referee #2 (Remarks to the Author):

I don't see a reason to delay the publication of this study

Referee #4 (Remarks to the Author):

In the revised manuscript by Oh et al., the authors have done extensive work to address comments from myself and other reviewers: the addition of two cohorts, both increasing the number of validation cohorts and increasing the sample size in the training cohort, as well as the addition of new analyses.

The authors have also addressed the issue of statistical reporting that was highlighted by multiple reviewers. They have expanded the reporting of statistical tests in the: main text, figure legends, supplementary and extended materials, as well as adding specific sections to the methods. The total number of tests run, statistical significance thresholds and correction for multiple testing are all outlined for each of the analyses presented.

The authors addressed my specific comment on the comparison of the performance of the CognitionBrain age gap with the gold standard biomarker p-Tau 181 level in predicting AD progression. They used the suggested cox proportional hazard model approach and extended it, to not only show that their CognitionBrain age gap does add predictive information beyond the gold standard proteomic biomarker, but that it also provides information beyond what is provided by the genetic marker (AD PRS).

I would also like to highlight specific additions that strengthen the revised manuscript. First, the forest plots and meta-analysis approach that nicely demonstrate the consistency of their age gap-outcome association results across the multiple validation cohorts. Second, the discussion of the U-shaped relationships between age and certain age-related traits, e.g. BMI and diastolic blood pressure. This is an issue not often discussed or accounted for in ageing clocks papers. Third, the clustering into e-ageotypes and showing that individuals who are extreme agers in one organ are not necessarily extreme agers in other organs. This demonstrates nicely that the organ specific approach is adding knowledge to the field of ageing clocks in a way that single clocks for the whole body are

not able to.

I would recommend the revised manuscript for publication providing a few minor comments are addressed.

Minor comments

- 1) Regarding the pathway enrichment analysis results presented in Figure 4i. In the methods section it says that all human genes were used as a background. I would suggest that if only proteins included in models (e.g. CognitionArtery and CognitionOrganismal) were included in this analysis, that the genes encoding the ~893 SomaScan proteins measured and used throughout this work would be a more appropriate background. Given that, by only capturing this subset of proteins, as opposed to the whole proteome, there has already been a selection. This is only an issue if p-values for a formal enrichment analysis are included, if the authors wanted to only be descriptive and comment on which pathways these CognitionArtery proteins belonged to this would not be an issue.
- 2) It would be helpful to have 95% confidence intervals for the correlations of organ predicted age vs chronological age across cohorts, shown in supplementary figure 3b.
- 3) Extended data figure 7 – the x-axis labels read “ahronological”, is this potentially a typo and should be chronological?

Referee #5 (Remarks to the Author):

The revised manuscript by Oh et al. takes a novel approach to assessing organ age from profiles of circulating plasma proteins assessed using a targeted proteomics platform. The authors identify both interesting patterns of organismal and organ aging, as well as “age-gap” profiles conferring substantial risk for multiple disease phenotypes that are age-related and represent some of the major public health challenges confronting aging populations around the world. The authors have responded to the initial reviews by incorporating additional data and analyses, which further support their conclusions. Addressing some additional issues would help place the results in clinical and biological context.

1. It would be helpful to show the relationship of the hazard ratios (HR) to the inferred organ age gap. That is, how does the HR change as the age gap varies from 1, 2, 5, to 10 years.
2. It is helpful to know that the age-gap measures are ‘independent’ risk factors and the HRs often remain significant after adjustment. However, this is different from demonstrating whether they are clinically useful. Leaving aside the issue of whether these trajectories can be altered (a fascinating question obviously beyond the current scope), independent risk factors even with significant HRs may not materially influence how we assess many subjects. To get at this question, Receiver Operating Characteristic (ROC) analyses describing how much the Area Under the Curve (AUC) is changed by the new model as compared to the best current model (both p-value and degree of AUC

change) or inclusion of the new model with the old would be most helpful. In addition, some assessment of how many subjects are changed into a different category of risk (net reassignment analysis) that might influence treatment strategies, would also be informative. For heart disease, for example, there are well-established thresholds at which 10 year risk warrants pharmacological intervention. These approaches would provide a better sense of how often the plasma proteomic / age-gap analyses identify subjects who would benefit from intervention (according to current guidelines) that are otherwise missed by current models. (Even if a hazard ratio survives adjustment for multiple other variables and is independent, it may simply indicate a higher risk in subjects already known to be above the threshold risk for concern).

3. Obviously, the 18.4% of individuals with accelerated organ aging are of particular interest. It is somewhat surprising that only 1.7% showed extreme aging of multiple organs, particularly given the clinical observation that co-morbidities across organs are common among age-related disease phenotypes. Is it possible that this is a result of the approach in that mutually exclusive single organ-specific proteins were used? There are examples of “oligo-organ enriched” proteins (i.e. proteins enriched across a small number of organs). Perhaps analyses that included such proteins would identify more subjects with concordant, accelerated aging across several interrelated organs (heart-kidney, heart-brain, etc).

4. These observations also raise questions about the boundary between normal aging and organ pathology. It seems possible that at least some of the subjects identified as having accelerated aging of a specific organ, actually have subclinical disease in that organ, which has altered the plasma proteome. For example, for cardiac age, NT-BNP and Troponin were major contributors to the model. While there is some (not entirely consistent) evidence each of these can increase in healthy aging, they are also indicators of cardiac pathology and already recognized predictors of adverse cardiovascular outcomes. It seems plausible that these analyses include subjects who have existing disease that hasn't yet presented clinically and thus as labeled as 'healthy'. This might be revealed by more detailed phenotyping (e.g. cardiac MRI or even echo with GLS). Assuming such phenotyping data are not available, this possibility should at least be acknowledged and discussed. The clinical implications may still be similar (i.e., identification of subjects at high risk who should be considered for preventive measures) but the interpretation, particularly as it relates to the biology of normal aging and uncovering the responsible mechanisms, might be considerably different.

Referee #6 (Remarks to the Author):

Manuscript:

Organ-specific aging signatures in the plasma proteome 1 track health and disease

Authors:

Hamilton Oh, Jarod Rutledge, ... , Tony Wyss-Coray

The manuscript H. Oh, J. Rutledge et al., explores proteomic aging signatures in 11 major human

organs in five independent cohorts that include a total of 5678 adults across their lifespan. In the original manuscript, reviewers identified imitations in statistical validity, a small cohort size, limited accessibility of the raw data and the lack of comparison to other established biological clock datasets. The authors addressed these concerns which thereby significantly enhanced the overall quality of the manuscript.

1) Statistical validity.

Reviewers pointed out that p-values were not accounted for multiple testing or at least have not been described properly in the original manuscript. In the revised manuscripts, authors report all tests for all models and report false discovery rate control using the Benjamini-Hochberg method (which is a suitable method for multiple testing correction in proteomics).

2) Low cohort numbers / limited age range for machine learning aging models.

The original manuscript contained 1,727 participants and resulted in low R values for several organs. The authors included 2 new cohort studies the Knight Alzheimer's Disease Research Center (KADRC) cohort (n=3,075) and the Stanford Aging and Memory Study (SAMS) cohort (n=192). The inclusion of these two cohorts improved the age distribution, also covering younger aging adults and strengthens the findings.

3) Data sharing and Transparency.

The original manuscript was lacking full raw data from all the performed analysis, which has been addressed in the revision. The revised manuscript includes Suppl. Tables with the raw data and performed analysis.

4) Comparison of organ-specific aging biomarkers to more established biological clocks from previous studies.

The authors now included comparison of the organ-specific clocks to previous proteomics, Levine's PhenoAge, and MRI-based brain age in the revised manuscript. They find statistically significant associations of their organ aging models with other aging clocks from previous studies (in Fig. 1, Extended data Fig. 5 and Suppl. Fig 6); some of the correlations are rather small, but adding this information from previous studies still strengthens the manuscript to better understand the association of the organ aging model with more established biological clocks. The authors also discuss these limitations in the discussion.

However, some outstanding comments still remain that need to be addressed:

1. To support their conclusions and findings, the manuscript relies heavily on the assignment of proteins to specific tissues/organs.

“We mapped the organ-specific plasma proteome using human organ bulk RNA-seq data from the Genotype-Tissue Expression (GTEx) project. We classified genes as “organ enriched” if they were expressed at least 4 times higher in one organ compared to any other organ, according to the definition proposed in the Human Protein Atlas”

A possible problem using this approach would arise from the fact that the correlation between mRNA levels and protein levels is typically poor in human tissues as well as shown by animal studies. See for example, a recent proteome/ transcriptome analysis of 32 human tissues that showed a median Spearman correlation of only 0.46 and between 90.2 and 0.6 roughly (Jiang et al., Cell 2020 Jiang et al., 2020 Cell 183, 269-283). Thus, it remains puzzling what it means for protein levels that mRNAs are expressed 4 times more in one cell or tissue or organ than in other?

2. Do the authors know how much proteins can be found in the studied tissues/organs, and how this would then translate to protein plasma levels? For example, a protein X could be more enriched in a tissue Y, but the changes in plasma levels might be coming from many different tissues; meaning many tissues may contribute to the plasma level.

3. The Somascan assay provides an indirect detection and measurement of proteins. This is not mass-spectrometry (MS) where the sequence positively identifies the protein. Here, the proteins were detected by binding to a probe that might have non-specific binders. This happens mainly when some proteins are really abundant and override the affinity of the probe for a specific protein.

The authors said that this platform has been used multiple times and that this provides “validation”. However this indeed may not validate the platform. I’d say that one should do some sort of validation by MS to be more confident. Since the platform provides so-called a directed assay by selecting the targets it does not have a power to be discovery-free as the MS. The main advantage might be the cost, but it has severe limitations as well.

4. The targets are preselected and cover roughly only 4% of the human proteome. Checking online I couldn’t get the whole dataset from there. The company lists panels, such neurological, cardio, metabolic, ect... similar to Olink (although the technology is different). I guess that the authors mixed several panels, but this has not been explained in the manuscript. One can have proteins that are really abundant, relevant, markers, etc... but that they don’t quantitate because they have not been included in the panel. So this is a major limitation.

5. Overall, the analysis was done with a low number of proteins, 856 organ enriched, 17.9% of their plasma detected proteins and around a 4% of the proteome. Really low numbers in general.

6. Technically, I didn’t see information about how the blood samples were collected for different cohorts. Some refer to previous papers, but there is a potential concern how comparable were the extraction conditions and plasma preparation, storage, etc.

7. The cohorts are very different, irrespective of their individual collection purposes (like ALZ) in

terms of age range, genetic background and number of participants, which potentially skew data analysis and interpretation.

8. The authors state:

We trained our models in 1,398 cognitively unimpaired participants from the Knight-ADRC cohort. We evaluated their performance in the Covance (n = 1,029), LonGenity (n = 962), SAMS (n= 192), Stanford-ADRC (n=409) cohorts, and Knight-ADRC cognitively impaired subjects (n = 1,677)

But, do we know how the age-gap for each organ was performed? The authors say that one can use plasma proteome to determine the age-gap and refer to previous papers, but applying the same concept to organs could be a stretch without proper validation and evidence. It is also not clear to this reviewer how the authors established the chronology for aging in different organs, particularly from the cohorts that includes individuals of different ages, but the numbers of individuals at different ages within different cohorts are not shown. For example, the authors state that they used the KADRC to train the model. This group has individuals from 27-104 with the SD of 75 yrs; it is not shown how many young individuals are in this group. Similarly, they used only 1398 “healthy patients”, but they don’t give the age range for this subset either.

9. They seem to be “very precise” by defining the age gap, for example:

Individuals with hypertension had kidneys that were approximately 1 year older than their same-aged peers, while individuals with diabetes had kidneys approximately 1.3 years older (Fig. 2a-b, Supplementary Table 8,10

With just a few “organ specific” proteins can one have such a level of exactness? For example the paper uses only 12 kidney-specific proteins and established 1 year older. What’s the error of this calculation?

10. The usual naming of handpicked molecules and generalizations.

“synaptic proteins complexin 1 (CPLX1), complexin 2 (CPLX2), and neuroligin 3 (NLGN3) – which all have genetic links to cognition and AD44–48, and stathmin 2 (STMN2) and olfactomedin 1 (OLFM1) – which are involved in neurite outgrowth and axon growth cone collapse^{49,50}”

One can hand pick any neuronal protein and find correlation to disease. What is the biological meaning of this kind of statements?

Author Rebuttals to First Revision:

Referee #1 (Remarks to the Author):

The authors have gone to extraordinary lengths to address my earlier comments; they have clarified the text and figures, corrected for multiple comparisons and included some new and stronger analyses. I congratulate them on a really exciting paper that deserves to be published.

We thank this reviewer for their kind comments and thorough review, which helped us greatly improve the manuscript.

Referee #2 (Remarks to the Author):

I don't see a reason to delay the publication of this study

We thank this reviewer for their thorough review, which helped us add more interesting biological/clinical context/interpretation into our manuscript.

Referee #4 (Remarks to the Author):

I would recommend the revised manuscript for publication providing a few minor comments are addressed.

We thank this reviewer for their thorough review, which helped us greatly improve the manuscript. We respond to the minor comments below.

Minor comments

1) Regarding the pathway enrichment analysis results presented in Figure 4i. In the methods section it says that all human genes were used as a background. I would suggest that if only proteins included in models (e.g. CognitionArtery and CognitionOrganismal) were included in this analysis, that the genes encoding the ~893 SomaScan proteins measured and used throughout this work would be a more appropriate background. Given that, by only capturing this subset of proteins, as opposed to the whole proteome, there has already been a selection. This is only an issue if p-values for a formal enrichment analysis are included, if the authors wanted to only be descriptive and comment on which pathways these CognitionArtery proteins belonged to this would not be an issue.

This is a good point. Since our goal here is to be descriptive and comment on potentially relevant pathways, with no claims about statistical enrichment from the plasma proteome, we have changed the x-axis to be % pathway overlap, defined as the number of genes in a given pathway divided by the number of genes queried (aka "precision"). This is updated in Figure 4i.

2) It would be helpful to have 95% confidence intervals for the correlations of organ predicted age vs chronological age across cohorts, shown in supplementary figure 3b.

Given the presence of 5 cohorts per row, we believe the 95% confidence intervals would be too cluttered. Instead, we have added the confidence interval statistics to Supplementary Tables 8 and 19.

3) Extended data figure 7 – the x-axis labels read “ahronological”, is this potentially a typo and should be chronological?

Thank you for catching this. It has been fixed.

Referee #5 (Remarks to the Author):

The revised manuscript by Oh et al. takes a novel approach to assessing organ age from profiles of circulating plasma proteins assessed using a targeted proteomics platform. The authors identify both interesting patterns of organismal and organ aging, as well as “age-gap” profiles conferring substantial risk for multiple disease phenotypes that are age-related and represent some of the major public health challenges confronting aging populations around the world. The authors have responded to the initial reviews by incorporating additional data and analyses, which further support their conclusions. Addressing some additional issues would help place the results in clinical and biological context.

We thank this reviewer for their thorough review.

1. It would be helpful to show the relationship of the hazard ratios (HR) to the inferred organ age gap. That is, how does the HR change as the age gap varies from 1, 2, 5, to 10 years.

This is a great suggestion to better understand potential non-linear associations between age gaps and disease risk. Given low sample size with other traits, we decided to do this recommended analysis for mortality risk in the LonGenity cohort where we have 864 individuals, 173 of whom died within up to 15 years. Specifically, we binned individuals into different age gap groups:

- Bin -2 ($-2.5 < \text{age gap} < -1.5$)
- Bin -1 ($-1.5 < \text{age gap} < -0.5$)
- Bin 0 ($-0.5 < \text{age gap} < +0.5$)
- Bin +1 ($+0.5 < \text{age gap} < +1.5$)
- Bin +2 ($+1.5 < \text{age gap} < +2.5$)
- Bin +3 ($+2.5 < \text{age gap} < +3.5$)
- Bin -3 and other more extreme bins were removed due to low sample size

We then compared every non-zero group with the zero group (denoting the non-zero group as 1 and the zero group as 0) for changes in mortality risk. We did this analysis for each of the aging models. We did not adjust for multiple comparisons because the assumptions were not met: each statistical test is done in a different subset of individuals, and tests for different bins in the same organ are generally correlated. Below are the results, which we provide as a new Supplementary Figure 4 and Table 13. These results are described in the Supplementary Text, and now referenced in the main text, lines 193-195.

Interestingly, the association between the age gap and mortality risk is non-linear for some organs, such as the heart, brain, pancreas, and kidney. The relationship with the heart age gap seems to be U-shaped where both high (+1, +2, +3) and extremely low heart age gaps (-2) are associated with increased mortality risk. The kidney age gap is also interesting in that it is not associated with mortality risk when looking at the whole age gap distribution (Fig. 2j), but the +3 age gap group is positively associated with mortality, suggesting the “extreme agers” framework may be more useful for certain organs and traits.

Other organs, including the organismal, adipose, artery, and immune, show a more linear relationship with mortality risk. Whether these nonlinear dynamics also exist for other aging biomarkers, such as methylation clocks, is unknown. This analysis points to a need for additional studies on the relationship between extreme aging and disease risk.

Supplementary Figure 4. Age gaps versus mortality risk, stratified by age gap bins.

a, Binned cox proportional hazard regression analysis in mortality risk, controlling for age and sex, within 15 years in the LonGenity cohort. 173 events out of 864 individuals were grouped into different z-scored age gap bins: -2, -1, 0, +1, +2, +3 (-3 was removed due to low sample size). Bin limits were +/- 0.5. Each non-zero group was compared with the zero group (denoting the non-zero group as 1 and the zero group as 0) for changes in mortality risk: $MortalityRisk \sim AgeGapBin (binary) + Age + Sex$. This analysis was performed for each aging model separately. Hazard ratios, 95% confidence intervals, p-values, and sample size for age gap bins are shown. While some age gaps, including the organismal, adipose, artery, and immune, show a relatively linear association with mortality risk, others such as the heart, brain, pancreas, kidney appear to be non-linear. The relationship with the heart age gap is U-shaped where both high (+1, +2, +3) and low (-2) heart age gaps are associated with increased mortality risk. For the kidney, which is not significantly associated with mortality when looking at the entire age gap range (Fig. 2j), only the +3 group is positively associated with mortality, suggesting the “extreme agers” framework may be more useful for certain organs and traits.

2. It is helpful to know that the age-gap measures are ‘independent’ risk factors and the HRs often remain significant after adjustment. However, this is different from demonstrating whether they are clinically useful. Leaving aside the issue of whether these trajectories can be altered (a fascinating question obviously beyond the current scope), independent risk factors even with significant HRs may not materially influence how we assess many subjects. To get at this question, Receiver Operating Characteristic (ROC) analyses describing how much the Area Under the Curve (AUC) is changed by the new model as compared to the best current model (both p-value and degree of AUC change) or inclusion of the new model with the old would be most helpful. In addition, some assessment of how many subjects are changed into a different category of risk (net reassignment analysis) that might influence treatment strategies, would also be informative. For heart disease, for example, there are well-established thresholds at which 10 year risk warrants pharmacological intervention. These approaches would provide a better sense of how often the plasma proteomic / age-gap analyses identify subjects who would benefit from intervention (according to current guidelines) that are otherwise missed by current models. (Even if a hazard ratio survives adjustment for multiple other variables and is independent, it may simply indicate a higher risk in subjects already known to be above the threshold risk for concern).

This is a very sharp point. We agree, an analysis to determine the added clinical utility of our models to the current standard is important and may lead to a higher impact discovery. Unfortunately, we do not have the statistical power or appropriate clinical chemistry measurements of Troponin, proNT-BNP, and other markers to perform this analysis in heart disease risk. We were able to perform the recommended analysis in regard to risk of cognitive decline in the Stanford-ADRC cohort, where we have 324 individuals, 48 of whom declined cognitively within up to 5 years. Specifically, we compared the concordance-indexes (an AUC equivalent for proportional hazards models) between a proportional hazards model that included clinically relevant covariates but no CognitionBrain age gap, versus a model which additionally included the CognitionBrain age gap (Reviewer Figure 1a-b).

We observed a slight increase in the concordance-index suggesting the addition of the CognitionBrain age gap provides additional information, though the degree to which this is clinically useful is unclear given the small change and relatively low sample size. A comparison of the risk scores from both models showed very high correlation and no obvious signs of risk category reassignment (Reviewer Figure 1c). Given the relatively low sample size, heterogeneity in baseline dementia scores, and potential non-linear relationships between age gaps and cognitive decline, we believe a more rigorous analysis in a larger cohort, stratified by baseline dementia ratings, age gap bins, and other covariates is warranted in future studies to determine clinical relevance. We have adjusted claims in the text around clinical relevance.

- Lines 267-268: Taken together, these data suggest CognitionBrain age gap provides ~~clinically relevant~~ molecular information about brain aging not captured by other approaches.

Reviewer Figure 1

a-b. Cox proportional hazard regression analysis in risk of cognitive decline within up to 5 years in the Stanford-ADRC cohort. 48 events out of 324 individuals. Hazard ratios, 95% confidence intervals, p-values, and concordance-index (AUC) are shown. **a.** Risk of cognitive decline (2-point increase in CDR-Sum of boxes) ~ baseline CDR-Global + Plasma Ptau181 + Age + AD Polygenic Risk Score. **b.** Risk of cognitive decline (2-point increase in CDR-Sum of boxes) ~ CognitionBrain age gap + baseline CDR-Global + Plasma Ptau181 + Age + AD Polygenic Risk Score.

c. Per individual hazard model risk scores based on model a and model b are shown.

3. Obviously, the 18.4% of individuals with accelerated organ aging are of particular interest. It is somewhat surprising that only 1.7% showed extreme aging of multiple organs, particularly given the clinical observation that co-morbidities across organs are common among age-related disease phenotypes. Is it possible that this is a result of the approach in that mutually exclusive single organ-specific proteins were used? There are examples of “oligo-organ enriched” proteins (i.e. proteins enriched across a small number of organs). Perhaps analyses that included such proteins would identify more subjects with concordant, accelerated aging across several interrelated organs (heart-kidney, heart-brain, etc).

This reviewer makes a good point, that based on the epidemiological studies of comorbidities, there should be more multi-organ agers in the population. We actually find this to be true based on our data as well, just not in the “extreme” aging framework. When looking at all individuals, not just the extreme agers, we find our data aligns with the epidemiology - individuals with heart disease, diabetes, hypertension, Alzheimer’s disease are aged across many organs (Extended Data Fig. 4e), though they are still primarily aged in their respective relevant organs.

Extended Data Figure 4.

a, Extreme agers largely cluster into distinct groups when considering individuals with organs above the extreme aging threshold, absolute z-score > 2.

b, Mean aging profile per extreme aging group shows there are not pairs of extremely aged organs at the population level.

d, A cross-cohort meta-analysis of associations between extreme ageotypes versus diagnosis of 9 major age-related diseases annotated in at least 2 independent cohorts, controlling for age and sex (logistic regression model: $\text{AgeGap} \sim \text{Disease} + \text{Age} + \text{Sex}$). Log odds ratios and significance are shown. P-values were Benjamini Hochberg corrected. Asterisks represent q-value thresholds: *q < 0.05; **q < 0.01; ***q < 0.001. The strongest associations per disease are highlighted with black borders.

e, A cross-cohort meta-analysis of associations between organ age gaps versus diagnosis of 9 major age-related diseases annotated in at least 2 independent cohorts, controlling for age and sex (linear model: $\text{AgeGap} \sim \text{Disease} + \text{Age} + \text{Sex}$). Disease covariate effects and significance are shown. P-values were Benjamini Hochberg corrected. Asterisks represent q-value thresholds: *q < 0.05; **q < 0.01; ***q < 0.001. The strongest associations per disease are highlighted with black borders.

The fact that extreme multi-organ agers make up 1.7% of the population specifically is partially due to our mathematical definition of “extreme agers” as individuals with a 2-standard deviation increase/decrease in at least one organ age gap. Since a +2 standard deviation is roughly equivalent to the 2nd percentile under a normal distribution, and since we find very few individuals with extreme youth in any organ (-2 standard deviations, <1% of the population), it makes sense that we find that each organ ager group is ~2% of the population.

However, it is somewhat surprising that there are not more individuals with extreme aging in pairs of organs or a small number of related organs. This is indeed what we see in the data, as shown in Extended Data 4a and 4b, where the clustering process and average aging profile per organ for each extreme organ aging type are shown. What we find is that there is not consistent co-occurrence of paired organ aging at the population level which easily explains epidemiological co-occurrence of disease, though it does not mean that specific individuals never have multiple aged organs. This reviewer makes an interesting comment about “oligo-organ enriched proteins” to model multi-organ aging. We have thought about this as well and think different combinations of oligo-organ aging models would be interesting to follow-up on, but for simplicity, in this manuscript we decided to generate an “organismal” model which captures proteins expressed by multiple organs. We indeed find that organismal agers cluster more closely with multi-organ agers (Fig. 1e, Extended Data Figure 4a-b).

4. These observations also raise questions about the boundary between normal aging and organ pathology. It seems possible that at least some of the subjects identified as having accelerated aging of a specific organ, actually have subclinical disease in that organ, which has altered the plasma proteome. For example, for cardiac age, NT-BNP and Troponin were major contributors to the model. While there is some (not entirely consistent) evidence each of these can increase in healthy aging, they are also indicators of cardiac pathology and already recognized predictors of adverse cardiovascular outcomes. It seems plausible that these analyses include subjects who have existing disease that hasn't yet presented

clinically and thus as labeled as ‘healthy’. This might be revealed by more detailed phenotyping (e.g. cardiac MRI or even echo with GLS). Assuming such phenotyping data are not available, this possibility should at least be acknowledged and discussed. The clinical implications may still be similar (i.e., identification of subjects at high risk who should be considered for preventive measures) but the interpretation, particularly as it relates to the biology of normal aging and uncovering the responsible mechanisms, might be considerably different.

This reviewer brings up a great point about how individuals who seem “healthy” may have actually accumulated some degree of heart pathology already and that perhaps NT-BNP and troponin are detecting this. We agree with this notion and believe that it is the central hypothesis of the aging field: that the “normal” aging process is the progressive accumulation of damage that results in a gradual change from healthy to subclinical disease to active disease.

We believe NT-BNP and Troponin are great examples of this hypothesis. They increase with normal aging across all cohorts in our study (Reviewer Figure 2) and in published studies totaling over 50,000 people of diverse ancestry across the adult lifespan¹⁻⁴, they are associated with future heart failure and mortality over 15-year follow-up (Fig. 2i-j), and they are also established biomarkers of acute heart damage, suggesting that increased levels in “healthy” individuals may represent varying degrees of subclinical disease.

Reviewer Figure 2
Change with age of NT-proBNP and Troponin-T in individuals without history of cardiovascular disease.

As this reviewer suggests, it would be amazing to associate the heart age gap with more detailed phenotypes of heart pathology to more thoroughly test associations between accelerated normal aging and subclinical disease, as the ability to detect subclinical disease with plasma proteins would improve monitoring of disease risk in seemingly healthy individuals. We have added the following sentence in lines 164-171 to reflect the importance of this future study:

“Heart aging proteins were expressed primarily by cardiomyocytes (Fig. 2g-h) and had known roles in heart biology and disease. Pro-brain natriuretic peptide (NPPB), a negative regulator of blood pressure that increases in response to heart damage, and troponin T (TNNT2), a heart muscle protein involved in contraction, had the strongest weights in the heart aging model (Fig. 2g). They are both established clinical markers of acute heart failure²², and NPPB has been previously associated with heart attack risk³⁷. This suggests the possibility of a link between subclinical heart disease and the “normal” heart aging process, which should be investigated further with more detailed heart imaging and electrophysiology.

Referee #6 (Remarks to the Author):

However, some outstanding comments still remain that need to be addressed:

1. To support their conclusions and findings, the manuscript relies heavily on the assignment of proteins to specific tissues/organs.

“We mapped the organ-specific plasma proteome using human organ bulk RNA-seq data from the Genotype-Tissue Expression (GTEx) project. We classified genes as “organ enriched” if they were expressed at least 4 times higher in one organ compared to any other organ, according to the definition proposed in the Human Protein Atlas”

A possible problem using this approach would arise from the fact that the correlation between mRNA levels and protein levels is typically poor in human tissues as well as shown by animal studies. See for example, a recent proteome/ transcriptome analysis of 32 human tissues that showed a median Spearman correlation of only 0.46 and between 90.2 and 0.6 roughly (Jiang et al., Cell 2020 Jiang et al., 2020 Cell 183, 269-283). Thus, it remains puzzling what it means for protein levels that mRNAs are expressed 4 times more in one cell or tissue or organ than in other?

We thank this reviewer for their concern about our approach. We recognized before starting our study that RNA levels are not always correlated with protein levels, and that this may hamper interpretability. Because of this, we considered determining organ-specificity based on protein levels from the publication mentioned, Jiang et al.⁵

We ultimately decided to determine organ-specificity based on a 4-fold cutoff from bulk RNA-seq data from the Gene Tissue Expression (GTEx) Atlas for three main reasons:

1. Determining organ specificity based on a 4-fold increase in RNA-seq expression (“organ-enriched”) from GTEx and other databases is a well-accepted approach, established by the Human Protein Atlas (HPA) in multiple studies⁶⁻⁸. In their *Science* 2015 study, they evaluated three tiers of organ-specificity:
 - a. Tissue-enriched (highly specific): 4-fold higher in one organ compared to any other organ.
 - b. Group-enriched (mildly specific): 4-fold higher in the average of 2-5 organs compared to any one organ.
 - c. Tissue-enhanced (lightly specific): 4-fold higher in one organ compared to the average of all other organs.

They showed that tissue-enhanced genes are significantly enriched for biological pathways involved in the function of their tissue organs (Uhlén et. al, *Science* (2015), Figure 2p). We find that organ-enriched genes are even more significantly enriched for relevant pathways, which we now include in a Supplementary Booklet. The tissue-enriched metric from the HPA is widely trusted and is provided in NCBI, GeneCards, and enrichment analysis tools such as gprofiler⁹. We believed this proof of concept was sufficient in showing that the established methodology enables interpretable, relevant findings for the biology of specific organs. Thus, we used the same highest tier of specificity (tissue-enriched), but with the updated, more deeply sequenced GTEX RNA-seq dataset and with a more generalizable framework for tissue->organ mapping. The Supplementary Booklet is now referenced in main text, lines 111-112.

2. Organ protein levels may be misleading in regard to determining the original organ source of the protein. Specifically, a protein may be present in an organ because it was trafficked there after being synthesized by another organ and secreted into the plasma. Jiang et al showed nicely that many proteins which are synthesized exclusively in one organ are not enriched at the protein level in that organ because they are secreted to another⁵. This is easy to understand in organs like the liver, pancreas, and pituitary, which are all examples discussed in Jiang et al. Albumin and complement proteins are not enriched at the protein level in the liver despite the fact that they are synthesized there, and there are proteins which are synthesized in the hypothalamus that are enriched in the pituitary because they are stored there before release. In the pancreas there is high correlation between the RNA and protein levels of organ-enriched digestive enzymes because they are stored locally in the pancreas. Generally, discordance between protein and RNA levels is interpreted in Jiang et al and other studies^{6,7,10} as a result of protein trafficking/export/secretion, while enrichment at the RNA level is recognized as the tissue of origin for protein synthesis. It may also be true that proteins which are present at the protein level in an organ but are not synthesized there also contain important information about said organ. We believe this idea of cross-organ communication in aging is an exciting area for future study. For the current manuscript, our goal was to determine the putative organ source of plasma proteins to infer organ age.
3. RNA-seq data contains nearly full coverage of the genome, while proteomics data has much lower coverage. In Jiang et al, only 6320 proteins were detected in >50% of samples, and these are heavily biased towards abundant proteins, which are detectable by mass spectrometry. The percentage of these mappable to the plasma proteome is even lower, and the percentage that overlaps with the SomaScan assay, which can detect much lower abundance proteins, is even lower. Determining organ-specificity based on RNA-seq data increased our coverage of the mappable organ-specific plasma proteome.

We added now an abbreviated summary of these points to the Supplementary Text.

2. Do the authors know how much proteins can be found in the studied tissues/organs, and how this would then translate to protein plasma levels? For example, a protein X could be more enriched in a

tissue Y, but the changes in plasma levels might be coming from many different tissues; meaning many tissues may contribute to the plasma level.

This is a very interesting question, which we hope can one day be tested in humans. As of now, to our knowledge, the precise mechanisms by which different organs export different proteins into the blood is largely unknown.

While not perfect, for this reason, we decided to use the organ-enriched framework to identify proteins that are most likely derived from a single organ source, because if a protein is expressed primarily in one organ, its levels in the blood likely reflect changes in that organ source. To minimize concerns about multiple potential organ sources, we did not include group-enriched proteins – which are enriched in pairs or groups of organs – or any other proteins of lower organ specificity in our analyses.

Fortunately, despite our inability to experimentally confirm the precise organ source of proteins, our RNA-based inference approach has yielded several highly significant associations between organ aging models and organ function and has revealed several organ-specific proteins expressed by various cell types in those organs. Still, we believe future studies are needed to better understand the biology of specific proteins, including where they come from, how they end up in plasma, and what they do in their organ source versus peripheral organs.

3. The Somascan assay provides an indirect detection and measurement of proteins. This is not mass-spectrometry (MS) where the sequence positively identifies the protein. Here, the proteins were detected by binding to a probe that might have non-specific binders. This happens mainly when some proteins are really abundant and override the affinity of the probe for a specific protein.

The authors said that this platform has been used multiple times and that this provides “validation”. However this indeed may not validate the platform. I’d say that one should do some sort of validation by MS to be more confident. Since the platform provides so-called a directed assay by selecting the targets it does not have a power to be discovery-free as the MS. The main advantage might be the cost, but it has severe limitations as well.

We thank this reviewer for their concern about the proteomics platform. The reviewer is correct in that there are limitations of the SomaScan assay and simple adoption in the field does not necessarily provide validation. Nevertheless, the company has analyzed close to 1 million samples with their technology for academics, well over 100 biotech and pharma companies including almost all top 25 pharma, resulting in some 700 publications (<https://somalogic.com/publications/>). I should note, that my lab has no commercial ties to Somalogic but after being frustrated by the inability of MS to detect most cytokines, chemokines and other key signaling proteins in plasma up to this day, we started by using filter-based antibody platforms (Raybiotech), then moved to bead-based assays (Luminex) before adopting Somalogic and Olink as the best technologies today.

Of course, all proteomics platforms have limitations. We carefully weighed the strengths and limitations of mass spectrometry, antibody-based assays, and the SomaScan assay before initiating our study.

We ultimately decided to move forward with the SomaScan assay for many reasons including, but not limited to:

- A. The assay is sensitive to the very large dynamic range of the human plasma proteome (see included technical note).
- B. The coverage of the proteome is the largest among plasma proteomics assays by several thousands of proteins (see included technical note). Mass spec is heavily limited to measuring mid to high abundance proteins and is less sensitive to dynamic range¹¹. For plasma it requires either fractionation of samples or depletion with very costly immunoaffinity columns that typically deplete the top 40 most abundant protein and often results in missing values^{11,12}. Very few labs would be able to process 1000s of samples this way and there is not independent validation possible since each lab uses their own custom pipeline.
- C. There is minimal replicate sample variability^{13,14} (coefficient of variation, CV). The majority of SomaScan protein measurements are stable and a subset of proteins have been validated as Lab Developed Tests (LDTs), and have been delivered out of Somalogic's CLIA-certified lab to physicians and patients in the context of medical management¹⁵. One of these LDTs, the Residual (secondary) Cardiovascular Risk Test has also advanced to FDA pre-submission as an IVD and qualified biomarker drug development tool (DDT).
- D. There has already been extensive validation of the assay not just in the 700 publications mentioned above and use by pharma and clinicians¹⁶, but via experimentation including comparisons with mass spec, antibodies, etc. In addition to their primary validation pipeline for all probes, 70% of probes have at least one orthogonal source of validation.

Since this specific reviewer comment focuses on lack of validation of the assay, we detail the extensive validation that has already been performed, below and in the attached PDF "SOMAmer confirmation and validations.pdf":

1. All ~7,500 probes on the assay undergo rigorous primary validation of binding and sensitivity to the target protein.
 - a. Determination of equilibrium binding affinity dissociation constant (K_D).
 - b. pull down assay of cognate protein from buffer
 - c. Demonstration of dose-responsive in the SomaScan Assay
 - d. Estimation of endogenous cognate protein signals in human plasma above limit of detection.
2. 70% of their probes have at least one orthogonal source of validation from:
 - a. mass-spectrometry: ~900 probes which measure mostly high and mid abundance proteins (due to sensitivity limitations of Mass Spec), have been confirmed with either DDA or MRM Mass Spec.
 - b. antibody: ~390 probe measurements correlate with antibody-based measurements.

- c. cis-pQTL: ~2,860 probe measurements are associated with genetic variation in the cognate protein-encoding gene.
- d. absence of binding with nearest neighbor: ~1,150 probes do not detect signal from the protein that is most closely related in sequence to the cognate protein.
- e. Correlation with RNA: ~1,460 probe measurements correlate with mRNA levels in cell lines.

We have now added this additional information to the methods section, lines 543-569, and the above figure has been added as Supplementary Figure 1b.

4. The targets are preselected and cover roughly only 4% of the human proteome. Checking online I couldn't get the whole dataset from there. The company lists panels, such as neurological, cardio, metabolic, etc... similar to Olink (although the technology is different). I guess that the authors mixed several panels, but this has not been explained in the manuscript. One can have proteins that are really abundant, relevant, markers, etc... but that they don't quantitate because they have not been included in the panel. So this is a major limitation.

We agree, the ability to measure the entire human proteome and all its isoforms and post-translational modifications (with estimates from 10s of thousands to billions) would be an incredible feat, which we cannot do given our current technology. To make an attempt at covering the largest breadth of the human plasma proteome given current limitations, we decided to use the SomaScan assay, which measures several thousands more proteins than antibody-based assays and mass spec in human plasma. These technologies are rapidly improving, with Somalogic having ~10,000 proteins and Olink having

~5,400 proteins on their next assay versions. We have added a statement in the manuscript, lines 372-375 recognizing the current limitations.

“Our current models rely on ~5,000 proteins measured with the SomaScan assay, but the approach is platform agnostic, and we expect that even more biological information could be gained with additional proteomic coverage, including cell and organ-specific splice isoforms and post-translational modifications.”

In regard to the comment about SomaScan panels, there seems to be some confusion. Somalogic does not provide separate panels, like Olink. Somalogic provides measurements for all proteins on the assay for all samples.

5. Overall, the analysis was done with a low number of proteins, 856 organ enriched, 17.9% of their plasma detected proteins and around a 4% of the proteome. Really low numbers in general.

As mentioned in the response to the previous comment, we are limited by the breadth of the proteome that can be measured by current technologies. It should also be noted that Somamers and antibodies (as well as mass spec) typically measure multiple proteoforms for any given protein. This could include different isoforms, posttranslational modifications, etc. The same is true for ELISAs unless the reagents are designed against a very specific isoform of a protein. We have opted to use the SomaScan assay, which has the largest coverage compared to other platforms. Here, we measured 4,979 plasma proteins per individual, which covers roughly 25% of the human proteome and is one of the largest plasma proteomic studies of aging to date.

Using these numbers of plasma proteins, we discover highly significant and reproducible associations between organ aging and disease, which have important implications in the monitoring of human health. Importantly organ aging models, which were trained on a lower number of proteins than the conventional aging model, outperformed the conventional aging model in disease associations, suggesting number of proteins is not as crucial as the purposeful selection of proteins.

6. Technically, I didn't see information about how the blood samples were collected for different cohorts. Some refer to previous papers, but there is a potential concern how comparable were the extraction conditions and plasma preparation, storage, etc.

This reviewer brings up a valid concern that different methods for blood processing across cohorts can affect protein levels, leading to batch effects. Though this is most likely true, our model training workflow is designed to minimize contributions from noisy proteins and select for proteins most significantly associated with age. We find that our approach is robust, as all our figures show consistent results across multiple cohorts that are resilient to these possible batch effects.

All cohorts collected blood based on typical medical practice, though slight variations in norms may be present between cohorts. Details on exact blood processing from unpublished cohort data are available in our methods (Stanford-ADRC, SAMS, Knight-ADRC). We have added additional information from the

Knight-ADRC, lines 506-509. Details for datasets in published studies (Covance, LonGenity) can be found in the cited publications and below:

- **lines 460-474, Stanford-ADRC**
“Blood collection and processing were done according to a rigorous standardized protocol to minimize variation associated with blood draw and blood processing. Briefly, about 10 cc whole blood was collected in a vacutainer EDTA tube (BD Vacutainer EDTA tube) and spun at 3000RPM for 10 mins to separate out plasma, leaving 1 cm of plasma above the buffy coat and taking care not to disturb the buffy coat to circumvent cell contamination. Plasma processing times averaged approximately one hour from the time of the blood draw to the time of freezing and storage. All blood draws were done in the morning to minimize the impact of circadian rhythm on protein concentrations. Plasma pTau-181 levels were measured using the fully-automated Lumipulse G 1200 platform (Fujirebio US, Inc, Malvern, PA) by experimenters blind to diagnostic information, as previously described.”
- **lines 486-494, SAMS**
“SAMS is an ongoing longitudinal study of healthy aging. Blood collection and processing were done by the same team and using the same protocol as in Stanford-ADRC”
- **lines 506-509, Knight-ADRC**
“Blood samples were collected in EDTA tubes (BD Vacutainer purple top) at the visit time, immediately centrifuged at 1500g for 10 minutes, aliquoted on 2D barcoded Micronic tubes (200ul per aliquot) and stored at – 80°C. The plasma was stored in monitored -80C freezer until it was pulled and sent to Somalogic for data generation.”
- **Ref 77, Covance**
“EDTA plasma samples had been collected from all these studies and the samples were centrifuged and frozen typically 2–10 h after collection, a timeframe that is representative of how blood is handled in typical medical practice. Aliquots of these samples were assayed on the proteomic platform without further processing after transport and thawing.”
- **Ref 79, LonGenity**
“Plasma was isolated from EDTA-treated blood acquired by venipuncture from participants at baseline wave in a fasting state. Plasma samples were stored at –80°C, and 150 µl of aliquots of plasma was sent to SomaLogic on dry ice.”

7. The cohorts are very different, irrespective of their individual collection purposes (like ALZ) in terms of age range, genetic background and number of participants, which potentially skew data analysis and interpretation.

Though our cohorts are indeed different, we find consistent, robust associations between organ age gaps with disease. This points to the power in our approach for uncovering generalizable human biology. We report all statistics per cohort to make it clear where cohort effects may play a role and visualize main figures with meta-analytic plots that display cohort effects when possible. No results highlighted in the text or figures are due to cohort-level effects. We agree that it would be interesting to follow-up and

investigate cohort differences and how that may be explained by cohort specific genetic and environmental variables.

8. The authors state:

We trained our models in 1,398 cognitively unimpaired participants from the Knight-ADRC cohort. We evaluated their performance in the Covance (n = 1,029), LonGenity (n = 962), SAMS (n= 192), Stanford-ADRC (n=409) cohorts, and Knight-ADRC cognitively impaired subjects (n = 1,677)

But, do we know how the age-gap for each organ was performed? The authors say that one can use plasma proteome to determine the age-gap and refer to previous papers, but applying the same concept to organs could be a stretch without proper validation and evidence. It is also not clear to this reviewer how the authors established the chronology for aging in different organs, particularly from the cohorts that includes individuals of different ages, but the numbers of individuals at different ages within different cohorts are not shown. For example, the authors state that they used the KADRC to train the model. This group has individuals from 27-104 with the SD of 75 yrs; it is not shown how many young individuals are in this group. Similarly, they used only 1398 “healthy patients”, but they don’t give the age range for this subset either.

We apologize for the confusion on age ranges. We have now made separate columns for the healthy versus Alzheimer’s disease individuals in the Knight-ADRC cohort in the demographics table (Supplementary Table 1). The age range, mean, and standard deviation for all cohorts are clearly shown. A more visual sense of the overall per-cohort age distribution is shown in Supplementary Figure 8a.

In regard to the confusion on the methodology of deriving organ age gaps, we described the derivation of the aging models and age gap calculations in precise detail in our methods section with visualizations and results in Extended Data Figure 2 and Supplementary Figure 3.

Briefly, we trained models to predict age in healthy individuals in one cohort and predicted age across several independent cohorts. For each cohort, we calculated the age gap, which is defined as the deviation between predicted age and the regression curve between predicted and chronological age. We derived a different predicted vs chronological regression curve for each cohort and each aging model to normalize for cohort differences and aging model prediction accuracy, respectively. Perhaps this is the detail that was missed. This allowed us to derive an age gap even for individuals outside the age of the training cohort age range. The full details are below.

Methods lines 583-618:

“Bootstrap aggregated LASSO aging models

To estimate biological age using the plasma proteome, we built LASSO regression-based chronological age predictors (Extended Data Fig. 2-3, Supplementary Fig. 3) using the scikit-learn⁹⁰ python package. We employed bootstrap aggregation for model training. Briefly, we resampled with replacement to generate 500 bootstrap samples of our training data (Knight-ADRC: 1,398 healthy

individuals). Each bootstrap sample was the same size as the training data, 1,398. For each bootstrap sample, we trained a model on z-scored \log_{10} normalized protein expression values with sex (F=1, M=0) as a covariate to predict chronological age. For model training, we performed hyperparameter tuning of the L1 regularization parameter, λ , with 5-fold cross validation using the GridSearchCV function from scikit-learn. To reduce model complexity and avoid overfitting, we selected the highest λ value that retained 95% performance relative to the best model. The mean predicted age from all 500 bootstrap models was used.

We trained our models in 1,398 cognitively unimpaired participants from the Knight-ADRC cohort. We evaluated their performance in the Covance (n = 1,029), LonGenity (n = 962), SAMS (n= 192), Stanford-ADRC (n=409) cohorts, and Knight-ADRC cognitively impaired subjects (n = 1,677). Models that included sex as a covariate and models trained separately on males and females showed similar age prediction performance on both sexes, so we controlled for sex to extend the generality of the findings and reduce analytic complexity (Supplementary Fig. 3 a-c). There was a correlation between age estimation accuracy and the number of proteins used as input to each model (Supplementary Fig. 3 c-d). However, several models with few protein inputs, such as the adipose (5 proteins) and heart models (10 proteins), predicted chronological age better than models with more protein inputs (Extended Data Fig. 3).

Age gap calculation and independent validation

To calculate each individual sample age gap for each aging model, we performed the following steps for each aging model. We fit a local regression between predicted and chronological age using the lowess function from the statsmodels⁹¹ python package with fraction parameter set to 2/3 to estimate the true population mean (Supplementary Fig. 3e). A local regression is used in place of a simple linear regression because of extensive evidence that the plasma proteome changes non-linearly with age¹, which we see replicated in all 5 cohorts (Supplementary Fig. 8). Individual sample age gaps were then calculated as the difference between predicted age and the lowess regression estimate of the population mean. Age gaps were calculated separately per cohort to account for cohort differences (Supplementary Fig. 3e). Age gaps were z-scored per aging model to account for the differences in model variability (Supplementary Fig. 3f). This allowed for direct comparison between organ age gaps in downstream analyses.”

Main text lines 90-91, 96-99:

We and others have previously shown that plasma proteins can be used to train machine learning models to estimate chronological age in independent cohorts... Based on this concept, we trained a bagged ensemble of least absolute shrinkage and selection operator (LASSO) aging models for 11 major organs using the mutually exclusive organ-enriched proteins we identified as inputs (Fig. 1a, Extended Data Fig. 2a-b, Supplementary Fig. 3, Supplementary Table 6-8).”

We also recognize our study has limitations in that the training cohort is skewed towards older individuals and therefore, may not apply equally well to young individuals. This concern was brought up in the first round of review. We acknowledge this limitation in lines 391-392: “While our current models serve as a proof of principle for this approach, since they are trained and evaluated largely on older adults, caution should be used when applying them to young people.” If there are additional concerns about cohort age ranges, we refer this reviewer to Reviewer 4 comments 6-7 from the first round of review.

9. They seem to be “very precise” by defining the age gap, for example:

Individuals with hypertension had kidneys that were approximately 1 year older than their same-aged peers, while individuals with diabetes had kidneys approximately 1.3 years older (Fig. 2a-b, Supplementary Table 8,10)

With just a few “organ specific” proteins can one have such a level of exactness? For example the paper uses only 12 kidney-specific proteins and established 1 year older. What’s the error of this calculation?

We apologize for the confusion on the age gap statistics in the main text. To clarify, for each disease we tested the following linear model: $z\text{-scored age gap} \sim \text{disease (1 or 0)} + \text{age} + \text{sex}$. The coefficient for the disease variable provides an estimate of the mean difference in z-scored age gaps between disease and control groups, controlled for age and sex. We then converted the mean difference in z-scored age gaps to mean difference in raw age gaps based on the standard deviation (Supplementary Table 8) of the raw age gaps. This raw mean difference in age gaps statistic is what is provided in the text. The exact statistic for the association between hypertension and kidney aging is 1.022 ± 0.075 years. For diabetes it is 1.270 ± 0.119 years, based on the standard errors from the associated models. Coefficients, standard errors, confidence intervals, p-values, and other relevant statistics are provided in Supplementary Tables (8-13 and 20-24).

We have added the following to the methods section, lines 658-666, to improve clarity:

“Most organ age gap vs trait associations in this study (Fig. 2a-d; Fig. 3c; Extended Data Fig. 4d-e, Extended Data Fig. 5c; Extended Data Fig. 6b-c; Extended Data Fig. 7, Extended Data Fig. 8c-d, Extended Data Fig. 9) were assessed using linear models controlled for age and sex as follows: $\text{age gap} \sim \text{trait} + \text{age} + \text{sex}$ and adjusted for multiple testing burden using the Benjamini–Hochberg method when appropriate. To describe disease associations in relation to years of additional aging in the main text, we took the coefficient for the trait variable – which provides an estimate of the mean difference in z-scored age gaps between disease and control – and converted that to an estimate of mean difference in raw age gaps, using the standard deviation of raw age gaps provided in Supplementary Table 8.”

10. The usual naming of handpicked molecules and generalizations.

“synaptic proteins complexin 1 (CPLX1), complexin 2 (CPLX2), and neurexin 3 (NRXN3) – which all have genetic links to cognition and AD44–48, and stathmin 2 (STMN2) and olfactomedin 1 (OLFM1) – which are involved in neurite outgrowth and axon growth cone collapse^{49,50}”

One can hand pick any neuronal protein and find correlation to disease. What is the biological meaning of this kind of statements?

Thank you for this question. To clarify, these proteins were not hand picked or cherry picked. They were the most important proteins that were selected by the brain aging model and by our feature importance

method, FIBA in an unbiased fashion. This is presented in the text, in Fig 3a-b, and in Supplementary Table 18. Importantly, not all neuronal proteins in our data changed with age or cognition (Supplementary Table 15, 18, 24), suggesting the ones that were selected have particular importance in aging and cognitive decline and deserve additional study. The cited studies above merely lend additional support to the potential role of these proteins in cognitive aging and Alzheimer's disease but are not meant to be the core message. As an aside, reviewers in the first round of reviews wanted us to highlight individual proteins and discuss their possible role in aging and disease.

References

1. Ferkingstad, E. *et al.* Large-scale integration of the plasma proteome with genetics and disease. *Nat. Genet.* **53**, 1712–1721 (2021).
2. Walker, K. A. *et al.* Large-scale plasma proteomic analysis identifies proteins and pathways associated with dementia risk. *Nat. Aging* **1**, 473–489 (2021).
3. Pietzner, M. *et al.* Mapping the proteo-genomic convergence of human diseases. *Science* **374**, eabj1541 (2021).
4. Sun, B. B. *et al.* Genomic atlas of the human plasma proteome. *Nature* **558**, 73–79 (2018).
5. Jiang, L. *et al.* A Quantitative Proteome Map of the Human Body. *Cell* **183**, 269–283.e19 (2020).
6. Uhlén, M. *et al.* Tissue-based map of the human proteome. *Science* **347**, (2015).
7. Uhlén, M. *et al.* The human secretome. *Sci. Signal.* **12**, (2019).
8. Uhlen, M. *et al.* A genome-wide transcriptomic analysis of protein-coding genes in human blood cells. *Science* **366**, (2019).
9. Raudvere, U. *et al.* g:Profiler: a web server for functional enrichment analysis and conversions of gene lists (2019 update). *Nucleic Acids Res.* **47**, W191–W198 (2019).
10. Liu, Y., Beyer, A. & Aebersold, R. On the Dependency of Cellular Protein Levels on mRNA Abundance. *Cell* **165**, 535–550 (2016).
11. Ignjatovic, V. *et al.* Mass Spectrometry-Based Plasma Proteomics: Considerations from Sample Collection to Achieving Translational Data. *J. Proteome Res.* **18**, 4085–4097 (2019).
12. Tu, C. *et al.* Depletion of Abundant Plasma Proteins and Limitations of Plasma Proteomics. *J. Proteome Res.* **9**, 4982–4991 (2010).
13. Katz, D. H. *et al.* Proteomic profiling platforms head to head: Leveraging genetics and clinical traits to compare aptamer- and antibody-based methods. *Sci. Adv.* **8**, eabm5164 (2022).
14. Candia, J., Daya, G. N., Tanaka, T., Ferrucci, L. & Walker, K. A. Assessment of variability in the plasma 7k SomaScan proteomics assay. *Sci. Rep.* **12**, 17147 (2022).
15. SomaSignal Tests - Products and Services. *SomaLogic* <https://somalogic.com/somasignal-tests-for-research-use/>.
16. Publications - SomaLogic, Inc. *SomaLogic* <https://somalogic.com/publications/>.

Reviewer Reports on the Second Revision:

Referees' comments:

Referee #5 (Remarks to the Author):

The authors have done a good job of responding to the questions raised. I have no further concerns or questions. I still find this study highly innovative and interesting with both fundamental and clinical implications that are potentially important.

Referee #6 (Remarks to the Author):

The authors have comprehensively addressed all my concerns and implemented the necessary changes. I endorse the publication of the revised manuscript and extend my congratulations for their outstanding work.